# Sensitivity of GNSS tropospheric gradients to processing options

Michal Kačmařík[1], Jan Douša[2], Florian Zus[3], Pavel Václavovic[2], Kyriakos Balidakis[3], Galina Dick[3], Jens Wickert[3,4]

[1] Department of Geoinformatics, VŠB – Technical University of Ostrava, Ostrava, The Czech Republic
[2] Geodetic Observatory Pecný, Research Institute of Geodesy, Topography and Cartography, Zdiby, The Czech Republic
[3] GFZ German Research Centre for Geosciences, Potsdam, Germany
[4] Institute of Geodesy and Geoinformation Science, Technical University of Berlin, Germany

*Correspondence to*: M. Kačmařík (michal.kacmarik@vsb.cz)

**Abstract.** An analysis of processing settings impact on estimated tropospheric gradients is presented. The study is based on the benchmark data set collected within the COST GNSS4SWEC action with observations from 430 GNSS reference stations in central Europe for May and June 2013. Tropospheric gradients were estimated in eight different variants of GNSS data processing using Precise Point Positioning (PPP) with the G-Nut/Tefnut software. The impact of the gradient mapping function, elevation cut-off angle, GNSS constellation, observation elevation-dependent weighting and real-time versus post-processing mode were assessed by comparing the variants by each to other and by evaluating them with respect to tropospheric gradients derived from two numerical weather prediction models (NWM). Tropospheric gradients estimated in post-processing GNSS solutions using final products were in a good agreement with NWM outputs. The quality of high-resolution gradients estimated in (near) real-time PPP analysis still remains challenging task due to the quality of the real-time orbit and clock corrections. Comparisons of GNSS and NWM gradients suggest the 3° elevation angle cut-off and GPS+GLONASS constellation for obtaining optimal gradient estimates provided precise models for antenna phase centre offsets and variations and tropospheric mapping functions are applied for low-elevation observations. Finally, systematic errors can affect the gradient components solely due to the use of different gradient mapping functions, and still depending on observation elevation-dependent weighting. A latitudinal tilting of the troposphere in a global scale causes a systematic difference up to 0.3 mm in the north gradient component, while large local gradients, usually pointing to a direction of increasing humidity, can cause differences up to 1.0 mm (or even more in extreme cases) in any component depending on the actual direction of the gradient. Although the Bar-Sever gradient mapping function provided slightly better results in some aspects, it is not possible to give any strong recommendation on the gradient mapping function selection.

## 1 Introduction

When processing data from Global Navigation Satellite Systems (GNSS), a total signal delay due to the troposphere is modelled by epoch- and station-wise Zenith Total Delay (ZTD) parameters, and, optimally, together with tropospheric gradients representing the first order asymmetry of the total delay. ZTDs, which are closely related to Integrated Water Vapour

(IWV), are operationally assimilated into Numerical Weather Prediction models (NWM) and have been proven to improve precipitation forecasts (Vedel and Huang, 2004, Guerova et al., 2006, Shoji et al., 2009). Previous studies demonstrated that the estimation of tropospheric gradients improves GNSS data processing mainly in terms of receiver position and ZTDs (Chen and Herring, 1997, Bar-Sever et al., 1998, Rothacher and Beutler, 1998, Iwabuchi et al., 2003, Meindl et al., 2004). Nowadays, tropospheric gradients are not assimilated into NWMs, however, they could be assimilated in future (see Zus et al., 2019) and they are essential for reconstructing slant total delays (STD). The STDs represent the signal travel time delay between the satellite and the station due to neutral atmosphere and they are considered useful in numerical weather prediction (Järvinen et al., 2007, Kawabata et al., 2013, Bender et al., 2016) and reconstruction of 3D water vapor fields using the GNSS tomography method (Flores et al., 2000, Bender et al., 2011).

Brenot et al. (2013) showed a significant improvement of IWV interpolated 2D fields when tropospheric gradients are taken into account. With the improved IWV fields, the authors studied small scale tropospheric features related to thunderstorms. Douša et al. (2018a) demonstrated the advantage of using tropospheric gradients in the 2-stage troposphere model combining NWM and GNSS data. Morel et al. (2015) presented a comparison study on zenith delays and tropospheric gradients from 13 stations at Corsica Island in the year 2011. Despite a good agreement in the ZTD, they found notable discrepancies in tropospheric gradients when estimated by using two different GNSS processing software, two different gradient mapping functions, and two different processing methods: 1) double-differenced network solution, and 2) Precise Point Positioning, PPP (Zumberge et al., 1997) solution. Douša et al. (2017) indicated a problem with systematic errors in tropospheric gradients due to absorbing instrumentation errors. Few attempts were made to compare the tropospheric gradients with independent estimates, i.e., those derived from Water Vapor Radiometer (WVR) or NWM data. For a selected number of stations such a comparison was made in Walpersdorf et al. (2001) where ZTDs and tropospheric gradients from GPS were compared with those derived from a high-resolution NWM ALADIN. A good correlation between GPS and NWM gradients was found for inland stations, but not for coastal ones. More recently Li et al. (2015) and Lu et al. (2016) showed that with the upcoming finalization of new systems such as Galileo and BeiDou the improved observation geometry yields more robust tropospheric gradient estimates. Li et al. (2015) found an improvement of about 20~35% for the multi-GNSS processing when compared with NWM and 21~28% when compared to WVR. Another multi-GNSS study on tropospheric gradients (Zhou et al., 2017) used data from a global network of 134 GNSS stations processed in six different constellation combinations in July 2016. An impact of gradients estimation interval (from 1 to 24 h) and cut-off elevation angle (between 3° and 20°) on a repeatability of receiver coordinates was examined. Better results were found for solutions where a shorter time interval of tropospheric gradient estimation was used and where the elevation cut-off angle of 7° or 10° was applied. However, strategies were not compared from the point of view of actually obtained gradient values. Finally, systematic differences and impacts of a gradient mapping function or observation elevation weighting on estimated gradients have not been studied yet.

In this work, we systematically evaluate the quality of tropospheric gradients estimated from a regional GNSS dense network under different atmospheric conditions. Using a unique data set, we study the impact of several approaches. ZTDs and tropospheric gradients are then compared with the ones estimated from two NWMs – ERA5, which is a global atmospheric

reanalysis, and a limited area short range forecast utilizing the Weather Research and Forecasting (WRF) model. Finally, we quantified systematic differences in tropospheric gradients coming from the gradient mapping function and the method of observation weighting during a local event with strong wet gradients.

## 2 Data and Methods

## 2.1 Benchmark data set

The benchmark campaign was realized within the European COST Action ES1206 GNSS4SWEC to support development and validation of a variety of GNSS tropospheric products. An area in central Europe covering Germany, the Czech Republic and part of Poland and Austria was selected as a domain while May and June 2013 as a suitable time period due to occurrence of severe weather events including extensive floods. Data from 430 GNSS stations were collected together with meteorological

observations from various instruments (synoptic, radiosonde, WVR, meteorological radar, etc.). In addition, tropospheric parameters from two global and one regional NWMs were generated. Detailed information about the benchmark campaign can be found in Douša et al. (2016). Although the presented study is based on the GNSS data collected within the benchmark campaign, all the presented GNSS and NWM solutions were newly prepared for this study.

## 2.2 Estimation of tropospheric gradients from GNSS

The STD as a function of the azimuth (*a*) and elevation (*e*) angle can be written as follows:

$$STD(a,e) = mfh(e) * ZHD + mfw(e) * ZWD + mfg(e) * (Gn * cos(a) + Ge * sin(a)) \quad\quad (1)$$

where ZHD denotes the Zenith Hydrostatic Delay and ZWD denotes the Zenith Wet Delay. The elevation angle dependency is given by mapping functions, which are different for the hydrostatic (*mfh*), wet (*mfw*) and gradient (*mfg*) part. The tropospheric horizontal gradient vector is defined in the local horizontal plane with two components, one for the north-south

direction (*Gn*) and one for the east-west direction (*Ge*). From the formula (1) is evident that GNSS gradient represents a gradient of both hydrostatic and wet part of the delay, therefore a total delay gradient.

During GNSS data processing, the ZHD is commonly taken from an a priori model, e.g. Saastamoinen (1972) or Global Pressure and Temperature (GPT, Boehm et al., 2007a) based on climatological data, or it can be derived from NWM data. The ZWD, or a correction to the modelled ZHD, and tropospheric gradients are estimated as unknown parameters using a

deterministic or stochastic model.

Current mapping functions for hydrostatic (*mfh*) and wet (*mfw*) delay components are based either on climatological data, e.g. Global Mapping Function, GMF (Boehm et al., 2006a) or NWM data, e.g. Vienna Mapping Function, VMF (Boehm et al., 2006b). An advantage of the first approach is its independence of external data. Several mapping functions for tropospheric gradients have also been developed in the past, e.g. by Bar-Sever et al. (1998), by Chen and Herring (1997), or the tilting

mapping function introduced by Meindl et al. (2004). The gradient mapping function (*mfg*) by Bar-Sever (BS) is given as

$$mfg = mfw * cot(e) \quad\quad\quad\quad\quad\quad\quad\quad\quad\quad\quad\quad\quad\quad (2)$$

and from the formula is apparent that it depends on the selected *mfw*. The Chen and Herring (CH) *mfg* reads as

$$mfg = 1 / (sin(e) * tan(e) + c) \tag{3}$$

where $c = 0.0032$. Since $c$ is related to the scale height, it experiences spatiotemporal variations. Nevertheless, based on Balidakis et al. (2018) a variable $c$ does not yield a statistically significant improvement in describing the atmospheric state over a constant $c$. Finally, the tilting mapping function is defined in a generic way as a derivative of the *mfw* with respect to the elevation angle

$$mfg = -\partial(mfw)/\partial e \tag{4}$$

Figure 1 illustrates the variability of the gradient contribution term $(Gn * cos(a) + Ge * sin(a))$ in Eq. (1) and the size of the mapping factors represented by actual values of the three *mfg*. We included gradient contributions corresponding to all GNSS observations in the benchmark campaign during a single day (May 31, 2013). Obviously, an actual magnitude of the gradient depends on the mapping factor. While the BS *mfg* generates higher mapping factors and thus smaller gradient contributions term (scatters in y-axis), the CH *mfg* provides lower mapping factors and thus higher gradient contribution terms. The tilting *mfg* gives then factors in between BS and CH *mfg* and results in gradient contributions in between the two. We can thus further focus on BS and CH *mfg* only as these can be considered as two extreme cases.

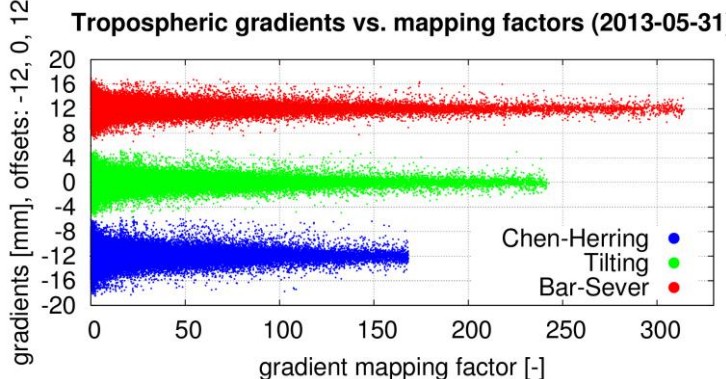

**Figure 1.** Variability of gradient mapping factors and tropospheric gradient contributions expressed in azimuths of individual satellites. Three *mfg* were studied on May 31, 2013: Chen and Herring *mfg* (blue), Bar-Sever *mfg* (red) and tilting *mfg* (green).

We use the G-Nut/Tefnut software (Václavovic et al., 2014) for GNSS data processing of the benchmark campaign. This software utilizes the PPP method and is capable of multi-GNSS processing in real-time (RT), near-real time (NRT) and post-processing (PP) mode with a focus on all the tropospheric parameters estimation: ZTDs, tropospheric gradients and slant delays (Douša et al., 2018b). Stochastic modelling of the troposphere allows an epoch-wise parameter estimation by extended Kalman filter in RT solutions (FLT) or its combination with a backward smoother which is used for NRT and post-processing solutions (FLT+SMT), see Václavovic and Douša (2015).

Table 1 describes all eight variants of solution for the benchmark campaign produced using the G-Nut/Tefnut which differ in (a) elevation cut-off angle (3° or 7°), (b) gradient mapping function (Chen and Herring = CH or Bar-Sever = BS), (c) constellations (GPS only = Gx or GPS+GLONASS = GR) and (d) processing mode (post-processing using the FLT+SMT processing or simulated real-time using the FLT processing only). Five variants based on the post-processing mode used the
backward smoother and the ESA final orbit and clock products (http://navigation-office.esa.int/GNSS_based_products.html). Three variants, abbreviated as RT1GxCH3, RT3GxCH3 and RTEGxCH3, were used to test the performance of the Kalman filter and RT orbit and clock corrections using the IGS01 (RT1GxCH3) and IGS03 (RT3GxCH3) corrections from the IGS Real-Time Service (RTS, http://rts.igs.org). The IGS01 RTS product is a GPS only single-epoch solution produced using software developed by ESA/ESOC. The IGS03 product is a GPS+GLONASS solution based on the Kalman filter and the
BKG's BNC software. The last solution, RTEGxCH3, applying the ESA final product is used to test a benefit of the backward smoothing on the one hand, and, an impact of the quality of RT corrections on the other hand. Unfortunately, the solution based on the processing of GPS+GLONASS data in the simulated RT mode had to be rejected due to a highly variable quality of RT corrections in 2013 affecting mainly the GLONASS contribution (and we noted temporal problems in GPS solutions too, see Figure 4).

The GPT model was used for calculating a priori ZHDs and the GMF was used for mapping hydrostatic and wet delays to the zenith. Estimated tropospheric parameters are thus independent from any meteorological information. GNSS observations were processed using 30-hour data batches when starting six hours before the midnight of a given day in order to eliminate the PPP convergence. In all variants, the observation sampling of 300 s was used with ZTDs and tropospheric gradients estimated for every epoch. The station coordinates were estimated on a daily basis. The random walk of 6 mm/sqrt(hour) was applied
for the ZTD and 1.5 mm/sqrt(hour) for the gradients. Absolute IGS model IGS08.ATX was used for the antenna phase centre offsets and variations. All variants used the elevation observation weighting of $1/sin^2(e)$.

**Table 1.** Processing parameters of individual variants from the G-Nut/Tefnut software. Mode FLT denotes to simulated real-time solution using Kalman filter only, FLT+SMT to post-processing solution using the Kalman filter and the backward smoother.

| Solution name | Elevation cut-off | Constellation | Gradient mapping function | Products | Mode |
|---|---|---|---|---|---|
| GxCH3 | 3 | GPS | Chen and Herring | ESA final | FLT+SMT |
| GRCH3 | 3 | GPS+GLONASS | Chen and Herring | ESA final | FLT+SMT |
| GRBS3 | 3 | GPS+GLONASS | Bar-Sever | ESA final | FLT+SMT |
| GxCH7 | 7 | GPS | Chen and Herring | ESA final | FLT+SMT |
| GRCH7 | 7 | GPS+GLONASS | Chen and Herring | ESA final | FLT+SMT |
| RT1GxCH3 | 3 | GPS | Chen and Herring | IGS01 RT | FLT |
| RT3GxCH3 | 3 | GPS | Chen and Herring | IGS03 RT | FLT |
| RTEGxCH3 | 3 | GPS | Chen and Herring | ESA final | FLT |

## 2.3 Estimation of tropospheric gradients from NWM

Tropospheric gradients and zenith delays were derived from the output of two different numerical weather models; the ERA5 (https://www.ecmwf.int/en/forecasts/datasets/archive-datasets/reanalysis-datasets/era5) and a simulation utilizing the Weather Research and Forecasting (WRF) model (Skamarock et al., 2008). The ERA5 is a reanalysis produced at the European Centre for Medium-Range Weather Forecasts (ECMWF). The pressure, temperature and specific humidity fields are provided with a horizontal resolution of approximately 31 km (T639 spectral triangular truncation) on 137 vertical model levels (up to 0.01 hPa) every hour. The WRF simulations are performed at GFZ Potsdam. The initial and boundary conditions for the limited area 24-hour free forecasts (starting every day at 0 UTC) stem from the analysis of the Global Forecast System (GFS) of the National Centers for Environmental Prediction (NCEP). The pressure, temperature and specific humidity fields are available every hour with a horizontal resolution of 10 km on 49 vertical model levels (up to 50 hPa).

The ray-trace algorithm by Zus et al. (2012) is used to compute STDs. The tropospheric gradients are derived from STDs as follows. At first, 120 STDs are computed at elevation angles 3°, 5°, 7°, 10°, 15°, 20°, 30°, 50°, 70°, 90° and all azimuths between 0° and 360° with an interval of 30°). Second, we compute azimuth-independent STDs from the local vertical refractivity profile. Third, the differences between the azimuth-dependent STDs and the azimuth-independent STDs are computed. Finally, the gradient components are determined by a least-square fitting. For details the reader is referred to the Appendix in Zus et al. (2015).

Using ten years of ERA5 global data, we tested different observation elevation weighting schemes (equal versus the elevation dependent weighting of $1/sin^2(e)$) and two *mfg*s (BS and CH) in the least squares parameter fitting. While using different observation elevation weighting schemes led to negligible differences in the tropospheric gradients, we found a significant systematic difference in the north gradient component between tropospheric gradients derived with BS and CH *mfg* (see Appendix A). In this regard it is important to note that NWM derived tropospheric gradients presented in this study were computed using CH *mfg*.

We note that tropospheric gradients can be computed with the closed form expression depending on the north-south and east-west horizontal gradient of refractivity (Davis et al., 1993). We compared the ERA5 tropospheric gradients derived with our method and the closed-form method with GNSS tropospheric gradients from the GRCH3 solution. We find that for the considered stations (over the entire benchmark period) the root-mean square deviation between NWM and GNSS tropospheric gradients is 10 % smaller if we apply our method instead of the closed-form method. This can be explained by the fact that our method is closer to the method actually applied in the GNSS analysis (parameter estimation).

We also compared our NWM tropospheric gradients with NWM tropospheric gradients provided by the TU Vienna (see Appendix B). We found a good agreement between the estimates, in particular between our tropospheric gradients and the so called refined horizontal gradients (Landskron and Boehm, 2018).

**2.4 Comparison of gradient estimates**

Absolute values of tropospheric gradient components stay typically below 1-2 mm under standard atmospheric conditions and can reach 4-6 mm during severe weather conditions. The gradient of 1 (6) mm corresponds to about 55 (330) mm slant delay correction when projected to 7° elevation angle. For an illustration an example time series of tropospheric gradients at station LDB2 (Brandenburg, Germany) for a period between May 15 and June 15, 2013 is given in Figure 2.

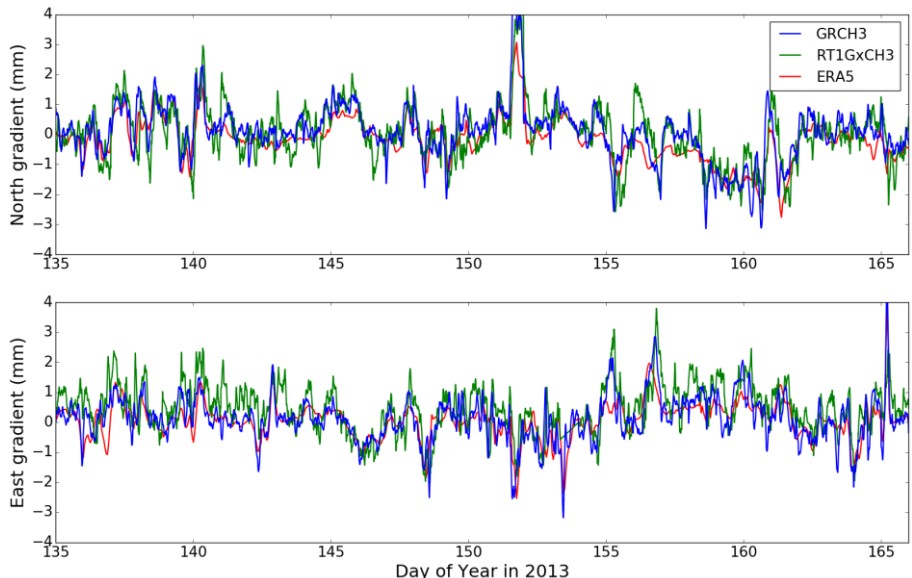

**Figure 2.** Tropospheric gradients retrieved from GNSS data processing (GRCH3, RT1GxCH3) and from NWM ERA5 at station LDB2 (52.209°N, 14.121°E, Germany) for a period from May 15, 2013 till June 15, 2013.

In the presented study, ZTDs and tropospheric gradients from all eight GNSS variants were compared to each other and also to the tropospheric parameters from ERA5 and WRF to evaluate the impact of various settings in GNSS data processing. Although about 430 GNSS stations are available in the benchmark data set, statistical results given in the Section 3 are based on a subset of 243 stations. Firstly, 84 stations without the capability of receiving GLONASS signals were excluded. Secondly, stations which did not have at least 5 % of all the observations in the range of elevation angles between 3° and 7° were excluded as well. This rule was applied to allow a systematic evaluation of elevation cut-off angle impact on tropospheric parameters. The majority of the stations (103) had to be excluded because of inability to provide a sufficient number of observations at very low elevation angles. During the statistics computation a standard data screening was applied to exclude outlier values. Moreover, epochs were RT GNSS variant of solution RT3GxCH3 provided unrealistic tropospheric gradients (see Section 3.3) were also excluded from all the statistics computation except from the coordinates repeatability evaluation.

Tropospheric parameters from the G-Nut/Tefnut software were provided every 5 minutes while the output from both NWM models was available every hour. Therefore, comparisons between GNSS solutions (Section 3.2) are based on a 5-minute interval while comparisons between GNSS and NWM solutions (Section 3.3) are based on a 1-hour interval.

**3 Impact of applied processing settings on GNSS tropospheric gradients estimation**

The section starts with an introductory evaluation of mean tropospheric gradients and formal errors of their estimates. This is followed by comparisons between individual GNSS solutions and comparisons between GNSS and NWM solutions.

**3.1 Comparison of mean tropospheric gradients and formal errors of their estimates**

Mean gradient magnitudes and azimuth angles (direction of gradient) over the whole benchmark period were computed for 243 GNSS stations and are presented in Table 2. Mean magnitudes of tropospheric gradients from all post-processing GNSS variants oscillated around 0.85 mm and 0.67 mm when using the CH *mfg* and the BS *mfg*, respectively. The latter shows about 17 % smaller gradients compared to the former if all the processing aspects remained identical. Both RT solutions also resulted with higher gradient magnitudes, namely +14 % for RT1GxCH3 and +42 % for RT3GxCH3 when compared to the

corresponding GxCH3 post-processing variant. A mean gradient magnitude of about 0.7 mm was found for both NWM solutions, i.e. of about 0.1 mm smaller than for the GRCH3 solution. This can be mainly explained by the limited horizontal resolution of the NWMs.

Table 2 shows that mean tropospheric gradients point towards the equator what is in an agreement with Meindl et al. (2004). Such a mean gradient direction does not depend on the gradient mapping function. By adding GLONASS observations the

mean gradient direction was changed by +2°, however, actual effects were found to be highly station-dependent with a typical range of ±5° for individual stations. The direction of mean gradient in both NWM solutions was in a very good agreement with all GNSS post-processing variants.

Directions of mean gradient over individual stations were mostly within ±15° when compared to the total mean gradient estimated for the stations and the solution variant. On the other hand, the performance was not identical for the individual

solutions. A change of cut-off elevation angle from 7° to 3° led to an increased number of stations with the mean gradient direction within ±15° of the total mean direction and to a decreased number of stations with a mean gradient direction differing for more than 30° (regarded as outlier stations in Table 2). Two GNSS stations were marked as outliers by all processed variants with their mean gradient direction differing by more than 50° from the total variant mean. Both of them are located in an urban area in south-west Germany and are using the same receiver and antenna type from Leica, which is however used by

many other stations in the same region where no issues with gradient mean angle were identified. Still, the reason of their different behaving can be of instrumental or environmental origin.

**Table 2.** Mean magnitudes and azimuth angles of tropospheric gradients from all individual GNSS variants of processing and NWMs ERA5 and WRF.

| Solution | Mean magnitude (mm) | Mean azimuth (°) | Percentage of stations with mean azimuth = total_mean ± 15° | Percentage of stations with mean azimuth = total_mean ± 30° | Number of outlier stations |
|---|---|---|---|---|---|
| GRCH3 | 0.82 | 170.0 | 89.7 | 99.2 | 2 |
| GRBS3 | 0.67 | 170.2 | 92.6 | 98.8 | 3 |
| GxCH3 | 0.83 | 168.2 | 88.5 | 97.5 | 6 |
| GxCH7 | 0.86 | 168.0 | 73.7 | 95.5 | 11 |
| GRCH7 | 0.84 | 170.2 | 79.0 | 97.1 | 7 |
| RT1GxCH3 | 0.95 | 151.9 | 92.6 | 98.7 | 5 |
| RT3GxCH3 | 1.18 | 162.7 | 96.3 | 98.8 | 3 |
| RTEGxCH3 | 0.75 | 168.3 | 85.6 | 97.5 | 6 |
| ERA5 | 0.68 | 169.3 | 96.3 | 100.0 | 0 |
| WRF | 0.73 | 170.9 | 100.0 | 100.0 | 0 |

Table 3 summarizes mean repeatability of daily coordinates as well as statistical comparison of formal errors of estimated ZTDs and tropospheric gradients from different GNSS processing variants. The station coordinates repeatability is improved when using combined GPS+GLONASS solutions compared to GPS-only solutions, namely by a factor of 2 and 1.2 in horizontal components and the height, respectively. The number of available satellites and their geometry plays a significant role in this context. An increase of the elevation angle cut-off (from 3° to 7°) resulted in improved height repeatability, which is consistent with the results of Zhou et al. (2017) suggesting optimal 7° cut-off for the height repeatability when comparing results of different elevation angle cut-off (3° - 15°). However, it should be noted that GPT+GMF models and the PPP method were used in both cases. Contrary, Douša et al. (2017) observed an improvement in the height repeatability even when using the elevation angle cut-off 3° (compared to 7° and 10°) when exploiting double-difference observations, the VMF1 mapping function (Boehm et al., 2006b) and the Bernese GNSS Software (Dach et al. 2015). Douša et al. (2017) indicated also worse results when using GPT+GMF compared to VMF1, which can be attributed to modelling errors in the former, particularly if applied in PPP (Kouba, 2009). We also notice a slightly better performance in case of the BS *mfg* when compared to the CH *mfg* while this difference was found to be statistically significant in the North and Up component by the Wilcoxon signed-rank test at the 5% significance level. The results of the forward filter processing didn't show any degradation when using the ESA final products (RTEGxCH3). When using the IGS real-time product, the repeatability of all coordinates got worse by a factor of 2-3 and 4-5 for RT1GxCH3 and RT3GxCH3 variant respectively. The latter is attributed to a lower quality of the IGS03 RT product during some periods, see Figure 4.

Formal error of the parameter can be generally regarded as an estimation uncertainty. Formal errors increase when the number of observations and/or the geometry decrease. Naturally, smaller formal errors correspond to the lower elevation angle cut-off, which can be observed for both ZTDs and tropospheric gradients in Table 3. Formal errors are about 17% and 11% smaller when using the 3° cut-off (GRCH3) compared to the 7° cut-off (GRCH7) for horizontal gradients and ZTDs, respectively, thus

indicating a higher impact on the former. A decrease of formal errors of tropospheric gradients estimated with a 3° cut-off compared to 10° cut-off was previously reported also by Meindl et al. (2004). Interestingly, using the BS *mfg* resulted in smaller formal errors of tropospheric gradients, but we haven't observed any change in formal errors of other estimated parameters. The smaller formal errors may suggest an improvement in estimated parameters using BS *mfg*, as also found from the coordinates repeatability.

**Table 3.** Mean position repeatability and formal errors and their standard deviation for tropospheric parameters from individual GNSS processing variants.

| GNSS solution | Position repeatability | | | ZTD formal error | | N gradient formal error | | E gradient formal error | |
|---|---|---|---|---|---|---|---|---|---|
| | North (mm) | East (mm) | Height (mm) | Mean (mm) | SD (mm) | Mean (mm) | SD (mm) | Mean (mm) | SD (mm) |
| GRCH3 | 1.71 | 4.13 | 5.60 | 3.80 | 0.37 | 0.81 | 0.10 | 0.81 | 0.09 |
| GRBS3 | 1.69 | 4.13 | 5.53 | 3.82 | 0.37 | 0.74 | 0.09 | 0.75 | 0.09 |
| GxCH3 | 3.62 | 8.68 | 5.91 | 4.28 | 0.46 | 0.93 | 0.13 | 0.90 | 0.13 |
| GxCH7 | 3.46 | 9.26 | 5.43 | 4.84 | 0.44 | 1.14 | 0.14 | 1.05 | 0.14 |
| GRCH7 | 1.71 | 4.09 | 4.96 | 4.28 | 0.36 | 0.99 | 0.10 | 0.95 | 0.11 |
| RT1GxCH3 | 3.97 | 10.71 | 7.57 | 6.66 | 0.70 | 0.91 | 0.08 | 0.92 | 0.09 |
| RT3GxCH3 | 9.13 | 19.69 | 8.51 | 7.05 | 0.80 | 1.49 | 0.22 | 1.53 | 0.22 |
| RTEGxCH3 | 1.68 | 3.91 | 5.74 | 6.60 | 0.68 | 0.90 | 0.08 | 0.91 | 0.08 |

## 3.2 Comparison of individual GNSS variants with each other

Results for individual GNSS variants comparison based directly on ~3.4 million of pairs of values over 55 days and 243 GNSS stations are presented in Table 4. We notice a good agreement among all the post-processing variants (top part of Table 4). The mean differences stayed below 0.2 mm for ZTD and ±0.02 mm for tropospheric gradients with one exception for the latter parameter. This was a comparison between results provided by CH and BS *mfg*s where the mean differences reached -0.05 mm and 0.03 mm for north and east gradient component, respectively. These small systematic effects can be attributed to the average difference between tropospheric gradients computed with BS *mfg* compared to CH *mfg*. The standard deviation (SD) indicates the smallest impact due to the change of *mfg* for both ZTD estimates (0.2 mm) and tropospheric gradients (~0.14 mm). The impact increases then for both ZTD and gradients when comparing results of single and dual-constellation (1.2 mm for ZTD, ~0.17 mm for gradients). It should be noted that GLONASS observations were down-weighted by a factor of 1.5 in dual-constellation variants of solution to reflect both a lower quality of precise products and observations. The gradients estimated with improved geometry and using more observations are expected to provide more accurate and reliable estimates. It is notable in the comparisons of single-/dual-constellation at different elevation cut-off angles (the impact is larger for a higher cut-off). The largest impact is eventually observed due to the elevation cut-off angle, i.e. 2.2 mm and ~0.20 mm for ZTD and tropospheric gradients, respectively. Linear correlation coefficients (CorCoef) reach value of ~1 in all cases for the ZTD comparisons. The ZTDs were thus practically unaffected by different gradient models. For the gradient comparisons, the correlation coefficients are progressively decreasing from 0.99 to 0.95 while values of SD are increasing.

An increased scatter of RT processing is visible on significant mean differences and on the standard deviation values of ZTD and tropospheric gradients increased by a factor of 3. These are also emphasised by the reduction of correlation coefficients mainly for tropospheric gradients. The two RT solutions can be still considered of good quality if we take into consideration results found in Ahmed et al. (2016) or Kačmařík (2018), where mean biases and SD values up to 12 mm were reported for

comparisons between RT ZTD solutions based on IGS01 and IGS03 streams and post-processing solutions based on final products. Since virtually zero mean differences for both ZTD and tropospheric gradients are found in the RTEGxCH3 variant, when using the Kalman filter too, the degraded quality of RT tropospheric parameters is mainly a consequence of the poorer quality of IGS01 and IGS03 RT products (Douša et al., 2018b).

The differences of ZTDs and tropospheric gradients from all compared variants of solution were also statistically tested. And

in all cases, the differences were found to be statistically significant at the 5% significance level while using the Wilcoxon signed-rank test (https://docs.scipy.org/doc/scipy/reference/generated/scipy.stats.wilcoxon.html). This non-parametric test was used since none of the processed variant of solution evinced a normal distribution of their ZTDs and tropospheric gradients.

**Table 4.** Comparison of individual variants of GNSS data processing run in post-processing mode (top) and in simulated real-time mode
(bottom), units: Mean and SD in mm, CorCoef represents a linear correlation coefficient.

| Compared post-processing solutions | ZTD | | | N-S gradient | | | E-W gradient | | |
|---|---|---|---|---|---|---|---|---|---|
| | Mean | SD | CorCoef | Mean | SD | CorCoef | Mean | SD | CorCoef |
| GRCH3 – GRBS3 | 0.0 | 0.2 | 1.000 | -0.05 | 0.14 | 0.995 | 0.03 | 0.13 | 0.996 |
| GRCH3 – GxCH3 | 0.1 | 1.1 | 1.000 | 0.00 | 0.16 | 0.973 | -0.02 | 0.15 | 0.976 |
| GRCH7 – GxCH7 | 0.1 | 1.2 | 1.000 | -0.01 | 0.19 | 0.963 | -0.02 | 0.17 | 0.968 |
| GRCH3 – GRCH7 | 0.1 | 2.1 | 1.000 | 0.01 | 0.20 | 0.961 | 0.00 | 0.18 | 0.966 |
| GxCH3 – GxCH7 | 0.2 | 2.2 | 1.000 | 0.01 | 0.23 | 0.949 | -0.01 | 0.20 | 0.957 |
| | | | | | | | | | |
| Compared RT solutions | ZTD | | | N-S gradient | | | E-W gradient | | |
| | Mean | SD | CorCoef | Mean | SD | CorCoef | Mean | SD | CorCoef |
| RT1GxCH3- GxCH3 | 3.4 | 5.7 | 0.996 | -0.10 | 0.54 | 0.716 | 0.18 | 0.55 | 0.669 |
| RT3GxCH3 - GxCH3 | 2.7 | 6.2 | 0.996 | -0.05 | 0.66 | 0.699 | 0.09 | 0.68 | 0.651 |
| RTEGxCH3 - GxCH3 | 0.1 | 4.4 | 0.998 | -0.00 | 0.39 | 0.833 | -0.01 | 0.43 | 0.776 |
| RT1GxCH3 – RT3GxCH3 | 0.8 | 5.0 | 0.997 | -0.03 | 0.65 | 0.718 | 0.09 | 0.63 | 0.712 |

## 3.2 Comparison of individual GNSS variants with NWM

The statistics for the GNSS and NWM comparisons are summarized in Table 5. For ZTDs a mean difference of about 1 (4) mm is visible between GNSS and ERA5 with standard deviations around 9 (10) mm and correlation coefficients around 0.99

(0.99) for individual post-processing (RT) GNSS solutions. The negative mean difference of -3 mm in ZTD between GNSS and WRF might be due to the global NCEP GFS analysis which is used for the initial and boundary conditions for the WRF solution. A negative mean difference of -5 mm in ZTD between two GNSS reference solutions and a solution based on the NCEP GFS was already reported in the past (Douša et al., 2016). The standard deviations of differences are about 2 mm larger

when GNSS and WRF are compared. This is probably due to the fact that the solution from WRF is based on a 24-hour forecast (errors are supposed to grow with increasing forecast length) whereas the solution from ERA5 is based on a reanalysis.

**Table 5.** Comparison of individual variants of GNSS data processing run in post-processing mode (top) and in simulated real-time mode (bottom) with NWM solutions, units: Mean and SD in mm, CorCoef represents a linear correlation coefficient.

| Compared post-processing solutions | ZTD | | | N-S gradient | | | E-W gradient | | |
|---|---|---|---|---|---|---|---|---|---|
| | Mean | SD | CorCoef | Mean | SD | CorCoef | Mean | SD | CorCoef |
| GRCH3 – ERA5 | 1.0 | 8.8 | 0.992 | -0.02 | 0.46 | 0.743 | -0.01 | 0.46 | 0.744 |
| GRBS3 – ERA5 | 1.0 | 8.9 | 0.992 | 0.03 | 0.41 | 0.730 | -0.03 | 0.42 | 0.729 |
| GxCH3 – ERA5 | 1.0 | 9.0 | 0.991 | -0.01 | 0.47 | 0.727 | 0.01 | 0.46 | 0.737 |
| GxCH7 – ERA5 | 0.7 | 10.0 | 0.989 | -0.02 | 0.54 | 0.653 | 0.02 | 0.51 | 0.685 |
| GRCH7 – ERA5 | 0.8 | 9.7 | 0.990 | -0.02 | 0.51 | 0.680 | -0.00 | 0.50 | 0.699 |
| GRCH3 – WRF | -2.8 | 11.1 | 0.987 | -0.04 | 0.51 | 0.688 | 0.00 | 0.52 | 0.681 |
| GRBS3 – WRF | -2.7 | 11.2 | 0.987 | 0.01 | 0.47 | 0.675 | -0.02 | 0.49 | 0.664 |
| GxCH3 – WRF | -2.8 | 11.3 | 0.987 | -0.04 | 0.52 | 0.673 | 0.02 | 0.53 | 0.675 |
| GxCH7 – WRF | -3.1 | 11.9 | 0.985 | -0.04 | 0.58 | 0.611 | 0.03 | 0.56 | 0.632 |
| GRCH7 – WRF | -2.9 | 11.7 | 0.985 | -0.05 | 0.56 | 0.633 | 0.01 | 0.55 | 0.644 |
| | | | | | | | | | |
| Compared RT solutions | ZTD | | | N-S gradient | | | E-W gradient | | |
| | Mean | SD | CorCoef | Mean | SD | CorCoef | Mean | SD | CorCoef |
| RT1GxCH3 – ERA5 | 4.4 | 10.1 | 0.989 | -0.12 | 0.55 | 0.650 | 0.20 | 0.56 | 0.621 |
| RT3GxCH3 – ERA5 | 3.4 | 10.3 | 0.989 | -0.05 | 0.71 | 0.573 | 0.11 | 0.72 | 0.573 |
| RTEGxCH3 – ERA5 | 1.0 | 9.6 | 0.990 | -0.01 | 0.46 | 0.713 | -0.00 | 0.45 | 0.714 |
| RT1GxCH3 – WRF | 0.5 | 12.1 | 0.984 | -0.14 | 0.59 | 0.610 | 0.20 | 0.61 | 0.560 |
| RT3GxCH3 – WRF | -0.4 | 12.2 | 0.984 | -0.07 | 0.74 | 0.537 | 0.12 | 0.76 | 0.523 |
| RTEGxCH3 – WRF | -2.7 | 11.6 | 0.986 | -0.04 | 0.50 | 0.668 | 0.01 | 0.51 | 0.647 |
| | | | | | | | | | |
| ERA5 – WRF | -3.9 | 11.1 | 0.987 | -0.02 | 0.40 | 0.771 | 0.01 | 0.44 | 0.722 |

With regards to the tropospheric gradients, the mean differences between post-processed GNSS and NWM stayed within a range from -0.05 to 0.03 mm. The existing differences between two GNSS variants of solution based on different *mfg*s can be attributed to usage of CH *mfg* for derivation of NWM tropospheric gradients and to the existing systematic difference between tropospheric gradients estimated using these two *mfg*s (see Section 2.2 and Appendix A). The standard deviations between GNSS and NWM were approximately doubled or tripled when compared to standard deviations between individual variants of GNSS solutions (Table 4). They were also found to be higher for the WRF than for ERA5. Again, this can be probably explained by the fact that the solution from WRF is based on a 24-hour free forecast whereas ERA5 is based on a reanalysis. Both NWMs lead to consistent results: standard deviations are smaller and correlation coefficients higher for GNSS solutions using a lower cut-off elevation angle (3° instead of 7°) and/or more observations (GPS+GLONASS). For example, the SD for north gradient component between GNSS and ERA5 is 0.54 mm for the GxCH7 variant while 0.46 mm for the GRCH3 variant. This represents a decrease of 15 %. In this regards we also derived tropospheric parameters from both NWMs using a 7° cut-off elevation angle and repeated the comparisons to test if GNSS variants of solution with a 7° cut-off would be closer to

NWM solutions based also on the 7° cut-off angle. And we always found a better agreement between any evaluated GNSS variant of solution and the NWM solution based on the 3° cut-off angle – in terms of mean difference, standard deviation and correlation coefficient. From two GNSS variants differing only in the *mfg*, the solution applying the BS mapping function is closer to the NWMs in terms of standard deviation. Since the CH *mfg* was used to derive tropospheric gradients from NWMs,

the opposite situation could be expected, and we generally note that presented results of comparisons between tropospheric gradients from the GNSS GRBS3 solution and NWMs should be taken only as informative. The lower values of standard deviation can be partly understood as the magnitudes computed as $\sqrt{Gn^2 + Ge^2}$ of GNSS tropospheric gradients using the BS *mfg* are smaller compared to the CH *mfg* (see Section 2.2) and the magnitudes of NWM tropospheric gradients are more smoothed compared to the GNSS tropospheric gradients.

In order to evaluate the statistical significance of differences of ZTDs and tropospheric gradients from all variants of GNSS solution and both NWMs we applied again the Wilcoxon signed-rank test. Again, the differences were found to be statistically significant at the 5% significance level in all cases.

Maps showing tropospheric gradients were generated for all the variants of GNSS solutions and both NWM solutions and visually evaluated for the whole benchmark period. For better visualization we included all the GNSS stations of the benchmark

campaign, i.e. not just the subset of 243 stations used for the presented statistics. Generally, GNSS provided homogenous fields of tropospheric gradients without a noisy behaviour at the level of individual stations and a very good agreement in gradient directions and usually also in gradient magnitudes was found between GNSS and NWM gradient maps. In Figure 3, two examples are shown for different events when weather fronts were passing over the studied area. For a description of meteorological conditions prevailing during these events the reader is referred to Douša et al. (2016). Tropospheric gradients

derived from NWM provided more smoothed gradient fields, but somehow limited to render local structures mainly due to the spatial resolution of both NWMs. As the ERA5 model has coarser spatial resolution than the WRF model, such behaviour was a little bit more apparent in its results. On the other hand, when compared to results of the 1° × 1° resolution global models ERA-Interim and NCEP GFS (Douša et al., 2016), the presented NWMs tropospheric gradients have larger magnitudes.

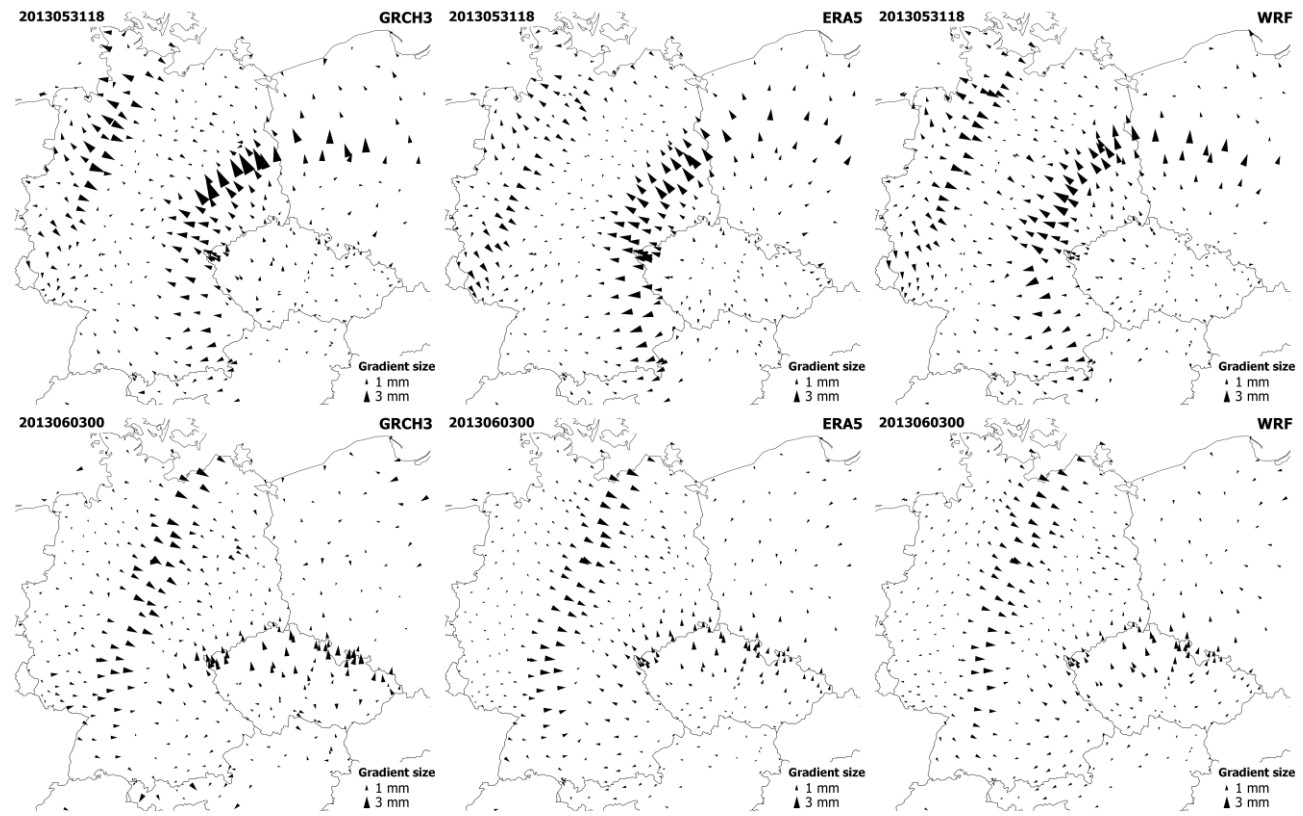

**Figure 3.** Tropospheric gradient maps from GNSS GRCH3 solution (left), NWM ERA5 solution (middle) and NWM WRF solution (right) on 31 May 2013, 18:00 UTC (top) and on 03 June 2013 00:00, UTC (bottom).

Comparing GNSS to NWM products in Table 5 indicated that the RTEGxCH3 solution driven by the Kalman filter and the ESA final product shows a comparable performance to the GxCH3 solution driven by the Kalman filter and the backward smoother. An increase of mean difference and standard deviation values for other solutions based on RT mode indicates that the quality of the RT tropospheric solution is dominated by an actual quality of RT orbit and clock corrections. In this regard, we examined systematically all tropospheric gradient maps and found that gradients from the RTEGxCH3 solution are always in a very good agreement with post-processing solutions. Although there were imperfections in matching RT1GxCH3 gradients and post-processing solutions, the performance can be still considered as generally good and stable. This was however not the case of the RT3GxCH3 solution where we observed a varying quality of estimated tropospheric gradients. For the majority of epochs, in particular during the periods with strong gradients, the tropospheric gradients could be evaluated as acceptable. However, situations when gradients from all the stations point to the same direction occurred from time to time, obviously without a physical relation to the actual weather situation. An example of this behaviour is presented in Figure 4 where tropospheric gradients from the RT3GxCH3 solution behave normally on 31 May 2013, 18:00 UTC, and became unrealistic on 6 May 2013, 18:00 UTC where all the stations point to the south-west direction and reveal high gradient magnitudes. Such issues occurred occasionally for a limited period of time in the RT3GxCH3 solution only. The reason is an instability of the

RT3 stream during the initial period (the first half of 2013) affected by many interruptions and data gaps thus caused frequent parameter re-initialization in PPP.

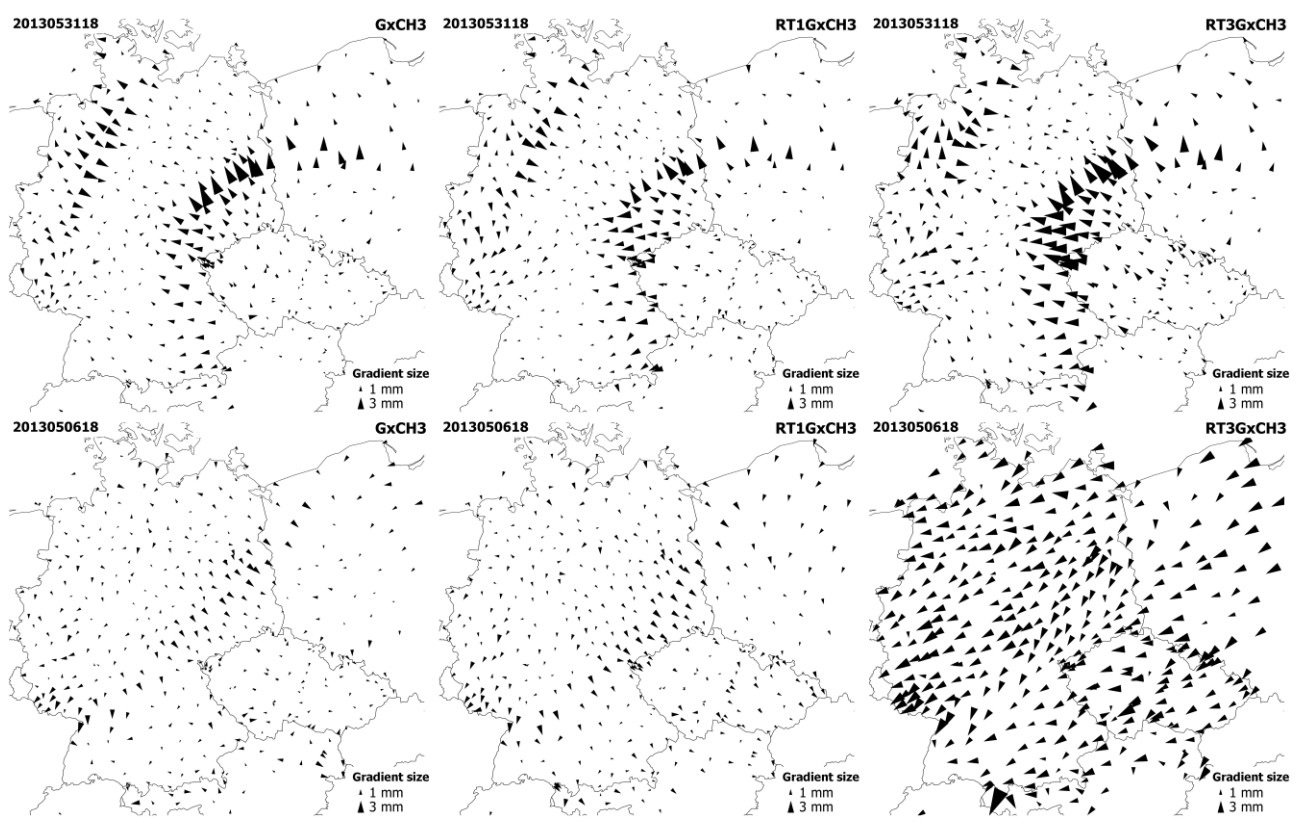

**Figure 4.** Tropospheric gradient maps from GNSS GxCH3 solution (left), GNSS RT1GxCH3 solution (middle) and GNSS RT3GxCH3 solution (right) on 31 May 2013, 18:00 UTC (top) and on 06 May 2013, 18:00 UTC (bottom).

## 4 Systematic effects induced by different gradient mapping functions and elevation-dependent weighting

In this section, we focus on studying systematic differences induced purely by different *mfg* and observation elevation-dependent weighting (OEW) during eight days from May 25 to June 1, 2013. For two solutions defined in Section 2.2 and utilizing CH *mfg* (GRCH3) and BS *mfg* (GRBS3), we additionally generated four variants using various OEW schemes: 1) EQUAL, equal weighting, 2) SINEL1, $1/sin(e)$ , 3) SINEL2, $1/sin^2(e)$, and 4) SINEL4, $1/sin^4(e)$. Generally, in the SINEL OEW schemes, the contribution of low-elevation observations to all estimated parameters decreases with increasing power *y* in $1/sin^y(e)$.

Figure 5 displays example distributions of carrier-phase post-fit residuals with respect to the elevation for the SINEL2 observation weighting (left panel), and without any weighting, i.e. EQUAL (right panel). While the residuals from the former are affected by the *mfg* only below 15° elevation, the residuals in the latter are affected at any elevation angles even close to

the zenith direction. Above the 30° elevation, the residuals distribution is more smoothed for the SINEL2 compared to the EQUAL. It is closer to the expected behaviour when considered errors in GNSS observations and models, including contributions from the atmosphere, multipath, uncertainty of receiver antenna phase centre variations, lower signal-to-noise ratio or cycle slips. All these errors generally increase with a decrease of observation elevation angle and, accordingly, minimum errors are thus expected in the zenith direction. The plots are also in agreement with our previous findings when studying extensively the distribution of carrier-phase post-fit residuals (not shown here for brevity sake). Using a weak or none elevation dependent weighting, the hydrostatic/wet delay mapping separation errors can introduce significant errors in both ZTD and height coordinate component (Kouba, 2009). Thought we generally recommend the use of SINEL2 elevation weighting, we show below also impact of other weighting schemes on estimated gradients.

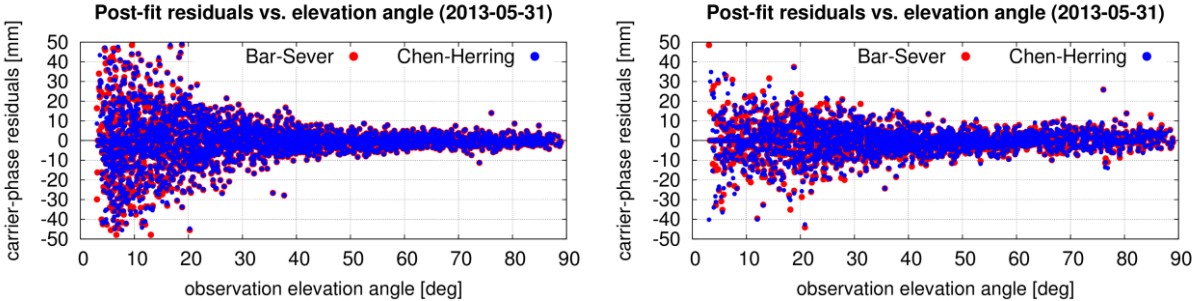

**Figure 5.** Post-fit phase residuals distribution when using different gradient mapping functions, Bar-Sever (red) and Chen and Herring (blue), and observation weighting: SINEL2 (left) and EQUAL (right).

Figure 6 displays maps of situation with large tropospheric gradients observed on May 31, 2013 at 18:00 UTC when using GRCH3 (left panels) and GRBS3 (right panels) solutions and applying the SINEL2 OEW scheme. The day is interesting due to a presence of occlusion front over Germany clearly captured by strong tropospheric gradients achievable from both GNSS and NWM analyses. The impact of *mfg* on estimated gradients shows systematic changes in gradient magnitudes – the gradients estimated with CH *mfg* (left panels) are always larger than with BS *mfg* (right panels) independently of the OEW scheme (not showed). It should be also noticed here, that the magnitudes of gradients estimated using the SINEL4 scheme were significantly reduced compared to any other OEW scheme.

Figure 7 shows mean differences, calculated over all epochs in May 31, 2013, in north (left panels) and east (right panels) gradient components between the two *mfg* (BS minus CH) when using the SINEL2 scheme. Although the magnitudes of CH gradients are always larger compared to BS gradients, the sign of the component differences depends on the gradient direction (north/south for *Gn* and east/west for *Ge*). Positive differences in north and east component appear when the estimated gradients point to south and west, respectively, and negative differences occur when the gradients point to opposite directions.

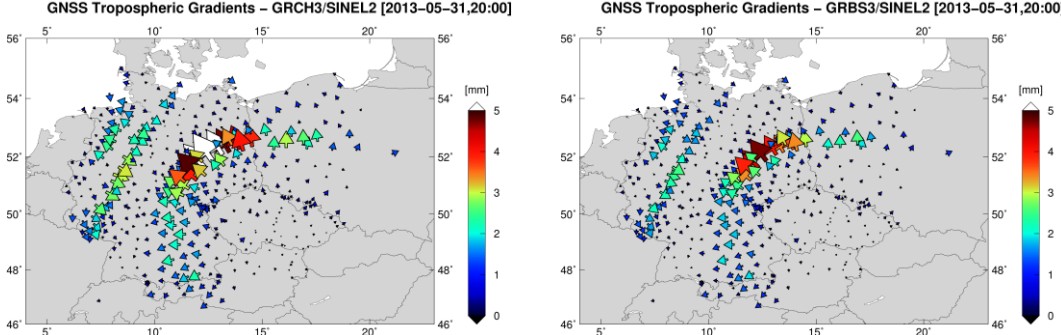

**Figure 6.** Tropospheric gradient maps on May 31, 2013 (18:00 UTC) from GNSS solutions using the SINEL2 observation weighting scheme: Chen and Herring *mfg* (left panels), Bar-Sever *mfg* (right panels).

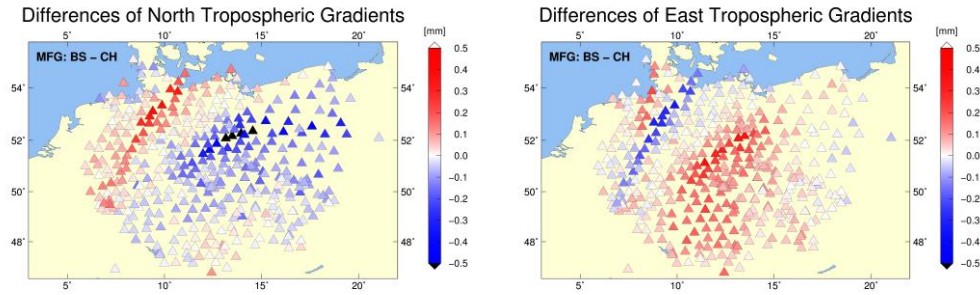

**Figure 7.** Mean differences (calculated over full day May 31, 2013) of tropospheric gradient north component (left panels) and east component (right panels) due to different *mfg*: Chen and Herring (CH), Bar-Sever (BS) when using the SINEL2 observation weighting schemes.

Figure 8 shows scatter plots of tropospheric gradient differences of all the stations in the network when using different *mfg* and OEW schemes on May 31, 2013. Obviously, the impact of the *mfg* on estimated gradients is significantly reduced for SINEL4 (well below 0.2 mm), while it is higher for all other schemes. This corresponds to the fact that large gradients are related to a horizontal anisotropy of the troposphere affecting more significantly low-elevation observations. The strongest effect can be observed for the EQUAL scheme reaching systematic differences of 1.0 mm or even higher. Such systematic differences reached twofold values of the SD obtained from comparisons of gradients using independent sources such as GNSS and NWM, see Section 3.3 or Douša et al. (2017).

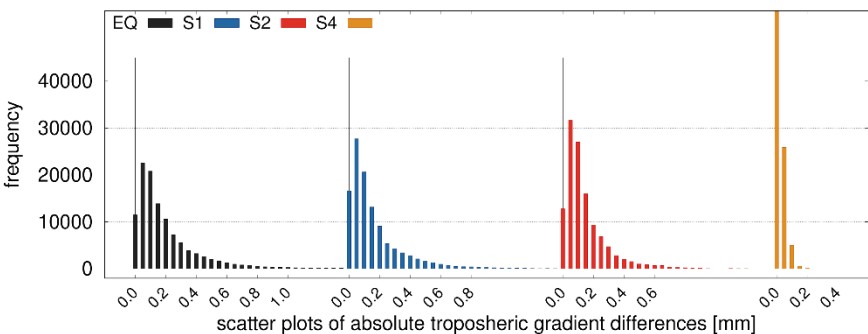

**Figure 8.** Differences of tropospheric gradients between Chen and Herring and Bar-Sever *mfg* for four observation weighting schemes: EQUAL (EQ), SINEL (S1), SINEL2 (S2), and SINEL4 (S4).

Figure 9 compares magnitudes of estimated gradients (east component only) and corresponding scatter plots of total gradient differences over all stations in the network on eight consecutive days (May 25 – June 1, 2013) when using CH and BS *mfg* and the SINEL2 OEW scheme. We can notice the days with a stronger tropospheric anisotropy (May 27-28, May 31, June 1) identifiable by a presence of gradients larger than 1.0 mm. The scatter plots systematically deviate from the zero on some days, prevailing negative and positive east components indicate that gradients in the network point westwards and eastwards, respectively. Differences in gradient magnitudes are then showed in the bottom panel. The impact due to utilizing different *mfg* clearly corresponds to the original gradient magnitudes. Both are high during the days with a strong tropospheric anisotropy, while differences due to the *mfg* choice demonstrate systematic effects up to 1 mm or more in such extreme cases.

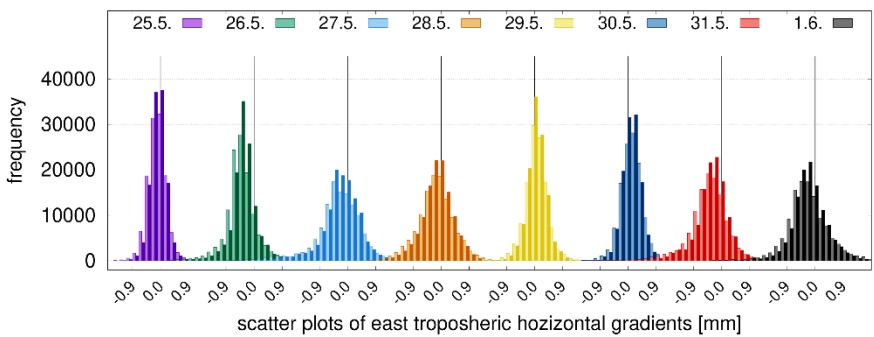

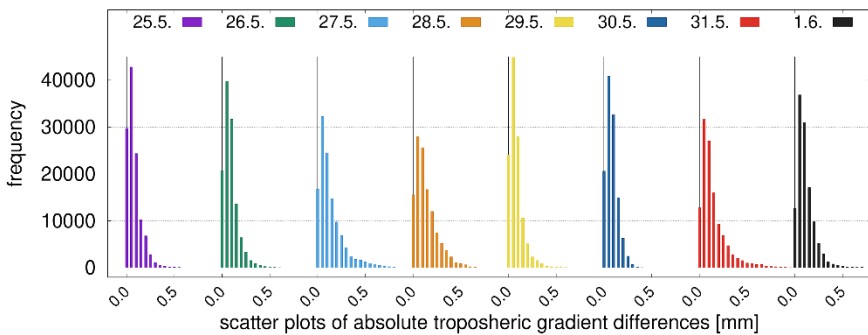

**Figure 9**. East tropospheric horizontal gradients (top) estimated using Chen and Herring (light columns) and Bar-Sever (dark columns) *mfg* and the differences (bottom) of gradients magnitudes between them. The SINEL2 OEW scheme was applied over eight days in May/June 2013.

## 5. Conclusions

We presented an impact assessment of selected GNSS processing settings on estimated tropospheric gradients together with an evaluation of differences resulting from gradient mapping function and observation elevation weighting. We exploited the GNSS4SWEC benchmark campaign covering May and June in 2013 with prevailing wet weather. Although the time period covered some severe weather events, it also contained a lot of days with standard weather conditions with tropospheric gradients close to zero. Presented results could be therefore considered representative for European conditions during the warmer part of the year.

ZTD values and tropospheric gradients were estimated in eight variants of GNSS data processing and derived from two NWMs (a global reanalysis and a limited area short range forecast). All solutions gave tropospheric parameters in high temporal resolution (5 minutes). Since no meteorological data providing any information about prevailing atmospheric conditions during the evaluated time period entered the GNSS data processing, estimated tropospheric gradients can be regarded as fully

independent, and therefore can provide additional interesting information, along with the ZTD, in support of NWMs (see Douša et al., 2016, Guerova et al., 2016).

When lowering elevation angle cut-off (from 7° to 3°), more accurate tropospheric gradient estimates were obtained. The standard deviation of differences of GNSS gradients to NWM gradients were reduced by 10%, formal errors of tropospheric

gradients were reduced, and station-wise mean gradient directions were also more stable. On the other hand, the usage of lower cut-off angle led to a slightly worse station height repeatability (10 %), which is partly in contradiction with the results of Douša et al. (2017), but in agreement with Zhou et al. (2017). The discrepancy is attributed to the use of PPP method with simplified modelling (GPT+GMF) for low-elevation observations. The 3° elevation angle cut-off can be nevertheless recommended for an optimal gradient estimation from GNSS data.

A small decrease of standard deviation of estimated gradients (2 %) was observed when using GPS+GLONASS instead of GPS only and compared to NWM gradients. This indicates that the post-processing tropospheric gradients can be reliably estimated solely with GPS constellation. However, it may still depend on applied software, strategy, products and processing, e.g. (near) real-time. In this regard, Li et al. (2015) and Lu et al. (2016) demonstrated that tropospheric gradients from multi-GNSS PPP processing improved their agreement with those estimated from NWM and WVR when compared to standalone

GPS processing.

Using a simulated real-time processing mode, the agreement of GNSS versus NWM tropospheric gradients revealed an increase in standard deviation of about 19 % (53 %) for IGS01 (IGS03) RT products when compared to the corresponding GNSS post-processing gradients. We also show that the quality of real-time tropospheric parameters is dominated by the quality of real-time orbit and clock corrections, and to a much lesser extent by the processing mode, i.e. Kalman filter without

backward smoothing. Tropospheric gradients from the RT solution using the IGS03 RT product showed occasionally a large misbehaving of tropospheric gradients at all GNSS stations obviously not related to weather conditions. This was caused by frequent PPP re-initializations due to interruptions and worse quality of the IGS03 RT product, while normal results were achieved by using the IGS01 RT product. Thus, providing high-resolution gradients in (near) real-time solution still remains challenging, which would require optimally a multi-GNSS constellation and high-accuracy RT products.

We studied systematic differences in estimated tropospheric gradients. Unlike for ZTDs, average systematic differences up to 0.5 mm over a day, and up to 1.0 mm or even more for individual gradient components during extreme cases, can affect the magnitude of estimated tropospheric gradients solely due to utilizing different gradient mapping functions or observation elevation-dependent weightings. While the *mfg* choice affects the magnitude of estimated gradient, it does not affect the direction of the gradient. However, any difference in the magnitude causes systematic errors in gradient components which

depend on the gradient direction too.  At global scale, the long-term mean gradient pointing to the equator causes systematic differences up to 0.3 mm in the north gradient component between Bar-Sever and Chen and Herring *mfg* (see Appendix A). Both smaller gradient formal errors and slightly improved height repeatability which was found to be statistically significant suggest more accurate modelling when using the Bar-Sever *mfg*. Without an accurate and independent gradient product, it is still difficult to make a strong recommendation among different *mfg*s, i.e. resulting in different absolute gradient values. In

any case, we could strongly recommend to use the same *mfg* implemented in the same form whenever comparing or combining tropospheric gradients derived from different sources (GNSS, WVR or NWM). On the other hand, if tropospheric gradients are used solely for reconstructing slant total delays, different *mfg*s should provide very similar results.

**Appendix A**

In the upper panel of Figure 10 the systematic difference in the derived tropospheric gradients based on ERA5 data (average over 10 years) is shown for any point on Earth's surface between tropospheric gradients estimated utilizing the BS *mfg* and tropospheric gradients estimated utilizing the CH *mfg*. Whereas there is no considerable systematic difference in the east gradient component, it reaches up to 0.3 mm in the north gradient component (positive in the northern and negative in the southern hemisphere). We note that the mean tropospheric gradients point to the equator, i.e., the north gradient component is

negative in the northern hemisphere and positive in the southern hemisphere. This can be seen in the lower panel of Figure 10, showing the mean north- and east gradient component utilizing the CH *mfg*, and can be explained by the fact that the mean zenith delays increase towards the equator. The systematic difference between these two *mfg*s is due to the fact that for the same slant total delays the magnitude of tropospheric gradients which are estimated utilizing a smaller mfg are larger than the magnitude of tropospheric gradients which are estimated utilizing a larger *mfg*. The product of the *mfg* and the tropospheric

gradients, i.e., the azimuth dependent part of the tropospheric delay, remains approximately the same.

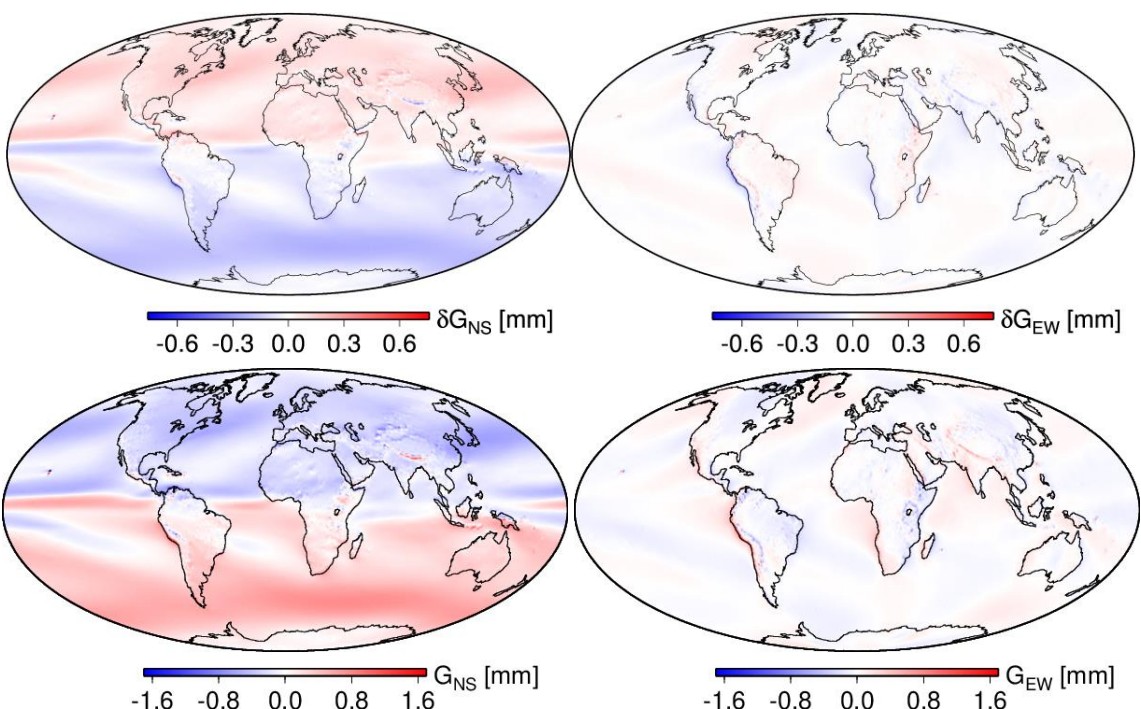

**Figure 10.** Upper panel: Systematic difference (average over 10 years) for any point on Earth's surface between tropospheric gradients estimated utilizing the gradient mapping function of Bar-Sever and tropospheric gradients estimated utilizing the gradient mapping function of Chen and Herring. Lower panel: Mean north- and east gradient component (average over 10 years) for any point on Earth's surface utilizing the mapping function of Chen and Herring. Left panels show the north gradient component, right panels the east gradient component. The results are based on ERA5 data.

## Appendix B

NWM tropospheric gradients presented in this paper were also compared with NWM tropospheric gradients provided by TU Vienna (see http://vmf.geo.tuwien.ac.at/). Specifically, we compared the NWM tropospheric gradients based on ERA5 with the so-called Linearized Horizontal Gradients (LHG) (Boehm et al. 2007b). We note that the LHGs are based on the closed form expression depending on the north-south and east-west horizontal gradient of refractivity (Davis et al., 1993). The LHGs are solely available for several stations, their provision ended in 2017 and they are no longer supported. Recently, Landskron and Boehm (2018) provided refined horizontal gradients based on a least square adjustment which are currently recommended to be used. We decided to look at three stations available in all data sets: ONSA, POTS and WTZR and we provide the comparisons in Figure 11. As to expect, we find a better agreement between ERA5 tropospheric gradients and the refined horizontal gradients. We also find that the magnitude of the ERA5 tropospheric gradients is larger than the magnitude of the refined horizontal gradients. This is not surprising since the NWM that is used in the generation of the refined horizontal gradients has a horizontal resolution of 1° only (ERA-Interim provided by the ECMWF). For example, Zus et al. (2016)

showed how an increased horizontal resolution of the NWM amplifies the tropospheric gradient components under severe weather conditions.

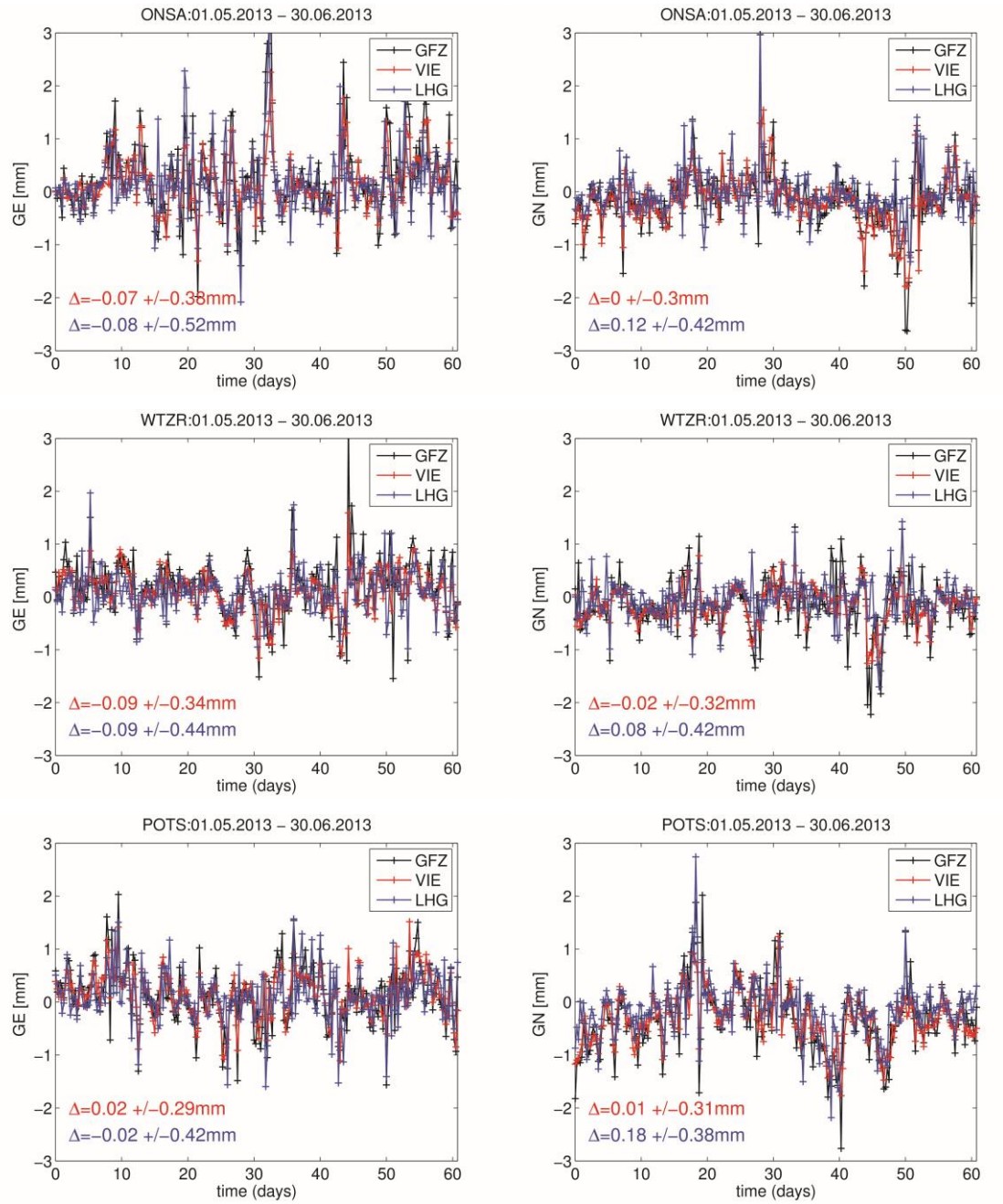

**Figure 11:** The left panels show the time series (May 1 – June 30, 2013) of the east-gradient component for the station ONSA, WTZR and POTS respectively. The right panels show the time series of the north-gradient component for the same stations. The black line corresponds to the ERA5 tropospheric gradients (GFZ, regarded in the paper as NWM ERA5), the red line corresponds to the refined horizontal gradients provided by TU Vienna (VIE) and the blue line corresponds to the so-called linearized horizontal gradients provided by TU Vienna (LHG).

The red numbers represent the mean and standard deviation between VIE and GFZ. The blue numbers are the mean and standard deviation between LHG and GFZ.

**Acknowledgement**

The authors thank all the institutions which provided GNSS observations for the COST ES1206 Benchmark campaign (Douša et al., 2016). F.Z. wants to thank Dr. Thomas Schwitalla (Institute of Physics and Meteorology, University Hohenheim) for the introduction to the WRF system. The ECMWF is acknowledged for making publicly available ERA5 reanalysis fields that were generated using Copernicus Climate Change Service Information 2018 (https://www.ecmwf.int/en/forecasts/datasets/archive-datasets/reanalysis-datasets/era5). The GFS analysis fields are provided by the National Centers for Environmental Prediction (http://www.ftp.ncep.noaa.gov/data/nccf/com/gfs/prod). The study was realized during a mobility of M.K. at GFZ Potsdam funded by the EU ESIF project No. CZ.02.2.69/0.0/0.0/16_027/0008463. J.D. and P.V. acknowledge the Ministry of the Education, Youth and Science of the Czech Republic for financing the study with project No LO1506 and supporting benchmark data with project No LM2015079.

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
