# Peer review of "Sensitivity of GNSS tropospheric gradients to processing options"

_Annales Geophysicae, 2018_

## Referee Comment (RC1) · Anonymous Referee #1 · 18 Oct 2018

* General comments *

This paper investigates the estimation and modeling of GNSS tropospheric gradients from a benchmark dataset set up for the COST GNSS4SWEC project. Different analysis strategies are evaluated (gradient mapping function, GNSS constellation, cutoff angle, satellites orbit and clock latency: PP vs RT, data weighting) by cross comparisons and also compared with respect to NWP model retrievals. The results are pretty conclusive; comparisons of tropospheric gradient maps are noteworthy (except for some RT cases). PP analysis agree. Positive impact of low elevation observations and multi-constellation is observed. RT analysis induces increase of standard deviations wrt NWP models. Systematic differences induced by the modeling of elevation-dependency of gradients (mapping functions) are also observed; they may be reduced

by the use of an observation elevation weighting. Some recommendations about the use of gradient mapping-function are then expressed according these results.

This paper is very interesting, clear, well organized and also well written. References are relevant and appropriate (and also well formatted).

I recommend the editor to accept the papers with minor revisions according to the following specific comments and technical corrections.
* * *
* Specific comments *

.p03/l13: is it not hazardous to include post-fit residuals into STD formulation? PFRs represent mis-modeling of troposphere, but also for antenna mis calibration, multipath, liquid water, unmodeled solid earth displacements, etc.

.p03/l30: why do not describe further the tilted mapping function as BS and CH?

.p03/l33: in my opinion "Gn*cos(a)+Ge*sin(a)" is not "the projection of the horizontal gradient vector in the direction of the individual satellites": it has to be multiplied by mfg(e), otherwise it is the projection onto zenith of horizontal gradient magnitude.

.p03/l31-p04/l08: I wonder if figure 1 is really useful. A simple comparison of mapping functions plotted according elevation will highlight the maximum values of each mf. The right part is shortly described in text, but it is not used to support any statements. Moreover, the black dots (for a single epoch, 20:30UTC) do not help to support any statements either. Maybe you could just replace this figure by a mfg comparison.

.p03/l15: why did you not use the tilted mfg? I think that its use is not essential since it takes values between BS and CH, but you have to mention it clearly (as a consequence of figure 1).

.p06/l5-p06/l10: as you mention, gradients retrieved from NWP depends on mfg (BS or CH). Why do not use your ray-tracing algorithm to compute gradient with their closed

form expression depending on NS and EW horizontal gradient of refractivity? (See Davis et al., 1993, RS)

.p06-p07: Did the gradient modeling affect the estimation of positions? Maybe you could complete Table2 with comparisons of position (height?) repeatability?

.p07/Table2: I think it is important to have an overview of gradient time series in order to understand the comparisons. Especially, unlike for ZTD we do not have many ideas about gradients magnitude (maybe some ideas from figure 1): is a 0.01 mm bias significant? and a 0.76 mm stdev? These values may be put into perspectives with gradient magnitude.

.p07/Table2: I wonder if the computation of correlation will be helpful to investigate the comparisons. A linear fit?

.p07/Table3: same comments as for table2. Maybe the computation of correlation or linear fit will be more relevant here.

.p10/l13: the RT3GxCH3 do not use Glonass satellite. Why will this solution be affected by Glonass RT corrections?

.p11/l20: Are there any other indications to help to identify these two outlier stations? ZTD, position estimates? Formal errors?

.p13/l3: I think that figure 4 may be described more deeply. First by comparing the impact of the two OEW, then the combined impact of OEW and mfg.

.p13/l5-12: I do not succeed in fully understanding Figure 5 and your remarks related to it (see also next comments). It should be clarified.

.p13/l5: Are differences cumulated during the full day?

.p13/l6: "In this case ... ": I am not sure to understand: did you mean that this figure helps to highlight that systematic differences affect both magnitude and direction?

.p13/l7-l9: Could you explain these two sentences: "A positive difference... points to east" & "Negative... opposite directions". I do not understand (1) how can differences remain positive if you compute A minus with B>A for example (2) how negative values are obtained when gradients point to the opposite direction. This is maybe trivial, but I do not succeed in getting it!

.p13/l10: The decrease of maximum systematic differences with OEW SINEL2 is not obvious.

.p13/l11: Why do not show other weighting, especially SINEL4 which is mentioned to reduce systematic differences?
* * *
* Technical corrections *

I recommend the authors to improve legibility of figures (by using a better resolution)? I also recommend the use of an equation editor for mathematical expressions.

---

## Referee Comment (RC2) · Anonymous Referee #2 · 22 Oct 2018

General Comments

The manuscript evaluates eight different approaches when estimating horizontal gradients in the atmospheric refractive index using signals from two GNSS, namely GPS and GLONASS. As far as I know the content is unique and provides new knowledge, but it also raise questions that I think shall be addressed.

Most important, I think, is the long section with the Conclusions. My interpretation is that the present version has the form of a summary of the results, rather then what is your message to the community on how to handle tropospheric gradients. My conclusion is that it does not really matter which of the different processing option that are chosen given the data that you have studied (excluding the near real time and real time solutions, as expected). Also the small impact of adding GLONASS data may be an

issue to raise for further investigations, possibly related to a higher temporal resolution of the estimated gradients.

Another important question is to what extent your conclusions holds during more general circumstances, because it seems as you have selected the two most extreme months for the benchmark data set. It is of course a lot of work to address this question and give a reliable answer, but it does not prevent you from an initiated discussion in the present manuscript.

An overall question is that I would like to see a more critical discussion related to the numerical weather prediction models. First of all their resolution is poor, given that probably most of the large gradients occur in the atmospheric boundary layer. For example, for an elevation angle of $3°$ the propagation path at the height of 500 m will be approximately 10 km horizontally from the ground-based reference station. That corresponds to the resolution of the limited area model (WRF). One possibility to investigate the scale (temporal as well as spatial) of the gradients is to use the WVR data mentioned in Section 2.1. Since you mention that these data exist the reader will wonder why you do not use them for an assessment, even if the WVR data only exist at a couple of sites.

In terms of how to present your results, I find that your maps in many of your figures give excellent pictures of the systematic spatial variability at specific time epochs. However, I miss examples showing the temporal variability of the gradients over a longer time period that, for example, can give information on for how long time does a large gradient exist and how frequent are the very large gradients.

Specific comments

In the abstract, in Section 2.1, and in the conclusions, you mention observations from 430 GNSS reference stations. It is misleading because as far as I understand the study use data from 243 stations only. This is stated in Section 3. Perhaps the results presented in Section 4 are based on 430 stations? In any case, this issue can be

explained in a better way.

I do not understand Figure 1. I assume that one data point represents one observation from each one of the 243 (or all of the 430?) stations towards each visible (GPS and GLONASS?) satellite? It is stated that it "shows the fractional contribution of the tropospheric gradients". A fractional measure has the unit percent, ppm (or similar), but the units are in mm? Why is the figure included? Even though I did not understand it it did not stop me from reading (and understanding, I hope) the rest of the manuscript. I think that Figure 1 can be removed or otherwise explained more clearly. Furthermore, as I understand, the figure displays results from your analysis, and if you think these results are important you can move the figure to one of the existing result sections (or a new additional one).

In the first paragraph of Section 3 you say that the GNSS gradients are updated every 5 minutes, the WRF model every hour, and the ERA model every 3rd hour. Then you say that the GNSS - NWM comparisons are done every 3 hours. This raise two questions: (1) How did you calculate the GNSS values to be compared to the NWM models (averaging or the actual values at the time epochs given in the NWM time series)? I assume it depends on what is represented by the values in the NWM models. (2) Why not use also the higher temporal resolution available from the WRF model?

When you derive the gradients from the numerical weather models you use a ray tracing method down to elevation angles of 3 degrees. It could then be expected that you find the best agreement when comparing to the GNSS gradients estimated including observations down to an elevation angle of $3°$. I wonder if you can answer the question: if the ray tracing of the numerical weather models would have stopped at an elevation angle of $7°$, would then the GNSS-based gradients, using observations down to $7°$, be the solution with the best agreement?

Technical Corrections

page 1, line 26: vapor? American English, although Ann. Geophys. is a European

journal?

page 1, line 27: numerical weather models -> Numerical Weather Models

page 4, line 13: FLT is a strange acronym for "Kalman filter in RT solutions". Also the acronym SMT is difficult to relate to an expression? I cannot find a definition in the manuscript.

page 4, line 20: "Three additional solutions" are these not the same three solutions that are mentioned in the previous sentence. If so they are not "additional".

page 5, line 9: I assume it shall read (1/sin(ele))^2 ? You say that all variants used this weighting, but it is no longer true in Section 4 where other weighting schemes are investigated.

References: I am not sure how important it is for Ann. Geophys. For most of the journals you do not use the common abbreviations, e.g. Journal of Geophysical Research -> J. Geophys. Res. and Geophysical Research Letters -> Geophys. Res. Lett.

---

## Author Comment (AC1) · 15 Nov 2018

**Author's response on review comments on *Sensitivity of GNSS tropospheric gradients to processing options* by Kačmařík et al.**

Reviews are given in standard black text, author's responses in red italics.

Anonymous Referee #1

\* General comments \*

This paper investigates the estimation and modeling of GNSS tropospheric gradients from a benchmark dataset set up for the COST GNSS4SWEC project. Different analysis strategies are evaluated (gradient mapping function, GNSS constellation, cutoff angle, satellites orbit and clock latency: PP vs RT, data weighting) by cross comparisons and also compared with respect to NWP model retrievals. The results are pretty conclusive; comparisons of tropospheric gradient maps are noteworthy (except for some RT cases). PP analysis agree. Positive impact of low elevation observations and multi-constellation is observed. RT analysis induces increase of standard deviations wrt NWP models. Systematic differences induced by the modeling of elevation dependency of gradients (mapping functions) are also observed; they may be reduced by the use of an observation elevation weighting. Some recommendations about the use of gradient mapping-function are then expressed according these results.

This paper is very interesting, clear, well organized and also well written. References are relevant and appropriate (and also well formatted). I recommend the editor to accept the papers with minor revisions according to the following specific comments and technical corrections.

\*\*\*\*\*\*\*\*\*\*\*\*\*\*\*\*

\* Specific comments \*

.p03/l13: is it not hazardous to include post-fit residuals into STD formulation? PFRs represent mis-modeling of troposphere, but also for antenna mis calibration, multipath, liquid water, unmodeled solid earth displacements, etc.

*We agree and we removed post-fit residuals term from the formula and from the text.*

.p03/l30: why do not describe further the tilted mapping function as BS and CH?

*We now provide a formula for the tilting gradient mapping function as for BS or CH together with a reference to Meindl et al. (2004).*

.p03/l33: in my opinion "Gn\*cos(a)+Ge\*sin(a)" is not "the projection of the horizontal gradient vector in the direction of the individual satellites": it has to be multiplied by mfg(e), otherwise it is the projection onto zenith of horizontal gradient magnitude.

*The projection is just in the horizontal plane, thus not using the gradient mapping function. We substituted the word 'direction' with 'azimuth' as it was used in the figure caption.*

.p03/l31-p04/l08: I wonder if figure 1 is really useful. A simple comparison of mapping functions plotted according elevation will highlight the maximum values of each mf. The

right part is shortly described in text, but it is not used to support any statements. Moreover, the black dots (for a single epoch, 20:30UTC) do not help to support any statements either. Maybe you could just replace this figure by a mfg comparison.

*We decided to remove the dependence on elevation angle and black dots for a single epoch. We then kept the original left figures when directly comparing mapping factors (range in x-axis) and ranges of gradients (scatters in y-axis) for three gradient mapping functions. Based on this figure, we could focus on the two extreme mfg in the following part.*

.p03/l15: why did you not use the tilted mfg? I think that its use is not essential since it takes values between BS and CH, but you have to mention it clearly (as a consequence of figure 1).

*Corrected. We have added a conclusion about tilting from Figure 1, meaning we could further focus on BS and CH only.*

.p06/l5-p06/l10: as you mention, gradients retrieved from NWP depends on mfg (BS or CH). Why do not use your ray-tracing algorithm to compute gradient with their closed form expression depending on NS and EW horizontal gradient of refractivity? (See Davis et al., 1993, RS)

*We calculate the tropospheric gradients from ray-traced delays by least square adjustment to be as close as possible to the method applied in the GNSS analyses for parameters estimation.*

.p06-p07: Did the gradient modeling affect the estimation of positions? Maybe you could complete Table2 with comparisons of position (height?) repeatability?

*There were already published some studies which dealt with positioning changes related to tropospheric gradients estimation which we cite in our paper. Since our focus is only the quality of tropospheric gradients, we would rather not provide results for positioning.*

.p07/Table2: I think it is important to have an overview of gradient time series in order to understand the comparisons. Especially, unlike for ZTD we do not have many ideas about gradients magnitude (maybe some ideas from figure 1): is a 0.01 mm bias significant? and a 0.76 mm stdev? These values may be put into perspectives with gradient magnitude.

*We do not provide gradient time series in this paper, the reader can find them i.e. in the cited publication Li et al. (2015). However, we added a paragraph into the section 3.1 to describe typical values of gradient components under standard and severe weather conditions and we provide information on ZTD rate of change due to tropospheric gradient with an increasing distance from the receiver. The bias of 0.01 mm is not significant in respect to typical values of gradient components and also according to estimated standard deviation of comparisons.*

.p07/Table2: I wonder if the computation of correlation will be helpful to investigate the comparisons. A linear fit?

*We now provide linear correlation coefficients in table 2 to and table 3 and we updated the text in this regards.*

.p07/Table3: same comments as for table2. Maybe the computation of correlation or linear fit will be more relevant here.

*See answer for the previous comment.*

.p10/l13: the RT3GxCH3 do not use Glonass satellite. Why will this solution be affected by Glonass RT corrections?

*You are right, it can't be affected. The manuscript was updated.*

.p11/l20: Are there any other indications to help to identify these two outlier stations? ZTD, position estimates? Formal errors?

*We checked ZTDs and formal errors for ZTDs and tropospheric gradients, but we haven't found anything extraordinary for these two outlier stations. We have not checked their coordinates so far.*

.p13/l3: I think that figure 4 may be described more deeply. First by comparing the impact of the two OEW, then the combined impact of OEW and mfg.

*More description is provided in updated manuscript according to the proposed way.*

.p13/l5-12: I do not succeed in fully understanding Figure 5 and your remarks related to it (see also next comments). It should be clarified.

*We updated the corresponding paragraph, hopefully improving its clarity.*

.p13/l5: Are differences cumulated during the full day?

*We substituted 'cumulated differences during the full day' with 'mean differences calculated over the full day', and we hope it is more precise and clear in understanding.*

.p13/l6: "In this case ... ": I am not sure to understand: did you mean that this figure helps to highlight that systematic differences affect both magnitude and direction?

*No. We just noted that both the magnitude and direction of an estimated gradient resulted in different systematic errors when decomposed into North/East gradient components. The paragraph has been reworded.*

.p13/l7-l9: Could you explain these two sentences: "A positive difference... points to east" & "Negative... opposite directions". I do not understand (1) how can differences remain positive if you compute A minus with B>A for example (2) how negative values are obtained when gradients point to the opposite direction. This is maybe trivial, but I do not succeed in getting it!

*Indeed, the description was reversed when considering B>A. It seems that we misattributed B and A after some changes in the representation of gradient differences. Thank you very much for careful reading!*

.p13/l10: The decrease of maximum systematic differences with OEW SINEL2 is not obvious.

*When displaying now all the OEW schemes, we hope the decrease of systematic errors in Figure 4 and 5 become more visible.*

.p13/l11: Why do not show other weighting, especially SINEL4 which is mentioned to reduce systematic differences?

*Initially, we thought it's too much showing all of them, in particularly when SINEL1/SINEL2 is the most commonly used OEW in analyses. We added now all of them for a complete comparison.*

\*\*\*\*\*\*\*\*\*\*\*\*\*\*\*\*

\* Technical corrections \*

I recommend the authors to improve legibility of figures (by using a better resolution)? I also recommend the use of an equation editor for mathematical expressions.

*In the updated manuscript version all the equations are created using an equation editor.*

General Comments

The manuscript evaluates eight different approaches when estimating horizontal gradients in the atmospheric refractive index using signals from two GNSS, namely GPS and GLONASS. As far as I know the content is unique and provides new knowledge, but it also raise questions that I think shall be addressed.

Most important, I think, is the long section with the Conclusions. My interpretation is that the present version has the form of a summary of the results, rather then what is your message to the community on how to handle tropospheric gradients. My conclusion is that it does not really matter which of the different processing option that are chosen given the data that you have studied (excluding the near real time and real time solutions, as expected). Also the small impact of adding GLONASS data may be an issue to raise for further investigations, possibly related to a higher temporal resolution of the estimated gradients.

*From our point of view, we provide a recommendation to the user everywhere we think it can be given based on our own results: 1, we recommend using observations from very low elevation angles to get better gradients (this was already shown also in paper by Meindl et al. (2004) which we cite). 2, we find a small positive impact coming from adding GLONASS in our processing (it can be however different when using other products with satellite ephemerides and clock error corrections, different weighting of observations from various GNSS, etc., and some other investigations related to multi-GNSS data processing in general will follow). 3, we present the penalty in quality of tropospheric gradients from real-time processing and we show that this penalty is mainly related to the quality of used products. 4, we show that selection of gradient mapping function does not affect general quality of estimated tropospheric gradients but their magnitudes (one has to be careful then with comparing gradients from various sources due to existing systematic differences).*

*We updated the conclusion section of the paper to address your comment.*

Another important question is to what extent your conclusions holds during more general circumstances, because it seems as you have selected the two most extreme months for the benchmark data set. It is of course a lot of work to address this question and give a reliable answer, but it does not prevent you from an initiated discussion in the present manuscript.

*We based our analyses on a data set from wet spring/summer season when the gradients could provide a valuable information for meteorological applications. Although the time period covers some severe weather events, it also contains a lot of days with standard weather conditions with tropospheric gradients close to zero. So, the results should provide a good overview on the situation in Europe during the warmer part of the year. On the other hand, we agree that new studies based on different GNSS software a data sets should be done to strengthen and confirm our results.*

An overall question is that I would like to see a more critical discussion related to the numerical weather prediction models. First of all their resolution is poor, given that probably

most of the large gradients occur in the atmospheric boundary layer. For example, for an elevation angle of 3° the propagation path at the height of 500 m will be approximately 10 km horizontally from the ground-based reference station. That corresponds to the resolution of the limited area model (WRF). One possibility to investigate the scale (temporal as well as spatial) of the gradients is to use the WVR data mentioned in Section 2.1. Since you mention that these data exist the reader will wonder why you do not use them for an assessment, even if the WVR data only exist at a couple of sites.

*We split our reaction into two parts:*

*1, we would be very careful about the statement that large gradients occur in the atmospheric boundary layer. Please see i.e. a paper from Elosegui et al. (1999). According to his findings GNSS tropospheric gradients are more sensitive to tropospheric features at larger heights, in different words – i.e. the same type of tropospheric feature at the height of 3 km would cause a larger gradient value than while occurring at the height of 0.5 km. And also sometimes (even during not winter season) a hydrostatic gradient can prevail in total tropospheric gradient estimated by GNSS. And these hydrostatic gradients are related to large scale (up to several hundreds of km) features, not to local station asymmetry. Of course, we are aware of limitations of NWMs we use in this study and we also state them in the paper. On the other hand, this is the first time when a NWM with a 10 km horizontal resolution was used for comparisons with GNSS results and there is a visible increase of its gradient magnitudes compared to outputs of global NWMs with 1° horizontal resolution used in Douša et al. (2016).*

*2, the usage of WVR for this study is problematic from several reasons: a) it is available only for a single station (POTS, Germany) for the benchmark campaign; b) WVR measures IWV therefore it can deliver only wet delay gradient, a hydrostatic gradient would need to be added from an external source (NWM) to get a total gradient which is delivered by GNSS; c) data quality of WVR observations at elevation angles below approximately 20-25° is generally poor (see i.e. Kačmařík et al., 2017, AMT).*

In terms of how to present your results, I find that your maps in many of your figures give excellent pictures of the systematic spatial variability at specific time epochs. However, I miss examples showing the temporal variability of the gradients over a longer time period that, for example, can give information on for how long time does a large gradient exist and how frequent are the very large gradients.

*In this paper we focused on GNSS data processing and we provided only few examples of maps of tropospheric gradients to demonstrate some systematic effects. Currently we are working on another study where we utilize mainly these maps including longer time series so we are not going to extend this paper in this direction. The duration of large gradient presence is simply given by the prevailing meteorological situation and therefore is strongly variable. Usually it ranges between several tens of minutes and several hours.*

Specific comments

In the abstract, in Section 2.1, and in the conclusions, you mention observations from 430 GNSS reference stations. It is misleading because as far as I understand the study use data from 243 stations only. This is stated in Section 3. Perhaps the results presented in Section 4 are based on 430 stations? In any case, this issue can be explained in a better way.

*In total, benchmark data set contains data from 430 GNSS stations which were processed for all introduced variants of solution. However, from given reasons the statistics presented in section 3 are based only on a subset of 243 stations. To make it more clear for the reader we updated the manuscript in the introduction par of section 3 and in the conclusion section.*

I do not understand Figure 1. I assume that one data point represents one observation from each one of the 243 (or all of the 430?) stations towards each visible (GPS and GLONASS?) satellite? It is stated that it "shows the fractional contribution of the tropospheric gradients". A fractional measure has the unit percent, ppm (or similar), but the units are in mm? Why is the figure included? Even though I did not understand it did not stop me from reading (and understanding, I hope) the rest of the manuscript. I think that Figure 1 can be removed or otherwise explained more clearly. Furthermore, as I understand, the figure displays results from your analysis, and if you think these results are important you can move the figure to one of the existing result sections (or a new additional one).

*You are right, we removed the word 'fractional' and kept just 'contribution'. Horizontal gradients, projected into azimuths of individual satellites, are expressed in mm. Figure 1 shows gradients from all 430 stations. Based on comments from the Reviewer #1, we removed one of the plots, improved the text description and concluded with further study of extreme mfg only (BS and CH).*

In the first paragraph of Section 3 you say that the GNSS gradients are updated every 5 minutes, the WRF model every hour, and the ERA model every 3rd hour. Then you say that the GNSS - NWM comparisons are done every 3 hours. This raise two questions: (1) How did you calculate the GNSS values to be compared to the NWM models (averaging or the actual values at the time epochs given in the NWM time series)? I assume it depends on what is represented by the values in the NWM models. (2) Why not use also the higher temporal resolution available from the WRF model?

*Newly we base our GNSS versus NWM comparisons on 1-hour interval, all the results in the paper were updated accordingly to that and they changed insignificantly. To answer the question, we used GNSS results from the time epochs of NWM outputs for which they were estimated in the GNSS data processing, we have not done any averaging. NWM output is always given for a specific time epoch.*

When you derive the gradients from the numerical weather models you use a ray tracing method down to elevation angles of 3 degrees. It could then be expected that you find the best agreement when comparing to the GNSS gradients estimated including observations down to an elevation angle of 3°. I wonder if you can answer the question: if the ray tracing of the numerical weather models would have stopped at an elevation angle of 7°, would

then the GNSS-based gradients, using observations down to 7°, be the solution with the best agreement?

*To answer your question, we derived a new complete set of tropospheric gradients from both ERA5 and WRF models using a 7° cut-off angle for the ray-tracing. Firstly, we compared gradients from such two versions of solution from one NWM and found a difference at the level of 0.1 mm or below 0.1 mm for RMSE. Secondly, we compared these outputs with GNSS solutions and we always found a better agreement between any evaluated GNSS variant of solution and the NWM solution based on the 3° cut-off angle – in terms of bias, standard deviation and correlation coefficient. We comment on this in section 3.2 in the updated version of the manuscript.*

*We interpret this situation as follows: GNSS use only a limited set of observations and have to deal with a set of (unknown) parameters like receiver position, troposphere, satellite clock error, etc. And errors in estimation of some parameter(s) can influence other parameter(s). Observations from very low elevation angles include a strong influence from tropospheric delay and therefore can help to accurately estimate it. On the other hand, the applied NWM ray-tracing technique uses a much larger number of "observations" which are not affected by other effects or observation noise to estimate the tropospheric gradients and is therefore not dependent on observation elevation weighting or applied cut-off angle.*

Technical Corrections

page 1, line 26: vapor? American English, although Ann. Geophys. is a European journal?

*Corrected.*

page 1, line 27: numerical weather models -> Numerical Weather Models

*Corrected.*

page 4, line 13: FLT is a strange acronym for "Kalman filter in RT solutions". Also the acronym SMT is difficult to relate to an expression? I cannot find a definition in the manuscript.

*FLT means a filter, SMT a smoother. We do not find these acronyms as problematic, therefore we kept them in the paper.*

page 4, line 20: "Three additional solutions" are these not the same three solutions that are mentioned in the previous sentence. If so they are not "additional".

*The word additional was deleted.*

page 5, line 9: I assume it shall read $(1/\sin(ele))^2$ ? You say that all variants used this weighting, but it is no longer true in Section 4 where other weighting schemes are investigated.

*Formula was corrected to avoid misunderstanding. All variants of solution introduced in section 2.2 (table 1) and evaluated in section 3 used this weighting scheme. We don't want to confuse the reader with extra information on elevation weighting in section 2, therefore we keep the sentence as it was.*

References: I am not sure how important it is for Ann. Geophys. For most of the journals you do not use the common abbreviations, e.g. Journal of Geophysical Research -> J. Geophys. Res. and Geophysical Research Letters -> Geophys. Res. Lett.

*Corrected, manuscript was updated.*

---

## Editor Decision (ED1)

Dear Authors,

I have read your answers to the two Referee's comments. In general, they are satisfying and revisions should be implemented accordingly in the manuscript. However, some of the referee comments are not or are incompletely answered, and some answers call for further argumentation and/or additional details. Below you will find my specific remarks on the referees' comments and your answers. You will also find my own comments on the manuscript in the last section. Please provide a line-by-line response to all remarks and comments and submit a revised version of the manuscript.

Best regards,

Olivier BOCK

Co-editor ANGEO for the GNSS4WEC Special Issue

**Remarks on answers to Referee No. 1.**

Referee comments in blue, your answers in red, my remarks and comments in black.

As a preliminary general comments, I would like to draw your attention to the fact that this Referee raised 5 comments on page 3 (section 2.2 / Figure 1) and 7 comments on page 13 (Section 4 and Figs. 4 and 5), and that Referee No. 2 was also concerned about Figure 1 and its related comments. Please provide a careful revision of these sections of the manuscript. Some suggestions are given below.

p03/l13: is it not hazardous to include post-fit residuals into STD formulation? PFRs represent mis-modeling of troposphere, but also for antenna mis calibration, multipath, liquid water, unmodeled solid earth displacements, etc.

We agree and we removed post-fit residuals term from the formula and from the text.

Please answer the issue raised by the referee regarding the information contained in the post-fit residuals, especially as post-fit residuals are shown in Fig. 6.

p03/l33: in my opinion "Gn*cos(a)+Ge*sin(a)" is not "the projection of the horizontal gradient vector in the direction of the individual satellites": it has to be multiplied by mfg(e), otherwise it is the projection onto zenith of horizontal gradient magnitude.

*The projection is just in the horizontal plane, thus not using the gradient mapping function. We substituted the word 'direction' with 'azimuth' as it was used in the figure caption.*

I think the misunderstanding here comes from what you call "gradients" in Fig. 1 which is actually the component of STD due to azimuthal asymmetry of the atmosphere (Chen and Herring, 1997). The term "gradient" should be restricted to the vector **G** (Gn, Ge). It would be much more clear if you define the quantity plotted in Fig. 1 by an equation as I already suggested in the preliminary review.

p03/l31-p04/l08: I wonder if figure 1 is really useful. A simple comparison of mapping functions plotted according elevation will highlight the maximum values of each mf. The right part is shortly described in text, but it is not used to support any statements. Moreover, the black dots (for a single epoch, 20:30UTC) do not help to support any statements either. Maybe you could just replace this figure by a mfg comparison.

We decided to remove the dependence on elevation angle and black dots for a single epoch. We then kept the original left figures when directly comparing mapping factors (range in x-axis) and ranges of gradients (scatters in y-axis) for three gradient mapping functions. Based on this figure, we could focus on the two extreme mfg in the following part.

I agree that Fig. 1 is useful here to illustrate the variation of the delay due to the horizontal gradients and the size of the mapping factors, given that it is clarified as suggested above.

p06/l5-p06/l10: as you mention, gradients retrieved from NWP depends on mfg (BS or CH). Why do not use your ray-tracing algorithm to compute gradient with their closed form expression depending on NS and EW horizontal gradient of refractivity? (See Davis et al., 1993, RS)

We calculate the tropospheric gradients from ray-traced delays by least square adjustment to be as close as possible to the method applied in the GNSS analyses for parameters estimation.

You justify what you did but you don't really answer the referee's question. I think the referee suggested computing Gn and Ge from the integrals defined e.g. in Davis et al., 1993. This is another approach. Did you compare both approaches? Is there a reason why your approach is preferred?

p06-p07: Did the gradient modeling affect the estimation of positions? Maybe you could complete Table2 with comparisons of position (height?) repeatability?

There were already published some studies which dealt with positioning changes related to tropospheric gradients estimation which we cite in our paper. Since our focus is only the quality of tropospheric gradients, we would rather not provide results for positioning.

I think the referee's question and suggestion are relevant. Inspecting the station repeatability is a standard approach to evaluating the quality of the processing, overall, and might thus help assess the relevance of the different gradient models tested. It would certainly be useful and interesting to check the station height repeatability. This is an additional validation to the NWP comparisons.

p07/Table2: I think it is important to have an overview of gradient time series in order to understand the comparisons. Especially, unlike for ZTD we do not have many ideas about gradients magnitude (maybe some ideas from figure 1): is a 0.01 mm bias significant? and a 0.76 mm stdev? These values may be put into perspectives with gradient magnitude.

We do not provide gradient time series in this paper, the reader can find them i.e. in the cited publication Li et al. (2015). However, we added a paragraph into the section 3.1 to describe typical values of gradient components under standard and severe weather conditions and we provide information on ZTD rate of change due to tropospheric gradient with an increasing distance from the receiver. The bias of 0.01 mm is not significant in respect to typical values of gradient components and also according to estimated standard deviation of comparisons.

The publication by Li et al. (2015) that you are citing here and in the manuscript only shows time series for Onsala station (not included in your study) and for a different period. They are not directly linked with the results of you study. By the way, the second referee also asked for time series. I agree with their comments that it would be useful to give some insight into the temporal variability and size of variations, rather than just giving values for means or extreme conditions.

I'm not sure that the rate of change of ZTD with distance is a very relevant information as it is not directly related with a measurable quantity. Instead, you could quantify the contribution of gradient to the STD given by the 2nd part in Eq. (1).

I agree with the both referees that some indication of the significance of the numbers should be given especially since the gradient values (Gn, Ge) are very small (e.g. 0.3 mm quoted in the Abstract). Significance statements should be supported by a statistical test.

p07/Table2: I wonder if the computation of correlation will be helpful to investigate the comparisons. A linear fit?

*We now provide linear correlation coefficients in table 2 to and table 3 and we updated the text in this regards*

Please answer the questions. Does the provision of correlations/linear fit help in the interpretation of the results? If it doesn't, you just have to mention it and not include the (unnecessary) results.

p07/Table3: same comments as for table2. Maybe the computation of correlation or linear fit will be more relevant here.

See answer for the previous comment.

Please answer the question.

I recommend the authors to improve legibility of figures (by using a better resolution)? I also recommend the use of an equation editor for mathematical expressions.

In the updated manuscript version all the equations are created using an equation editor.

What about the legibility of Figures?

For the final revised paper, I recommend that you provide the maps in uniform format.

**Remarks on answers to Referee No. 2.**

Referee comments in blue, your answers in red, my remarks and comments in black.

… My interpretation is that the present version has the form of a summary of the results, rather then what is your message to the community on how to handle tropospheric gradients. My conclusion is that it does not really matter which of the different processing option that are chosen given the data that you have studied (excluding the near real time and real time solutions, as expected). Also the small impact of adding GLONASS data may be an issue to raise for further investigations, possibly related to a higher temporal resolution of the estimated gradients.

From our point of view, we provide a recommendation to the user everywhere we think it can be given based on our own results: 1, we recommend using observations from very low elevation angles to get better gradients (this was already shown also in paper by Meindl et al. (2004) which we cite). 2, we find a small positive impact coming from adding GLONASS in our processing (it can be however different when using other products with satellite ephemerides and clock error corrections, different weighting of observations from various GNSS, etc., and some other investigations related to multi-GNSS data processing in general will follow). 3, we present the penalty in quality of tropospheric gradients from real-time processing and we show that this penalty is mainly related to the quality of used products. 4, we show that selection of gradient mapping function does not affect general quality of estimated tropospheric gradients but their magnitudes (one has to be careful then with comparing gradients from various sources due to existing systematic differences).

Please revise the Conclusion section according to the referee's comments and highlight clearly the four points you mention in your answer. Additionally, I would like to see a more insightful discussion of your results, including a comparison to other past and recent studies.

1. Revise the statements regarding the study by Meindl et al. (2004). The improvement in precision at lower elevation angles they noticed is only based on rms errors (formal errors) of the gradient parameters. This is a well-known result, explained by the use of more observations and better decorrelation of gradients from other parameters. It does not say whether they are more accurate. Accuracy of the gradient estimates can only be tested by comparison with independent data (e.g. from VLBI, MWR, NWP). Accuracy of the processing can also be tested by inspecting the coordinate repeatability, which e.g. Meindl et al. (2004) did when they compared two processing variants, with and without estimation of gradients, but not for different cutoff angles. This conclusion is actually not consistent with Zhou et al. (2017) who observed better results at 7° and 10° compared to 3°. Can you comment on this?

2. Discuss the sensitivity of results to addition of GLONASS data, and compare to other recent studies (e.g. Zhou et al., 2017).

3. RT1 solution is slightly more accurate than the RT3 solution. Can you elaborate a bit more your comments on the quality of the used RT clock and orbit products?

4. I don't understand what you mean by the 'quality' of the estimated gradients. I would suppose you mean their accuracy, but if their magnitude is affected this means their accuracy is impacted as well. Please clarify.

The conclusion that the magnitude of gradient parameter estimates is changing depending on the used mapping function is important and should be better highlighted. Given the topic of this paper, it is expected that this point is thoroughly discussed and that strong conclusions and recommendations are given and not a final remark such as "it is hard to assess which *mfg* is more suitable for the troposphere modelling in GNSS analyses".

The issue of time sampling of the gradient parameters should be discussed as well. Though it is not a limitation with the software you used, this question is of concern to many other scientists who use different software packages (e.g. Bernese, GAMIT) where it is more common to use a 24-h sampling.

Another important question is to what extent your conclusions holds during more general circumstances, because it seems as you have selected the two most extreme months for the benchmark data set. It is of course a lot of work to address this question and give a reliable answer, but it does not prevent you from an initiated discussion in the present manuscript.

We based our analyses on a data set from wet spring/summer season when the gradients could provide a valuable information for meteorological applications. Although the time period covers some severe weather events, it also contains a lot of days with standard weather conditions with tropospheric gradients close to zero. So, the results should provide a good overview on the situation in Europe during the warmer part of the year. On the other hand, we agree that new studies based on different GNSS software a data sets should be done to strengthen and confirm our results.

Please add these comments in the revised manuscript. But I don't understand your point about the necessity of new studies based on different software. Can elaborate it? Maybe it is worth discussing this point in the paper.

An overall question is that I would like to see a more critical discussion related to the numerical weather prediction models. First of all their resolution is poor, given that probably most of the large gradients occur in the atmospheric boundary layer. For example, for an elevation angle of 3° the propagation path at the height of 500 m will be approximately 10 km horizontally from the ground-based reference station. That corresponds to the resolution of the limited area model (WRF). One possibility to investigate the scale (temporal as well as spatial) of the gradients is to use the WVR data mentioned in Section 2.1. Since you mention that these data exist the reader will wonder why you do not use them for an assessment, even if the WVR data only exist at a couple of sites.

*We split our reaction into two parts:*

*1, we would be very careful about the statement that large gradients occur in the atmospheric boundary layer. Please see i.e. a paper from Elosegui et al. (1999). According to his findings GNSS tropospheric gradients are more sensitive to tropospheric features at larger heights, in different words – i.e. the same type of tropospheric feature at the height of 3 km would cause a larger gradient value than while occurring at the height of 0.5 km. And also sometimes (even during not winter season) a hydrostatic gradient can prevail in total tropospheric gradient estimated by GNSS. And these hydrostatic gradients are related to large scale (up to several hundreds of km) features, not to local station asymmetry. Of course, we are aware of limitations of NWMs we use in this study and we also state them in the paper. On the other hand, this is the first time when a NWM with a 10 km horizontal resolution was used for comparisons with GNSS results and there is a visible increase of its gradient magnitudes compared to outputs of global NWMs with 1° horizontal resolution used in Douša et al. (2016).*

*2, the usage of WVR for this study is problematic from several reasons: a) it is available only for a single station (POTS, Germany) for the benchmark campaign; b) WVR measures IWV therefore it can deliver only wet delay gradient, a hydrostatic gradient would need to be added from an external source (NWM) to get a total gradient which is delivered by GNSS; c) data quality of WVR observations at elevation angles below approximately 20-25° is generally poor (see i.e. Kačmařík et al., 2017, AMT).*

Please provide full detail of the references you are citing (Elosegui, Dousa, Kacmarik).

It seems that there is an interesting point of debate here on which atmospheric layer is impacting most the estimated gradient parameters. Please add some elements of this discussion in the manuscript regarding the sensitivity of tropospheric delay gradient parameters to hydrostatic and wet gradients in the atmospheric refractivity. I think this was also discussed in early papers by Davis et al., Radio Science (1993) or Chen and Herring, JGRB (1997).

Reconcile also your position with your statement in a previous publication "The first-order horizontally asymmetric delay … reflects local changes in temperature and particularly in water vapour." (*Kacmarik et al., AMT, 2017*).

Regarding your 1[st] point, I know at least one study which compared GPS gradients and gradients from a NWP with a resolution of 0.1°x0.1°, and it is from 2002:

Walpersdorf, A., E. Calais, J. Haase, L. Eymard, M. Desbois and H. Vedel, Atmospheric gradients estimated by GPS compared to a high resolution numerical weather prediction (NWP) model. Physics and Chemistry of the Earth, Part A: Solid Earth and Geodesy, Vol. 26 (3), pp. 147-152, 2002.

Regarding your 2nd point, I am surprised of your answer because: a) A single station was used in your previous publication (Kacmarik et al., AMT, 2017) as well as in Lu et al., JGR, 2016, and Li et al. GRL, 2015; b) You solved the problem of hydrostatic gradient in Kacmarik et al., AMT, 2017, for the same COST Benchmark dataset, so it would be a strong argument to re-use these WVR data to compute gradient parameters and use them in this study as an additional source of validation; c) you suspect the quality of WVR data is poor below an elevation of angle 20-25° (even 40° is the quoted in Kacmarik et al., 2017) but according to the numerous publications which used such data, one would rather suspect that there was an issue with the data or the comparisons used in your previous study. This last point merits probably further investigation and should be commented.

Lu, C., X. Li, Z. Li, R. Heinkelmann, T. Nilsson, G. Dick, M. Ge, and H. Schuh (2016), GNSS tropospheric gradients with high temporal resolution and their effect on precise positioning, *J. Geophys. Res. Atmos.*, 121, 912–930, doi: 10.1002/2015JD024255.

**Additional Editor comments.**

The additional comments below should help to strengthen the relevance of the manuscript for the final publication.

**General comments**

One of your strong conclusions is that changing the gradient mapping function impacts the magnitude of the estimated gradient parameters. I would surmise that this impacts also the ZTD estimates and possibly the STD estimates. Can you comment on this point? Some quantitative results would be useful (e.g. correlation analysis of ZTD/gradient/STD differences).

Compare and discuss the results from this study to similar recent studies such as Li et al., 2015; Lu et al., 2015, 2016; Zhou et al., 2017; etc. who also evaluated GPS and GNSS gradient estimates, namely in against NWP data and WVR data.

Provide more insight into the significance of the gradient magnitudes and differences as the numbers reported in the manuscript are very small (< 1 mm) and may seem negligible to many readers. This should be done by using a statistical significance test when comparing two results. For instance, a 0.01 mm difference when adding GLONASS data (change of SDEV from 0.48 to 0.49 mm, quoted as 2% in the conclusions) is probably not significant but a 0.06 mm difference when changing the mapping function (from 0.48 to 0.42 mm) probably is. This would also help you to clearly (objectively) state and conclude about the results.

Better insight into the results would be brought by changing slightly the order of presentation. Starting with present subsection 3.3 and Tables 4 and 5 would help introducing the magnitudes, directions and formal errors of gradients (answering some comments by the referees), and provide a first discussion of the sensitivity of different processing options based on statistics for GNSS results only. Time series could be included here. Then, the variants could be inter-compared (present subsection 3.1) and finally the validation using NWP data and maybe the WVR data.

Finally, a major revision is required in Section 4 and in the discussion/conclusions as also asked by the referees. In its present form Section 4 appears as an additional piece of work on the impact of the observation elevation-dependent weighting, but for a single day. Instead, it should be included in the main study, as a dedicated sub-section if you wish, and treated with the same methodology (243 stations, 55 days). It is an important aspect of the sensitivity study, as you mention in the

Introduction, which merits a proper treatment. The particular day discussed could be used as a case study, in a similar way as done with Fig. 2 and 3.

**Specific comments**

P1L13: include "observation elevation-dependent weighting" as one of the processing aspects that are investigated

P1L16: "a clear relation to real weather conditions" appeals for two comments: 1) such as statement would imply that you studied the results with more information and description on the weather situations (instead, I only found in the manuscript indication that "a weather front was passing over the studied area" P9L6); 2) Isn't it obvious that there is a relation with the gradient maps and weather conditions? (otherwise it would be really problematic using GNSS data for meteorology). Please be more specific. You can write that you illustrated gradient maps in the case of two weather fronts, etc.

P1L18: "systematic effects…" assumes that the results are based on some average (not just one day).

P1L20-22: the global scale results are out of scope for the paper; and the last part of the sentence is not clear (are the large local gradients the cause of differences? Difference wrt to what?)

P2L7: "Dousa et al. (2018a) demonstrated the advantage of similar pseudo-observations in the 2-stage troposphere model combining optimally NWM and GNSS data". What are pseudo-observations? What do you mean by combining optimally NWM and GNSS data? Be more specific on objectives, methods and results/conclusions of the cited study like you did for Brenot et al., Morel, etc.

P2L20: "systematic errors": Gradient estimates from NWP models used in this study are computed using the CH gradient mapping function (P6L8-10). This choice implies a systematic difference compared to e.g. BS gradient mapping function. For this reason, it is not possible to assess the bias, or systematic, or absolute errors of the GNSS gradient estimates. Please be careful here and throughout the manuscript about using the term "bias" and prefer systematic or mean difference instead.

P3 Eq. (2) and (3), and in other equations, use 'e' for denoting the elevation angle rather than 'ele'.

P4L4: "inversely proportional" implies a 1/x variation but this is not the case (and the demonstration would require fitting a function in the data). Please reformulate.

P6L1-2: "under the assumption of a spherically layered atmosphere" from the local vertical profile?

P6L3: say here that this fitting is done using the CH mapping function in this study.

P6L6: what is the "standard elevation angle dependent weighting"? Be more specific.

P6L8-10: "comparison with GNSS gradients estimated with BS mfg should be treated cautiously" This sounds like a warning to the readers or to the users of your results. I think it is your work to interpret your results properly given this limitation. Please reformulate and add comments on the BS results where presented.

P6L15: is there a reason/explanation why you chose 5% as the limit?

P7L3: explain why you expect "negative biases"

P7L9: "The two RT solutions can be still considered of good quality if we take into consideration results found in Ahmed et al. (2016) or Kačmařík (2018)." Be more specific.

Table 2: reverse the differences in the RT results: RT1GxCH3 - GxCH3 (etc.) instead of GxCH3 - RT1GxCH3, as it is standard to use the reference data to the right.

Indicate the number of data points used in the computation of statistics.

Could the increase in standard deviation for RT3 be due to unrealistic cases such as illustrated in Fig. 3? Some data screening should be applied before computing the statistics.

P8L12: "Obviously, NWMs cannot be regarded as a ground truth" Should be reformulated.

P9L6: "when a weather front was passing" use plural as they were different events

P9L11: "A detailed evaluation of tropospheric gradient maps with meteorological observations will be a subject of an upcoming study." This sentence suggests that the present study is not complete. I suggest removing it. But I suggest also to include more meteorological information on the cases illustrated in the present study, otherwise it is difficult to conclude on the relevance of GNSS gradients regarding the meteorological situations.

P11L21: "a developed area in south-west Germany" sound awkward, please reformulate.

P12L3: "It suggests to reflect only the impact of differences in mapping factors on calculating formal errors." I don't understand how the mapping factors influence the formal errors.

P13L12: "all the OEW schemes demonstrate a strong impact of low-elevation observations reflecting an actual local tropospheric asymmetry in the water vapour distribution" this sentence is a bit surprising for two reasons: 1) the OEW schemes are expected to reduce the impact of noisy low-elevation observations and 2) their impact is expected to be quite different. And second remark: how can you be sure the local tropospheric asymmetry is in the water vapour distribution?

P14L6-10: the description of figure 6 is not clear. To me the results in the right panel (EQUAL) look more homogeneous whereas in the left panel (SINEL2) there is a steep variation with elevation (even above 30° where it becomes more difficult to see but I guess the shape more or less varies as 1/sin(e)). Please correct.

P14L10-12: what is "a more realistic view"? This sentence is not clear and too long. Please reformulate.

The following part of the sentence (with corrections) could be introduced at the start of the paragraph to explain what is expected. "errors in GNSS observations which are expected to increase with a decrease of elevation angle,  due to multipath effects, uncertainty of receiver antenna phase centre variations, lower signal-to-noise ratio,  or cycle slips." Obstructions do not produce errors, they simply cut off the observations.

P15L8: "all post-processing solutions can be regarded as robust and their gradient estimates are clearly related to real weather conditions" I didn't get how you checked the robustness of the solutions, maybe use another term or specify based on which diagnostics you can conclude that. And the relation to real weather conditions was not demonstrated, only two cases were illustrated (and not based on meteorological diagnostics).

P15L9: "which are fully independent of meteorological input data" please clarify what you mean here. I think the GNSS processing is not completely independent of NWP data since mapping functions are derived from NWP data (even GMF).

P15L10: "tropospheric gradients thus provide additional interesting information in support of NWM forecasts" forecasting is not the only application, and in the Introduction you wrote that gradients are not assimilated. Please correct.

P15L11-13: Need be reformulated e.g. "Better agreement was found between GNSS gradients and NWM when the cut-off elevation angle was decreased from 7° to 3° (the standard deviation of differences decrease by 10 %), for both the single- and dual- constellation results."

P15L27-32: needs be reformulated and integrated into the main discussion.

P15L32-33: the long-term mean gradient results in the Appendix are not part of the main study.

P16L1-5: needs be reformulated (see comments above).

Appendix A: the discussion should be stand-alone. Instead of citing Meindl et al. (2004), I suggest that you add maps of mean GN and GE from ERA5, or maybe only GN but for CH and BS *mfg*. The figure should be labelled A1.

**Technical corrections:** see the annotated manuscript

[revised manuscript text omitted]

---

## Author Response (AR2)

*February 22, 2019*
Dear editor,
We now provide point-to-point reactions also to your own comments as well as an updated version of the manuscript.

Yours Sincerely,
Authors.

*January 16, 2019*
Dear editor,
We provide point-to-point reactions to your comments within the two Referee's comments. Our text is written in green italics. We updated the manuscript accordingly and provide its new version (and also a version with tracked changes).

Yours Sincerely,
Authors.

Dear Authors,

I have read your answers to the two Referee's comments. In general, they are satisfying and revisions should be implemented accordingly in the manuscript. However, some of the referee comments are not or are incompletely answered, and some answers call for further argumentation and/or additional details. Below you will find my specific remarks on the referees' comments and your answers. You will also find my own comments on the manuscript in the last section. Please provide a line-by-line response to all remarks and comments and submit a revised version of the manuscript.

Best regards,

Olivier BOCK

Co-editor ANGEO for the GNSS4WEC Special Issue

**Remarks on answers to Referee No. 1.**

Referee comments in blue, your answers in red, my remarks and comments in black.

As a preliminary general comments, I would like to draw your attention to the fact that this Referee raised 5 comments on page 3 (section 2.2 / Figure 1) and 7 comments on page 13 (Section 4 and Figs. 4 and 5), and that Referee No. 2 was also concerned about Figure 1 and its related comments. Please provide a careful revision of these sections of the manuscript. Some suggestions are given below.

p03/l13: is it not hazardous to include post-fit residuals into STD formulation? PFRs represent mis-modeling of troposphere, but also for antenna mis calibration, multipath, liquid water, unmodeled solid earth displacements, etc.

We agree and we removed post-fit residuals term from the formula and from the text.

Please answer the issue raised by the referee regarding the information contained in the post-fit residuals, especially as post-fit residuals are shown in Fig. 6.

*We fully agree with the referee about the content of post-fit residuals and we excluded the term from the equation 1 in previous revision. Anyway, the equation is included purely for the definition of tropospheric model used for estimating ZTD and horizontal gradient parameters, not for retrieving slant tropospheric delays.*
*We also slightly edited the paragraph describing Figure 7 (previously Figure 6) where the constitution of the post-fit residuals was also mentioned. However, the figure is again used purely for showing differences of applied GNSS models when looking at elevation dependence of post-fit residuals, and not considered for reconstructing slant delays.*

p03/l33: in my opinion "Gn*cos(a)+Ge*sin(a)" is not "the projection of the horizontal gradient vector in the direction of the individual satellites": it has to be multiplied by mfg(e), otherwise it is the projection onto zenith of horizontal gradient magnitude.

*The projection is just in the horizontal plane, thus not using the gradient mapping function. We substituted the word 'direction' with 'azimuth' as it was used in the figure caption.*

I think the misunderstanding here comes from what you call "gradients" in Fig. 1 which is actually the component of STD due to azimuthal asymmetry of the atmosphere (Chen and Herring, 1997). The term "gradient" should be restricted to the vector **G** (Gn, Ge). It would be much more clear if you define the quantity plotted in Fig. 1 by an equation as I already suggested in the preliminary review.

*The visualized quantities are Gn*cos(a)+Ge*sin(a) where a is the azimuth of the satellite. We returned the equation in the description text and removed the term 'gradient values' and if necessary we use 'gradient contributions'. We hope it is clear now.*

p03/l31-p04/l08: I wonder if figure 1 is really useful. A simple comparison of mapping functions plotted according elevation will highlight the maximum values of each mf. The right part is shortly described in text, but it is not used to support any statements. Moreover, the black dots (for a single epoch, 20:30UTC) do not help to support any statements either. Maybe you could just replace this figure by a mfg comparison.

We decided to remove the dependence on elevation angle and black dots for a single epoch. We then kept the original left figures when directly comparing mapping factors (range in x-axis) and ranges of gradients (scatters in y-axis) for three gradient mapping functions. Based on this figure, we could focus on the two extreme mfg in the following part.

I agree that Fig. 1 is useful here to illustrate the variation of the delay due to the horizontal gradients and the size of the mapping factors, given that it is clarified as suggested above.

*We kept the figure after clarifying values showed in the figure.*

p06/l5-p06/l10: as you mention, gradients retrieved from NWP depends on mfg (BS or CH). Why do not use your ray-tracing algorithm to compute gradient with their closed form expression depending on NS and EW horizontal gradient of refractivity? (See Davis et al., 1993, RS)

We calculate the tropospheric gradients from ray-traced delays by least square adjustment to be as close as possible to the method applied in the GNSS analyses for parameters estimation.

You justify what you did but you don't really answer the referee's question. I think the referee suggested computing Gn and Ge from the integrals defined e.g. in Davis et al., 1993. This is another approach. Did you compare both approaches? Is there a reason why your approach is preferred?

*We calculated the gradients from ray-traced delays by least square adjustment because*

*(1) we can assume that this is the most rigorous approach, that is, this approach is as close as possible to the method applied in the GNSS parameters estimation and (2) at the time writing the manuscript we did not have the routines to calculate gradients with their closed form expression depending on NS and EW horizontal gradient of refractivity (see Davis et al., 1993, RS). We now implemented the routines to calculate gradients with their closed form expression depending on NS and EW horizontal gradient of refractivity (see Davis et al., 1993, RS). We compared NWM (ERA5) tropospheric gradients derived with the two different methods with GNSS tropospheric gradients. We find that for the considered stations (over the entire benchmark period) the root-mean square deviation between NWM and GNSS tropospheric gradients is 10 % smaller if we apply the first instead of the second method. This is in line with our expectation. Our approach is well justified. We added a short paragraph with these results to section 2.3.*

p06-p07: Did the gradient modeling affect the estimation of positions? Maybe you could complete Table2 with comparisons of position (height?) repeatability?

There were already published some studies which dealt with positioning changes related to tropospheric gradients estimation which we cite in our paper. Since our focus is only the quality of tropospheric gradients, we would rather not provide results for positioning.

I think the referee's question and suggestion are relevant. Inspecting the station repeatability is a standard approach to evaluating the quality of the processing, overall, and might thus help assess the relevance of the different gradient models tested. It would certainly be useful and interesting to check the station height repeatability. This is an additional validation to the NWP comparisons.

*We analyzed station position repeatability both in horizontal and vertical direction and we provide results on this in section 3.3 (Table 5) of updated manuscript and discuss them. As expected, the station coordinates repeatability was improved when using combined GPS+GLONASS solutions compared to GPS-only solutions, namely by a factor of 2 and 1.2 in horizontal components and the height, respectively. This corresponds to Zhou et al. (2017) who suggested 7° for the height repeatability too (solutions ranged from 3° to 15°). It should be noted that GPT+GMF models and the PPP method were used in both these studies. Contrary, Douša et al. (2017) observed a small improvement in the height even when using the elevation angle cut-off 3° (compared to 7° and 10°), however, exploiting double-difference observation, VMF1 and the Bernese Software. This discrepancy might be attributed to a slightly worse modelling of low-elevation observations, in particular due to using GPT+GMF and PPP strongly depending on all aspects of undifferenced observation modelling.*

*Concerning the impact of elevation angle cut-off on actually estimated gradients, these achieved with the 3° cut-off agreed better with NWMs, which corresponds to results reported in Douša et al. (2017).*

*Douša, J., Václavovic, P. and Eliaš, M.: Tropospheric products of the second European GNSS reprocessing (1996-2014), Atmospheric Measurement Techniques, 10, 3589–3607, doi:10.5194/amt-10-3589-2017, 2017.*

p07/Table2: I think it is important to have an overview of gradient time series in order to understand the comparisons. Especially, unlike for ZTD we do not have many ideas about gradients magnitude (maybe some ideas from figure 1): is a 0.01 mm bias significant? and a 0.76 mm stdev? These values may be put into perspectives with gradient magnitude.

We do not provide gradient time series in this paper, the reader can find them i.e. in the cited publication Li et al. (2015). However, we added a paragraph into the section 3.1 to describe typical values of gradient components under standard and severe weather conditions and we provide information on ZTD rate of change due to tropospheric gradient with an increasing distance from the

receiver. The bias of 0.01 mm is not significant in respect to typical values of gradient components and also according to estimated standard deviation of comparisons.

The publication by Li et al. (2015) that you are citing here and in the manuscript only shows time series for Onsala station (not included in your study) and for a different period. They are not directly linked with the results of you study. By the way, the second referee also asked for time series. I agree with their comments that it would be useful to give some insight into the temporal variability and size of variations, rather than just giving values for means or extreme conditions.

*We now provide a time series of tropospheric gradients for a station LDB2 (Brandenburg, Germany) for a period from May 15, 2013 till June 15, 2013 in figure 2. It shows values from two GNSS variants of solution and from NWM ERA5 solution.*

I'm not sure that the rate of change of ZTD with distance is a very relevant information as it is not directly related with a measurable quantity. Instead, you could quantify the contribution of gradient to the STD given by the 2$^{nd}$ part in Eq. (1).

*We removed the information about the rate of change of ZTD with distance, as it was not useful.*

I agree with the both referees that some indication of the significance of the numbers should be given especially since the gradient values (Gn, Ge) are very small (e.g. 0.3 mm quoted in the Abstract). Significance statements should be supported by a statistical test.

*The value of 0.3 mm in the abstract does not correspond to any mean value of gradients, but to a systematic effect observed in the manuscript solely due to different applied mfg. The value of 0.3 mm used originally indicated a maximum difference on a global scale (i.e. prevailing hydrostatic N-S gradient). We added also the value of 0.9 mm corresponding to a systematic error observed in individual gradient components observed in a local scope.*

*In Section 3.1 and Table 2, we could observe only very small biases as these are overall statistics. In Section 4 we then demonstrate actual differences which are already significant with respect to achieved SDEV. We added a discussion on a significance of impacts of applied models, cite: By using common data, period, processing strategy and software in our analysis, a significance of the impact of different models can be assessed by confronting our SDEV with those obtained when comparing gradients from different software, processing methods and even observing techniques. Generally, the SDEV values in Table 2 reach 30-50% of those obtained from comparing two different GNSS software and processing methods with two different NWM sources, and still using the same data set from the benchmark campaign (Douša et al. 2016).*

*We also realized statistical significance tests and provide their results in sections 3.1 and 3.2.*

p07/Table2: I wonder if the computation of correlation will be helpful to investigate the comparisons. A linear fit?

*We now provide linear correlation coefficients in table 2 to and table 3 and we updated the text in this regards*

Please answer the questions. Does the provision of correlations/linear fit help in the interpretation of the results? If it doesn't, you just have to mention it and not include the (unnecessary) results.

*Computation of correlation coefficients was helpful to investigate the comparisons, e.g. demonstrated a perfect correlation for ZTD and progressively reduced correlation for impacts of different models. The correlation was also high when using different mfg which underlined a comparatively larger biases in this case though still strongly averaged over all stations and period. They also clearly showed a penalty of the current RT processing and gradient estimates. The correlation coefficients are therefore included in the manuscript.*

p07/Table3: same comments as for table2. Maybe the computation of correlation or linear fit will be more relevant here.

See answer for the previous comment.

Please answer the question.

*Computation of correlation coefficients was also helpful to investigate the comparisons and these results are therefore included in the manuscript.*

I recommend the authors to improve legibility of figures (by using a better resolution)? I also recommend the use of an equation editor for mathematical expressions.

In the updated manuscript version all the equations are created using an equation editor.

What about the legibility of Figures?

For the final revised paper, I recommend that you provide the maps in uniform format.

*We updated Figure 1 and 7 to increase their legibility (numbering corresponds to new updated version of the manuscript). We kept Figure 5 in its previous form (therefore different from Figures 3 and 4), because we found that it better emphasizes differences in tropospheric gradients delivered by different observation elevation weightings. We also kept Figure 6 in its previous form, i.e. different from Fig 5 due to displaying different characteristics (gradient differences in North & East separately).*

**Remarks on answers to Referee No. 2.**

Referee comments in blue, your answers in red, my remarks and comments in black.

… My interpretation is that the present version has the form of a summary of the results, rather then what is your message to the community on how to handle tropospheric gradients. My conclusion is that it does not really matter which of the different processing option that are chosen given the data that you have studied (excluding the near real time and real time solutions, as expected). Also the small impact of adding GLONASS data may be an issue to raise for further investigations, possibly related to a higher temporal resolution of the estimated gradients.

From our point of view, we provide a recommendation to the user everywhere we think it can be given based on our own results: 1, we recommend using observations from very low elevation angles to get better gradients (this was already shown also in paper by Meindl et al. (2004) which we cite). 2, we find a small positive impact coming from adding GLONASS in our processing (it can be however different when using other products with satellite ephemerides and clock error corrections, different weighting of observations from various GNSS, etc., and some other investigations related to multi-GNSS data processing in general will follow). 3, we present the penalty in quality of tropospheric gradients from real-time processing and we show that this penalty is mainly related to the quality of used products. 4, we show that selection of gradient mapping function does not affect general quality of estimated tropospheric gradients but their magnitudes (one has to be careful then with comparing gradients from various sources due to existing systematic differences).

Please revise the Conclusion section according to the referee's comments and highlight clearly the four points you mention in your answer. Additionally, I would like to see a more insightful discussion of your results, including a comparison to other past and recent studies.

1. Revise the statements regarding the study by Meindl et al. (2004). The improvement in precision at lower elevation angles they noticed is only based on rms errors (formal errors) of the gradient parameters. This is a well-known result, explained by the use of more observations and better decorrelation of gradients from other parameters. It does not say whether they are more accurate. Accuracy of the gradient estimates can only be tested by comparison with independent data (e.g. from VLBI, MWR, NWP). Accuracy of the processing can also be tested by inspecting the coordinate repeatability, which e.g. Meindl et al. (2004) did when they compared two processing variants, with and without estimation of gradients, but not for different cutoff angles. This conclusion is actually not consistent with Zhou et al. (2017) who observed better results at 7° and 10° compared to 3°. Can you comment on this?

*You are right with Meindl et al. (2004) and we thank you for this comment. We removed the sentence with our statement from the Conclusion section and now we compare our results with this paper only in section 3.3 while analyzing station position repeatability. In the same section, we also newly discuss similarities of our results with Zhou et al. (2017) and differences with Douša et al. (2017) when it concerns a utilization of different elevation angle cut-off. We attribute differences to the GPT+GMF used in PPP for the two former while using VMF1 and double-difference solution for the latter and thus most likely a worse modelling of low-elevation observations in the two former. As better agreement between GNSS and NWM tropospheric horizontal gradients were achieved for the 3° elevation angle cut-off, and optimal modelling in Douša et al. (2017) showed even better results for height repeatability when using the VMF1, we could recommend to use the 3° cut-off for an optimal estimation of tropospheric gradients from GNSS data.*

2. Discuss the sensitivity of results to addition of GLONASS data, and compare to other recent studies (e.g. Zhou et al., 2017).

*Please see updated version of the manuscript. In the Conclusion section we now discuss our results in respect to study presented by Li et al. (2015) and Lu et al. (2016). And in the section 3.3 we now discuss the addition of GLONASS data regarding the station position repeatability and compare our results with Zhou et al. (2017).*

3. RT1 solution is slightly more accurate than the RT3 solution. Can you elaborate a bit more your comments on the quality of the used RT clock and orbit products?

*The worse result of the RT3 solution can be attributed to the worse quality and stability of the new RT2+RT3 (IGS02, IGS03) products compared to RT1 (IGS01) within the first half of 2013. Douša et al. (2018b) performed a long-term quality evaluation of all IGS RT products for the period of 2013-2017. Eventually, the RT1 and RT3 products are based on different software, combination strategy and constellations, which has been described in Section 2.2..*

*Douša, J., Václavovic, P., Zhao, L. and Kačmařík, M.: New Adaptable All-in-One Strategy for Estimating Advanced Tropospheric Parameters and Using Real-Time Orbits and Clocks, Remote Sensing, 10, 232, doi:10.3390/rs10020232, 2018b.*

4. I don't understand what you mean by the 'quality' of the estimated gradients. I would suppose you mean their accuracy, but if their magnitude is affected this means their accuracy is impacted as well. Please clarify.

*We use the general term quality of gradients to speak about their accuracy and robustness (stability in time and space). Could you please specify where exactly in the manuscript you struggle to understand the meaning of the 'quality' of the estimated gradients term?*

The conclusion that the magnitude of gradient parameter estimates is changing depending on the used mapping function is important and should be better highlighted. Given the topic of this paper, it is expected that this point is thoroughly discussed and that strong conclusions and recommendations are given and not a final remark such as "it is hard to assess which *mfg* is more suitable for the troposphere modelling in GNSS analyses".

*Our main conclusion was the assessing the impact of gradient mapping functions, which resulted mainly in systematic effects what are not critical for the estimation of other parameters (e.g. minor difference in the quality of coordinates), but mainly for the use of gradients and their evaluation or inter-comparisons between different solutions and techniques (so far the impact was usually neglected). It is really hard to guess about correct systematic effect (absolute magnitude) of gradients without a possibility to compare with independent data of the same quality at least. It seems that neither NWM, nor WVR, nor VLBI can provide comparable gradients, in particular when multi-GNSS is used. From our point of view, we sufficiently inform the reader what happens with gradient estimates if Chen and Herring or Bar-Sever gradient mapping function is applied. We show the results and discuss them in section 2.2, 3, 4 and in Appendix A and summarize them in Abstract and in Conclusion.*

The issue of time sampling of the gradient parameters should be discussed as well. Though it is not a limitation with the software you used, this question is of concern to many other scientists who use different software packages (e.g. Bernese, GAMIT) where it is more common to use a 24-h sampling.

*We agree the question is important, but also strongly related to applied troposphere modelling (deterministic vs stochastic, constraining vs. random-walk). A detail study of gradients temporal resolution for one software/strategy was presented by Zhou et al. (2017) which we mentioned in the Introduction. We consider this aspect out of the scope of this study and most likely relevant to a dedicated study consider additionally a random-walk settings. In our case, both were pre-selected and fixed (based on the prior optimizing for this purpose by using many testing variants and visual inspections of the full benchmark data set).*

Another important question is to what extent your conclusions holds during more general circumstances, because it seems as you have selected the two most extreme months for the benchmark data set. It is of course a lot of work to address this question and give a reliable answer, but it does not prevent you from an initiated discussion in the present manuscript.

We based our analyses on a data set from wet spring/summer season when the gradients could provide a valuable information for meteorological applications. Although the time period covers some severe weather events, it also contains a lot of days with standard weather conditions with tropospheric gradients close to zero. So, the results should provide a good overview on the situation in Europe during the warmer part of the year. On the other hand, we agree that new studies based on different GNSS software a data sets should be done to strengthen and confirm our results.

Please add these comments in the revised manuscript. But I don't understand your point about the necessity of new studies based on different software. Can elaborate it? Maybe it is worth discussing this point in the paper.

*We added the comment regarding selection of the Benchmark data set into introduction part of the Conclusion section. Regarding the necessity to realize other studies using different software please see our reaction on your previous comment about the time sampling of tropospheric gradients estimates.*

An overall question is that I would like to see a more critical discussion related to the numerical weather prediction models. First of all their resolution is poor, given that probably most of the large gradients occur in the atmospheric boundary layer. For example, for an elevation angle of

3° the propagation path at the height of 500 m will be approximately 10 km horizontally from the ground-based reference station. That corresponds to the resolution of the limited area model (WRF). One possibility to investigate the scale (temporal as well as spatial) of the gradients is to use the WVR data mentioned in Section 2.1. Since you mention that these data exist the reader will wonder why you do not use them for an assessment, even if the WVR data only exist at a couple of sites.

*We split our reaction into two parts:*

*1, we would be very careful about the statement that large gradients occur in the atmospheric boundary layer. Please see i.e. a paper from Elosegui et al. (1999). According to his findings GNSS tropospheric gradients are more sensitive to tropospheric features at larger heights, in different words – i.e. the same type of tropospheric feature at the height of 3 km would cause a larger gradient value than while occurring at the height of 0.5 km. And also sometimes (even during not winter season) a hydrostatic gradient can prevail in total tropospheric gradient estimated by GNSS. And these hydrostatic gradients are related to large scale (up to several hundreds of km) features, not to local station asymmetry. Of course, we are aware of limitations of NWMs we use in this study and we also state them in the paper. On the other hand, this is the first time when a NWM with a 10 km horizontal resolution was used for comparisons with GNSS results and there is a visible increase of its gradient magnitudes compared to outputs of global NWMs with 1° horizontal resolution used in Douša et al. (2016).*

*2, the usage of WVR for this study is problematic from several reasons: a) it is available only for a single station (POTS, Germany) for the benchmark campaign; b) WVR measures IWV therefore it can deliver only wet delay gradient, a hydrostatic gradient would need to be added from an external source (NWM) to get a total gradient which is delivered by GNSS; c) data quality of WVR observations at elevation angles below approximately 20-25° is generally poor (see i.e. Kačmařík et al., 2017, AMT).*

Please provide full detail of the references you are citing (Elosegui, Dousa, Kacmarik).

*Douša, J., Dick, G., Kačmařík, M., Brožková, R., Zus, F., Brenot, H., Stoycheva, A., Möller, G. and Kaplon, J.: Benchmark campaign and case study episode in central Europe for development and assessment of advanced GNSS tropospheric models and products, Atmospheric Measurement Techniques, 9, 2989–3008, doi:10.5194/amt-9-2989-2016, 2016.*

*Elosegui, P. Davis, J. L. Gradinarsky, L. P. Elgered, G. Johansson, J. M. Tahmoush and D. A. Rius, A.: Sensing atmospheric structure using small-scale space geodetic networks, Geophysical Research Letters 26, 2445-2448. doi:10.1029/1999GL900585, 1999.*

*Kačmařík, M., Douša, J., Dick, G., Zus, F., Brenot, H., Möller, G., Pottiaux, E., Kapłon, J., Hordyniec, P., Václavovic, P., and Morel, L.: Inter-technique validation of tropospheric slant total delays, Atmospheric Measurement Techniques, 10, 2183-2208, doi:10.5194/amt-10-2183-2017, 2017*

It seems that there is an interesting point of debate here on which atmospheric layer is impacting most the estimated gradient parameters. Please add some elements of this discussion in the manuscript regarding the sensitivity of tropospheric delay gradient parameters to hydrostatic and wet gradients in the atmospheric refractivity. I think this was also discussed in early papers by Davis et al., Radio Science (1993) or Chen and Herring, JGRB (1997).

Reconcile also your position with your statement in a previous publication "The first-order horizontally asymmetric delay … reflects local changes in temperature and particularly in water vapour." (*Kacmarik et al., AMT, 2017*).

*We added a short sentence into section 2.2 which notifies the reader that GNSS gradient represents a gradient of total delay, therefore a sum of dry and wet gradient. We are going to elaborate the problematics of dry/wet gradients and the relation between GNSS gradients and real weather conditions (together with NWM data quality evaluation) in our upcoming separated study.*

Regarding your 1[st] point, I know at least one study which compared GPS gradients and gradients from a NWP with a resolution of 0.1°x0.1°, and it is from 2002:

Walpersdorf, A., E. Calais, J. Haase, L. Eymard, M. Desbois and H. Vedel, Atmospheric gradients estimated by GPS compared to a high resolution numerical weather prediction (NWP) model. Physics and Chemistry of the Earth, Part A: Solid Earth and Geodesy, Vol. 26 (3), pp. 147-152, 2002.

*Thank You for this input, we apologize for neglecting this study. Now we briefly mention it in the Introduction section.*

Regarding your 2[nd] point, I am surprised of your answer because: a) A single station was used in your previous publication (Kacmarik et al., AMT, 2017) as well as in Lu et al., JGR, 2016, and Li et al. GRL, 2015; b) You solved the problem of hydrostatic gradient in Kacmarik et al., AMT, 2017, for the same COST Benchmark dataset, so it would be a strong argument to re-use these WVR data to compute gradient parameters and use them in this study as an additional source of validation; c) you suspect the quality of WVR data is poor below an elevation of angle 20-25° (even 40° is the quoted in Kacmarik et al., 2017) but according to the numerous publications which used such data, one would rather suspect that there was an issue with the data or the comparisons used in your previous study. This last point merits probably further investigation and should be commented.

Lu, C., X. Li, Z. Li, R. Heinkelmann, T. Nilsson, G. Dick, M. Ge, and H. Schuh (2016), GNSS tropospheric gradients with high temporal resolution and their effect on precise positioning, *J. Geophys. Res. Atmos.*, 121, 912–930, doi: 10.1002/2015JD024255.

*The overall lower quality of WVR measurements at low elevation angles is a generally known fact (see example i.e. in Shangguan et al., 2015). Many studies therefore avoid using low elevation data from WVR in their evaluations – apart from both mentioned Li et al. (2015) and Lu et al. (2016) for example also Pottiaux and Wamant (2002) or Davis et al. (1993). The WVR used in Li et al. (2015) and Lu et al. (2016) at the Onsala station is operated in a so-called "sky-mapping" mode therefore making observations at regular azimuth and elevation angles in repeated cycles. However, the available WVR operated at GFZ Potsdam only tracks directly GPS satellites – since only one satellite is being tracked at one moment the WVR provides much less observations than GNSS receiver. We are therefore not in favor of evaluating differences between GNSS gradients estimated at 3° or 7° or with GPS or GPS+GLONASS observations while we would have to use WVR observations at elevations above i.e. 20 ° made only in direction of GPS satellites.*

*Pottiaux, E. and Warnant, R.: First comparisons of precipitable water vapor estimation using GPS and water vapor radiometers at the Royal Observatory of Belgium, GPS Solutions, 6, 11–17, doi: 10.1007/s10291-002-0007-5, 2002.*

*Davis, J., Elgered, G., Niell, A. and Kuehn, K.: Ground-based measurement of gradients in the "wet" radio refractivity of air, Radio Science, 28, 1003-1018, 1993.*

*Shangguan, M., Heise, S., Bender, M., Dick, G., Ramatschi, M., and Wickert, J.: Validation of GPS atmospheric water vapor with WVR data in satellite tracking mode, Ann. Geophys., 33, 55–61, doi: 10.5194/angeo-33-55-2015, 2015.*

**Additional Editor comments.**

The additional comments below should help to strengthen the relevance of the manuscript for the final publication.

**General comments**

One of your strong conclusions is that changing the gradient mapping function impacts the magnitude of the estimated gradient parameters. I would surmise that this impacts also the ZTD estimates and possibly the STD estimates. Can you comment on this point? Some quantitative results would be useful (e.g. correlation analysis of ZTD/gradient/STD differences).

*Results for ZTD differences due to a change of gradient mapping function are already incorporated into the manuscript, please see section 3.1 and mainly table 2, where values of bias, SDEV and correlation coefficient are given. It is evident that the impact on ZTD values is close to zero. When STD is reconstructed, the difference in magnitude of gradient values will be compensated by difference of gradient mapping function values (mapping factors) as is apparent from figure 1 and we shortly comment on this at the end of the conclusion section.*

Compare and discuss the results from this study to similar recent studies such as Li et al., 2015; Lu et al., 2015, 2016; Zhou et al., 2017; etc. who also evaluated GPS and GNSS gradient estimates, namely in against NWP data and WVR data.

*Please see an updated version of the manuscript – section 3.3 and mainly the Conclusion section.*

Provide more insight into the significance of the gradient magnitudes and differences as the numbers reported in the manuscript are very small (< 1 mm) and may seem negligible to many readers. This should be done by using a statistical significance test when comparing two results. For instance, a 0.01 mm difference when adding GLONASS data (change of SDEV from 0.48 to 0.49 mm, quoted as 2% in the conclusions) is probably not significant but a 0.06 mm difference when changing the mapping function (from 0.48 to 0.42 mm) probably is. This would also help you to clearly (objectively) state and conclude about the results.

*We realized statistical significance tests and provide found results in sections 3.1 and 3.2 in the updated version of the manuscript. We also added a discussion on a significance of impacts of applied models based on comparing the results of this study with results reached in our previous study (Douša et al., 2016), please see section 3.1.*

Better insight into the results would be brought by changing slightly the order of presentation. Starting with present subsection 3.3 and Tables 4 and 5 would help introducing the magnitudes, directions and formal errors of gradients (answering some comments by the referees), and provide a first discussion of the sensitivity of different processing options based on statistics for GNSS results only. Time series could be included here. Then, the variants could be inter-compared (present subsection 3.1) and finally the validation using NWP data and maybe the WVR data.

*We decided not to change the order of presentation and keep it in the current format as it would significantly impact the whole manuscript at this stage.*

Finally, a major revision is required in Section 4 and in the discussion/conclusions as also asked by the referees. In its present form Section 4 appears as an additional piece of work on the impact of the observation elevation-dependent weighting, but for a single day. Instead, it should be included in the main study, as a dedicated sub-section if you wish, and treated with the same methodology (243 stations, 55 days). It is an important aspect of the sensitivity study, as you mention in the

Introduction, which merits a proper treatment. The particular day discussed could be used as a case study, in a similar way as done with Fig. 2 and 3.

*Whole section 4 went through a significant update according to comments of reviewers. However, it is still based on the results from one specific day. From our point of view, it sufficiently represents an indication of influence of the selected observation elevation-weighting scheme (and gradient mapping function) on estimated tropospheric gradients and these will not dramatically change from day to day, we believe. Applying the same methodology as you mention would require another major calculation as well as the intervention into the paper which we do not find worthy for this case.*

**Specific comments**

P1L13: include "observation elevation-dependent weighting" as one of the processing aspects that are investigated
*Corrected.*

P1L16: "a clear relation to real weather conditions" appeals for two comments: 1) such as statement would imply that you studied the results with more information and description on the weather situations (instead, I only found in the manuscript indication that "a weather front was passing over the studied area" P9L6); 2) Isn't it obvious that there is a relation with the gradient maps and weather conditions? (otherwise it would be really problematic using GNSS data for meteorology). Please be more specific. You can write that you illustrated gradient maps in the case of two weather fronts, etc.
*We agree that in the manuscript we do not provide clear evidences for our statement cited above in your comment. Therefore, we reformulated the sentence in the abstract. Regarding your point 2), from a theoretical point of view you are completely right, however based on our experience the estimation of good quality GNSS tropospheric gradients (especially in high temporal resolution) requires an optimization of the processing options and only after this optimization the gradients really shows a clear relation to weather condition. Gradients are generally more sensitive than ZTDs and they can much easier absorb other errors or modelling/adjustment problems.*

P1L18: "systematic effects…" assumes that the results are based on some average (not just one day).
*Please see abstract in the updated version of the manuscript. This part was reformulated.*

P1L20-22: the global scale results are out of scope for the paper; and the last part of the sentence is not clear (are the large local gradients the cause of differences? Difference wrt to what?)
*The global scale results are presented in the appendix A, they are part of the paper, so we do not see a problem to mention them in the abstract. Please see the abstract in the updated version of the manuscript.*

P2L7: "Dousa et al. (2018a) demonstrated the advantage of similar pseudo-observations in the 2-stage troposphere model combining optimally NWM and GNSS data". What are pseudo-observations? What do you mean by combining optimally NWM and GNSS data? Be more specific on objectives, methods and results/conclusions of the cited study like you did for Brenot et al., Morel, etc.
*Thought in the paper we used term of pseudo-ZTD observations (i.e. those calculated at a virtual station by exploiting estimated ZTD and gradients at actual station). We now simplify the sentence by writing: "… Dousa et al. (2018a) demonstrated the advantage of using tropospheric gradients in the 2-stage tropospheric model combining NWM and GNSS data". By optimally we meant (in a simplified version) combining hydrostatic and wet components from NWM and GNSS, respectively, but it can be further specified in details what is used from NWM and GNSS (ZHD, ZWD, MF, parameters for vertical corrections etc.) and including a possible weighting in a combination.*

P2L20: "systematic errors": Gradient estimates from NWP models used in this study are computed using the CH gradient mapping function (P6L8-10). This choice implies a systematic difference compared to e.g. BS gradient mapping function. For this reason, it is not possible to assess the bias, or systematic, or absolute errors of the GNSS gradient estimates. Please be careful here and throughout the manuscript about using the term "bias" and prefer systematic or mean difference instead.

*We now do not use the term bias in the manuscript at all, we replaced it with proposed terms mean difference and systematic difference.*

P3 Eq. (2) and (3), and in other equations, use 'e' for denoting the elevation angle rather than 'ele'.

*Corrected.*

P4L4: "inversely proportional" implies a 1/x variation but this is not the case (and the demonstration would require fitting a function in the data). Please reformulate.

*Whole paragraph describing the Figure 1 was reformulated in the updated version of the manuscript.*

P6L1-2: "under the assumption of a spherically layered atmosphere" from the local vertical profile?

*Yes. We now write: "Second, we compute azimuth independent STDs from the local vertical refractivity profile."*

P6L3: say here that this fitting is done using the CH mapping function in this study.

*This information is given in the following paragraph, where the question of gradient mapping function selection is being discussed. Therefore we keep the sentence in its previous form.*

P6L6: what is the "standard elevation angle dependent weighting"? Be more specific.

*For clarification we now write: "(equal versus the elevation dependent weighting of $1/sin^2(e)$)"*

P6L8-10: "comparison with GNSS gradients estimated with BS mfg should be treated cautiously" This sounds like a warning to the readers or to the users of your results. I think it is your work to interpret your results properly given this limitation. Please reformulate and add comments on the BS results where presented.

*The sentence was reformulated, we now just inform the reader in it that NWM derived tropospheric gradients were estimated using CH mfg. We now comment on this limitation in Section 3.2 where we discuss results of GNSS versus NWM comparisons.*

P6L15: is there a reason/explanation why you chose 5% as the limit?

*The number of 5% comes from a very simple calculation: a range between 3 and 7 degrees equals to 4.44 % of total range between 0 and 90 degrees. This number was rounded up to 5% to strengthen the selection of stations with a reasonable number of low elevation observations.*

P7L3: explain why you expect "negative biases"

*See the updated manuscript, whole paragraph was rewritten. Previously it was a mistake in the text, the word "negative" should not be there.*

P7L9: "The two RT solutions can be still considered of good quality if we take into consideration results found in Ahmed et al. (2016) or Kačmařík (2018)." Be more specific.

*We provide more details on this topic in the updated version of the manuscript.*

Table 2: reverse the differences in the RT results: RT1GxCH3 - GxCH3 (etc.) instead of GxCH3 - RT1GxCH3, as it is standard to use the reference data to the right.

*Corrected.*

Indicate the number of data points used in the computation of statistics.

*The number of data points entering the computation of statistics is given in the text, see sentence where Table 2 is mention in the text for the first time.*

Could the increase in standard deviation for RT3 be due to unrealistic cases such as illustrated in Fig. 3? Some data screening should be applied before computing the statistics.

*A standard screening was applied for statistics computation when values exceeding MEAN±2.5\*SDEV were excluded as outliers. However, a manual exclusion of values computed under epochs of unrealistic cases was not applied and therefore these cases could partly influence the statistics for RT3 solution.*

P8L12: "Obviously, NWMs cannot be regarded as a ground truth" Should be reformulated.

*Sentence was slightly reformulated.*

P9L6: "when a weather front was passing" use plural as they were different events

*Corrected.*

P9L11: "A detailed evaluation of tropospheric gradient maps with meteorological observations will be a subject of an upcoming study." This sentence suggests that the present study is not complete. I suggest removing it. But I suggest also to include more meteorological information on the cases illustrated in the present study, otherwise it is difficult to conclude on the relevance of GNSS gradients regarding the meteorological situations.

*The mentioned sentence was removed from the manuscript. We do not provide more information on meteorological situation on the illustrated cases in the manuscript, since both of them were already described in Douša et al. (2016) – we added information on this into the manuscript.*

P11L21: "a developed area in south-west Germany" sound awkward, please reformulate.

*Word "developed" was replaced with "urban".*

P12L3: "It suggests to reflect only the impact of differences in mapping factors on calculating formal errors." I don't understand how the mapping factors influence the formal errors.

*The mfg coefficients contribute to the design matrix and thus affects parameter variances. However, the sentence was modified in significantly rewritten paragraph.*

P13L12: "all the OEW schemes demonstrate a strong impact of low-elevation observations reflecting an actual local tropospheric asymmetry in the water vapour distribution" this sentence is a bit surprising for two reasons: 1) the OEW schemes are expected to reduce the impact of noisy low-elevation observations and 2) their impact is expected to be quite different. And second remark: how can you be sure the local tropospheric asymmetry is in the water vapour distribution?

*There is no surprise Ad 1 – the reduction of impact of more noise low-elevation estimates also affects a possibility to estimate tropospheric gradients. It is due to a significantly reduced signal (most frequently due to the non-isotropic water vapour distribution in a horizontal space) as well as due to an increased correlation of estimated gradients with other parameters. The meaning of Ad 2 is not clear to us. The sentence was reworded and shortened: "The SINEL4 weighting then shows highly reduced gradient values indicating a strong impact of the low-elevation observations on their estimates."*

P14L6-10: the description of figure 6 is not clear. To me the results in the right panel (EQUAL) look more homogeneous whereas in the left panel (SINEL2) there is a steep variation with elevation (even above 30° where it becomes more difficult to see but I guess the shape more or less varies as 1/sin(e)). Please correct.

*The main point is that the homogeneous distribution of post-fit residuals is not what we are expecting, and it seems to be unrealistic (see text below). And mainly considering any degradation close to the zenith which we noticed in other figures (stations, days). We generally consider more important the behavior of the post-fit residuals roughly above 30 degrees. At lower degrees, the behavior could be more strongly affected by various other aspects (see text below). It is difficult to show all this in a single figure, but mainly looking at different cases.*

P14L10-12: what is "a more realistic view"? This sentence is not clear and too long. Please reformulate.
*The text was modified: "The SINEL2 resulted in a distribution of the post-fit residual reflecting the expectation due to contributing errors in both GNSS observations and models. The errors generally increase with a decrease of the elevation angle, and the lowest contribution is expected at the zenith. The effects include contributions from atmospheric models, multipath, uncertainty of receiver antenna phase centre variations, lower signal-to-noise ratio and cycle slips".*

The following part of the sentence (with corrections) could be introduced at the start of the paragraph to explain what is expected. "errors in GNSS observations which are expected to increase with a decrease of elevation angle,  due to multipath effects, uncertainty of receiver antenna phase centre variations, lower signal-to-noise ratio,  or cycle slips." Obstructions do not produce errors, they simply cut off the observations.
*The modification was applied as proposed.*

P15L8: "all post-processing solutions can be regarded as robust and their gradient estimates are clearly related to real weather conditions" I didn't get how you checked the robustness of the solutions, maybe use another term or specify based on which diagnostics you can conclude that. And the relation to real weather conditions was not demonstrated, only two cases were illustrated (and not based on meteorological diagnostics).
*We removed the sentence from the manuscript.*

P15L9: "which are fully independent of meteorological input data" please clarify what you mean here. I think the GNSS processing is not completely independent of NWP data since mapping functions are derived from NWP data (even GMF).
*We reformulated the sentence for its better clarification.*

P15L10: "tropospheric gradients thus provide additional interesting information in support of NWM forecasts" forecasting is not the only application, and in the Introduction you wrote that gradients are not assimilated. Please correct.
*The word "forecasts" was removed.*

P15L11-13: Need be reformulated e.g. "Better agreement was found between GNSS gradients and NWM when the cut-off elevation angle was decreased from 7° to 3° (the standard deviation of differences decrease by 10 %), for both the single- and dual- constellation results."
*Whole paragraph was rewritten, see the updated version of the manuscript.*

P15L27-32: needs be reformulated and integrated into the main discussion.
*The paragraph was partly rewritten in the updated version of the manuscript. Unfortunately, we do not understand what do you mean that the content needs to be integrated in the main discussion – the paragraph represents a summary of individual Section 4 and forms a logical part of the Conclusion section.*

P15L32-33: the long-term mean gradient results in the Appendix are not part of the main study.

*Yes, you are right. However, we do not see a reason why we should not mention it in the Conclusion section and invite the reader to read the Appendix itself.*

P16L1-5: needs be reformulated (see comments above).
*Whole paragraph was reformulated, see the updated version of the manuscript.*

Appendix A: the discussion should be stand-alone. Instead of citing Meindl et al. (2004), I suggest that you add maps of mean GN and GE from ERA5, or maybe only GN but for CH and BS *mfg*. The figure should be labelled A1.

*As suggested we added maps of mean GN and GE from ERA5 using CH mfg. We modified the text in the Appendix accordingly.*

**Technical corrections:** see the annotated manuscript
*All proposed technical corrections were applied except cases where the manuscript undergone rewriting and therefore corrections could not be applied*

[revised manuscript text omitted]

---

## Editor Decision (ED2)

Editor comments on angeo-2018-93-manuscript-version7:

General comments

Impact of gradient mapping function (mfg): you show that using the BS mfg instead of the CH mfg makes a significant difference in the values of estimated gradients parameters (GE and GN) either by GNSS observations or from the NWM. This is indeed expected from the fact that the two MF's differ significantly at low elevations (e.g. the ratio CH/BS is 0.55 at 3 deg). This has several implications: 1) when comparing GNSS/CH and GNSS/BS variants to NWM/CH one can expect that agreement will be better for the GNSS/CH; 2) it is not possible to evaluate the absolute accuracy of GNSS gradients using NWM/CH gradients as they may contain a bias due to the particular mfg used to compute NWM/CH; 3) other approaches must be used if one wants to assess which mfg provides the more accurate GNSS gradient estimates. These remarks and warnings should be clearly stated in the manuscript (e.g. at end of Section 2). In a previous version of the manuscript you wrote that these comparisons should be treated cautiously (a sentence already revised). But I think the message should be much stronger.

The results in Table 3 show that the mean difference GNSS/BS – ERA5 is the largest which is consistent with point 1). But the standard deviation (SD) of differences for GNSS/BS – ERA5 is the smallest which contradicts point 1). The explanation given by the authors is that NWM/CH provides smaller gradient estimates than GNSS which are in better agreement with the smaller GNSS/BS gradient estimates (i.e. by chance the biases are in the same direction). I agree with the explanation but this result doesn't add anything useful to the goal of the study. I think that this comparison should thus be removed from Table 3 to keep the flow of the discussion.

Regarding point 3) inspection of position repeatability may help to assess the accuracy and compare the different processing variants (done in Section 3). You also discuss the results of another approach for the computation of gradients from the NWM gridded data based on the closed-form formulation of Davis et al. (1993). You report that this approach is in less good agreement with your GNSS/CH solution than your NWM/CH solution. Given the above remark this result was actually expected. Did you also compare the closed-form formulation results to your GNSS/BS solution? May it be that the agreement is higher? This comparison might actually help addressing points 2 and 3.

I was wondering how your gradient estimates compare to the gradient data provided by Tech. Univ. of Vienna. If I'm right their computation is based on the closed-form formulation. It would be very useful to the community if you can comment on the consistency of these various gradient data sources.

A have a final concern with Section 4. As I already expressed in the previous review and also pointed out by the two referees, the added value of Section 4 is rather poor. Though the impact of observation elevation-dependent weighting (OEW) is not negligible and should be carefully addressed, Section 4 provides only a qualitative assessment for one single day, and this assessment is based on 3 figures and 18 plots! What is the goal for each of the figures? Which additional conclusions are drawn from each of the figures? Which OEW scheme and mfg are recommended in the end? This section should be seriously revised and each of these questions should be addressed. Here are a few options for the presentation. Figure 5 could show only results for one OEW scheme and the results from the other variants could be described in the text since all the plots are actually very similar. Difference could be quantified in a Table. Figure 6: are all plots necessary? Maybe scatter plots would suffice to support the discussion and again the results from different variants could be provided in a Table. Figure 7: what does the comparison of residuals add to the assessment of gradient estimates? Additional suggestions on Section 4 are given below in the specific comments.

Specific comments

Abstract:

"All solutions using final orbit and clock products provided tropospheric gradients with a clear relation to NWM outputs" what do you mean by a "clear relation to NWM outputs"?

"The state-of-the-art models should be then applied for low-elevation observations for obtaining the best repeatability of the station coordinates" which state-of-the-art models are you referring to? Be more specific. I could not find any discussion on state-of-the-art models in the discussion of repeatability except "We also notice a slightly better performance in case of the BS mfg when compared to the CH mfg." Do you mean applying BS mfg? Or are you referring to the observation elevation-dependent weighting?

"Although using simplified models..." which simplified models are you referring to? Do you suggest that the results are only preliminary and need be confirmed using more accurate models?

"Finally, systematic errors can affect the gradient components solely due to the use of different gradient mapping functions, and still depending on the applied observation elevation-dependent weighting. A latitudinal tilting of the troposphere in a global scale causes a systematic difference up to 0.3 mm in the north gradient component, while large local gradients, usually pointing to a direction of increasing humidity, can cause differences up to 0.9 mm in any component depending on the actual direction of the gradient." I have several concerns with this paragraph. I guess the 0.3 mm is referring to the Figure in the Appendix (though I don't get how the figure can be summarized in one single number when the map shows quite large latitudinal variation) and the 0.9 mm to another result from the main text (though I could not find surely it in any table or figure). The systematic errors which are mentioned seem to refer to the results of Section 4 (according to the title of this section) but this section presents results from a single, so it is not possible to conclude on systematic errors.

Please revise the Abstract including quantitative results from Section 3 to support your conclusions (e.g. systematic differences or errors are quantified in Table 2 and 3). Nothing is said about which gradient mapping function should be used.

Section 3: move the introductory text of this section (P7L26-P8L8) including Fig. 2 to a new sub-section 2.4 "Comparison of gradient estimates". The general information belongs logically to the data and methods section as it informs about the data/station selection method, time sampling, and shows an illustration of time series which is actually not further analysed in Section 3 but is quite useful to get familiar with the magnitude of gradient parameters. This change implies very minor editing.

This new sub-section should also be completed with the screening information that you mention in your answer to my previous comments. Especially, regarding the unrealistic cases with the RT3 solution, you write that some of them may not have been detected, and this is what I suspect from the bad results reported in Table 2 and 3 for RT3. I think the detection should be improved to remove all the unrealistic cases and statistics recomputed. This should not be difficult to implement given the specific features of these cases illustrated in Fig. 4 (e.g. compute an epoch-wise correlation coefficient of RT3 vs. ERA5 and detect the values below a fixed threshold).

P7L12: "Results for individual… in Table 2" do you mean that the bias and SD in Table 2 are computed directly from the ZTD and gradient differences of all pairs of values (55 days x 243 stations x 288 estimates per day)? Another approach was followed by Dousa et al., 2016, 2017, who computed bias

and SD of differences for each station and then statistics over the ensemble of stations. Please clarify which approach was used.

P7L14: "standard deviation (SDEV) indicates a negligible impact" but later (P8L4-8) you conclude that all the differences are significant. Please be consistent.

P7L16-17: "It should be noted that GLONASS observations were down-weighted by a factor of 1.5 in dual-constellation variants of solution." This sentence should be moved in Section 2.2 and completed with an explanation (why did you down-weight GLONASS, what happens when one doesn't do it?).

P7L21 to P8L4: Isn't it quite obvious that your GNSS comparisons will be more consistent than those in Dousa et al. 2016 (I guess you refer to Table 6 in this publication) who compared GNSS to NWM estimates where GNSS estimates were computed from different software? I suggest removing this paragraph. At least it is not useful for the analysis of your results in Table 2 which don't involve NWM results.

P8: The Wilcoxon signed-rank test is designed to test the null hypothesis that data come from a distribution whose median is zero. You write that "in all cases, the differences were found to be statistically significant". Are you sure about this? It is quite surprising that the null or very small ZTD biases (0.0 to 0.2 mm) and gradient biases (0.00 and +/- 0.01 mm) reported in Table 2 are significant. Please check.

P8L6: provide a reference to the Wilcoxon signed-rank test

P7-P8 : please review and revise your comments on results from Table 2. They would gain in legibility if you discuss ZTD first and then gradients, and proceed in the order bias, SD, and CC, and row by row. Especially, the impact of changing the mfg on the bias is noticeable and consistent with what is expected from Fig. 1. So at least it would be good to start with this one.

Table 2: update signs in gradient mean differences (due to reversal of differences RT* - GxCH3 from previous version of the manuscript)

P10: significance of biases reported in Table 3: again check the results because it is suspicious that the small values are significant.

P9-11: please review and revise your comments on results from Table 3.

P11L6-10: I suggest that you remove the comparison between GNSS/BS and NWMs.

Section 3.3: this sub-section would be usefully moved at the head of section 3 as it introduces the overall characteristics of the data set and provides a first intercomparison of the impact of the various processing options and the consistency with the NWMs. This knowledge would then help in the interpretation and discussion of the results from Table 2 and 3, namely regarding the significance of the results (mean and SD of difference of gradients are very small < 1 mm). The title could be changed to "Comparison of mean gradients and formal errors).

P15L2-3: "This discrepancy might be attributed to a slightly worse modelling of low-elevation observations when using the GPT+GMF" can you provide a reference to this?

P15L4-5: "We also notice a slightly better performance in case of the BS mfg when compared to the CH mfg." Please apply the statistical test to check whether the difference is significant or not. This might have a strong implication on the conclusions since position repeatability can be regarded as an objective criterion for the assessment of the accuracy of the GNSS solution and help to select the optimal processing variant.

P15L8-9: "lower quality of the IGS03 RT product during some periods, see Figure 4." Again, the results should not be corrupted by outliers as this prevents from assessing the real accuracy of the RT3 solution. Please compute again these statistics after removing the erroneous cases.

Section 4: the title is not reflecting the content. I suggest to change to: "Additional assessment of processing options".

P16L9-10: "Magnitudes of individually estimated gradients from nearby stations show better consistency…" you suggest that a more homogeneous gradient field is of better quality? Why?

Results from Figure 5: it is not possible to decide if one of the 8 displayed gradient maps is more accurate/realistic without comparing them to a reference map and/or using an objective metrics (RMSE, etc). You can only comment on the differences and the impact of OEW and mfg settings.

P18L2-5: "Such differences depend on both the magnitude and direction of estimated gradients when these are decomposed into two components. In our case, positive differences in north and east component appear when the estimated gradients point to south and west, respectively, and negative differences occur when the gradients point to opposite directions." => this sentence could be clarified as "We have seen previously that the magnitude of CH gradients is larger compared to BS gradients. The sign of the gradient differences depends thus on the direction (north/south for GN and east/west for GE) of the CH gradients, i.e. positive differences in north and east component appear when the estimated gradients point to south and west, respectively, and negative differences occur when the gradients point to opposite directions". However, in this reasoning it is assumed that for any given pair of gradients, the magnitude of CH gradients is larger than that of BS gradients. This is not demonstrated (Fig. 1 shows the overall distribution but not the point by point relationship). Hence, a scatterplot of BS gradient vs. CH gradient should be rather shown.

P20L1-4: "The SINEL2 OEW scheme in the left panel shows more homogenous distribution of carrier-phase post-fit residuals above the elevation angle of 30° when compared to the EQUAL scheme (right panel) …" => this is not what is seen in the Figure: the EQUAL residuals are more homogeneous while the SINEL2 residuals vary roughly as 1/sin2(e) as one can expect from the applied OEW scheme. I don't think this figure adds something to the analysis of the gradient modelling schemes.

Conclusions

P20L25: reference to (Guerova et al., 2016)

Guerova, G., Jones, J., Dousa, J., Dick, G., de Haan, S., Pottiaux, E., Bock, O., Pacione, R., Elgered, G., Vedel, H., and Bender, M.: Review of the state-of-the-art and future prospects of the ground-based GNSS meteorology, Atmos. Meas. Tech., 9, 5385-5406, 2016

P21L28-29: "It affects the gradient magnitudes, not their directions, however, the gradient direction results in different projections into gradient components." Awkward sentence. Please revise.

Syntax

Replace all PP acronyms with post-processing (only a few times in the document)

Replace all SDEV acronyms with SD

[revised manuscript text omitted]

---

## Author Response (AR3)

*April 12, 2019*

*Dear editor,*
*We provide point-to-point reactions to your below given comments as well as an updated version of the manuscript.*

*Yours Sincerely,*
*Authors.*

Editor comments on angeo-2018-93-manuscript-version7:

General comments

Impact of gradient mapping function (mfg): you show that using the BS mfg instead of the CH mfg makes a significant difference in the values of estimated gradients parameters (GE and GN) either by GNSS observations or from the NWM. This is indeed expected from the fact that the two MF's differ significantly at low elevations (e.g. the ratio CH/BS is 0.55 at 3 deg). This has several implications: 1) when comparing GNSS/CH and GNSS/BS variants to NWM/CH one can expect that agreement will be better for the GNSS/CH; 2) it is not possible to evaluate the absolute accuracy of GNSS gradients using NWM/CH gradients as they may contain a bias due to the particular mfg used to compute NWM/CH; 3) other approaches must be used if one wants to assess which mfg provides the more accurate GNSS gradient estimates. These remarks and warnings should be clearly stated in the manuscript (e.g. at end of Section 2). In a previous version of the manuscript you wrote that these comparisons should be treated cautiously (a sentence already revised). But I think the message should be much stronger.
*Ad 1) we mention on this and warn the reader about the comparison of GNSS vs. NWM results in Section 3.3 – since we moved the previous section 3.3 to section 3.1, the numbering has changed! Ad 2) we now mention in the Conclusion section that the same mfg can be implemented in a different form which can also lead to biased results if tropospheric gradients from two sources are being compared and it is therefore important to check also how the particular mfg was implemented in the processing.*
*We think that the reader is now sufficiently aware of limitations of our study and of the selection of an appropriate gradient mapping function.*

The results in Table 3 show that the mean difference GNSS/BS – ERA5 is the largest which is consistent with point 1). But the standard deviation (SD) of differences for GNSS/BS – ERA5 is the smallest which contradicts point 1). The explanation given by the authors is that NWM/CH provides smaller gradient estimates than GNSS which are in better agreement with the smaller GNSS/BS gradient estimates (i.e. by chance the biases are in the same direction). I agree with the explanation but this result doesn't add anything useful to the goal of the study. I think that this comparison should thus be removed from Table 3 to keep the flow of the discussion.
*We decided to keep the GNSS/BS comparisons in the paper for the completeness. We think that we provide the reader enough information to be aware of the limitations of provided results.*

Regarding point 3) inspection of position repeatability may help to assess the accuracy and compare the different processing variants (done in Section 3). You also discuss the results of another approach for the computation of gradients from the NWM gridded data based on the closed-form formulation of Davis et al. (1993). You report that this approach is in less good agreement with your GNSS/CH solution than your NWM/CH solution. Given the above remark this result was actually expected. Did you also compare the closed-form formulation results to your GNSS/BS solution? May it be that the agreement is higher? This comparison might actually help addressing points 2 and 3.

*We do not understand "Given the above remark this result was actually expected". As stated in the manuscript, we think that the NWM tropospheric gradients obtained from the closed-form expression Davis et al. (1993) are in less good agreement with the GNSS tropospheric gradients, because the method to obtain NWM tropospheric gradients from a least square fit is closer to the method used with the GNSS. No, we did not compare the closed-form formulation results to our GNSS/BS solution. We do also not understand what you mean by point 3) above ("other approaches must be used if one wants to assess which mfg provides the more accurate GNSS gradient estimates"). What do you mean by "more accurate GNSS gradient estimates."? When we estimate tropospheric gradients by a least square fit (we can do this with the e.g. the GNSS, a NWM or a Water Vapor Radiometer) then, per definition, the tropospheric gradients depend on the gradient MF. We can define them utilizing the gradient MF by CH, but we can also define them by utilizing the gradient MF by BS. We cannot state that tropospheric gradients estimated with CH are more (or less) accurate than tropospheric gradients estimated with BS.*

I was wondering how your gradient estimates compare to the gradient data provided by Tech. Univ. of Vienna. If I'm right their computation is based on the closed-form formulation. It would be very useful to the community if you can comment on the consistency of these various gradient data sources.

*The comparison with data provided by TU Vienna is an excellent idea. Thank you for this suggestion. You are correct, the so-called LHGs (Linearized Horizontal Gradient) provided by TU Vienna are based on the closed-formulation, see Böhm et al. 2007. Recently, Landskron et al. 2018 provide refined horizontal gradients. They are based on a least square fit. They recommend using the refined horizontal gradients. It is clear that LHGs are of somewhat limited value. For example, TU Vienna recommends applying so called reduction factors. The LHG model is no longer supported. For details see [http://vmf.geo.tuwien.ac.at/](http://vmf.geo.tuwien.ac.at/). Therefore, we decided to add in the manuscript a comparison of our NWM tropospheric gradients with the LHG and the refined horizontal gradients from TU Vienna (see end of Section 2.3 and mainly the Appendix B). The LHG model is only available for several stations. We decided to look at three stations available in all data sets: ONSA, POTS and WTZR. As to expect, we find a better agreement with the refined horizontal gradient model.*

*Added references:*
*Boehm, J. and Schuh, H.: Troposphere gradients from the ECMWF in VLBI analysis, Journal of Geodesy, 81, 403-408, doi: 10.1007/s00190-007-0144-2, 2007.*

*Landskron, D. and Boehm, J.: Refined discrete and empirical horizontal gradients in VLBI analysis, Journal of Geodesy, 92, 1387-1399, doi:10.1007/s00190-018-1127-1, 2018.*

*Zus, F., Douša, J., Dick, G., and Wickert, J.: Station specific NWM based tropo parameters for the Benchmark campaign, ES1206-GNSS4WEC COST Workshop, Iceland, 8–10 March 2016.*

A have a final concern with Section 4. As I already expressed in the previous review and also pointed out by the two referees, the added value of Section 4 is rather poor. Though the impact of observation elevation-dependent weighting (OEW) is not negligible and should be carefully addressed, Section 4 provides only a qualitative assessment for one single day, and this assessment is based on 3 figures and 18 plots! What is the goal for each of the figures? Which additional conclusions are drawn from each of the figures? Which OEW scheme and mfg are recommended in the end? This section should be seriously revised and each of these questions should be addressed. Here are a few options for the presentation. Figure 5 could show only results for one OEW scheme and the results from the other variants could be described in the text since all the plots are actually very similar. Difference could be quantified in a Table. Figure 6: are all plots necessary? Maybe scatter plots would suffice to support the discussion and again the results from different variants

could be provided in a Table. Figure 7: what does the comparison of residuals add to the assessment of gradient estimates? Additional suggestions on Section 4 are given below in the specific comments.

*Initially, we included only plots for two OEW schemes for gradients and differences. During the first review, we were asked why we haven't shown other weighting too. We then added them because a graphical visualization gives a clear view about behaviour of gradients (the size and orientation) at individual stations of the network showing patterns and impact due to both the gradient mapping function and the OEW scheme. The impact can be also study during events with significant gradients in a dense network only while it easily remains hidden in most other cases. That was our preference over a pure statistics or time series which can hardly show any details or overall behaviour.*

*In this context, we should also emphasize that our target was to estimate optimally tropospheric horizontal gradients during a severe weather event when the results could support numerical weather forecasting. Obviously, it doesn't necessarily have the same needs for estimation of gradients within the re-analysis. The reason is they are commonly included with a low temporal resolution and/or constrained for decorrelations. Then gradients are used mainly for improving the modelling of the troposphere for improving other parameters, mainly horizontal coordinates. So far in re-analysis, there were no aims at targeting optimal estimates of size and orientation of gradients.*

*Anyway, we fully revised Section 4.*

Specific comments

Abstract:
"All solutions using final orbit and clock products provided tropospheric gradients with a clear relation to NWM outputs" what do you mean by a "clear relation to NWM outputs"?
*We reformulated the sentence.*

"The state-of-the-art models should be then applied for low-elevation observations for obtaining the best repeatability of the station coordinates" which state-of-the-art models are you referring to? Be more specific. I could not find any discussion on state-of-the-art models in the discussion of repeatability except "We also notice a slightly better performance in case of the BS mfg when compared to the CH mfg." Do you mean applying BS mfg? Or are you referring to the observation elevation-dependent weighting?
*We reworded the whole sentence: "Comparisons of GNSS and NWM gradients suggest the 3° elevation angle cut-off and GPS+GLONASS constellation for obtaining optimal gradient estimates provided precise models for antenna phase centre offsets and variations and tropospheric mapping functions are applied for low-elevation observations."*

"Although using simplified models..." which simplified models are you referring to? Do you suggest that the results are only preliminary and need be confirmed using more accurate models?
*We removed the first part of the sentence – see above.*

"Finally, systematic errors can affect the gradient components solely due to the use of different gradient mapping functions, and still depending on the applied observation elevation-dependent weighting. A latitudinal tilting of the troposphere in a global scale causes a systematic difference up to 0.3 mm in the north gradient component, while large local gradients, usually pointing to a direction of increasing humidity, can cause differences up to 0.9 mm in any component depending on the actual direction of the gradient." I have several concerns with this paragraph. I guess the 0.3 mm is referring to the Figure in the Appendix (though I don't get how the figure can be summarized in one single number when the map shows quite large latitudinal variation) and the 0.9 mm to another result from the main text (though I could not find surely it in any table or figure). The systematic errors which are mentioned seem to refer to the results of Section 4 (according to the title of this

section) but this section presents results from a single, so it is not possible to conclude on systematic errors.

*We believe providing these numbers in abstract is relevant when these are purely referred to the extreme values – in both cases we thus used the words "can cause differences up to". Of course, an actual impact can be much smaller or even negligible, but we aimed at pointing to these potential differences which were commonly neglected in the community until today – either within gradient combinations, comparisons or interpretations.*

*From the global map (Appendix), maximum values can be clearly identified, and these are rather stable over time. For the regional maximum values, we identified the event with the most significant tropospheric gradients in order to demonstrate what can happen during a short period of up to several hours, i.e. in an actual situation when numerical weather forecasting can profit from accurate and high-resolution GNSS tropospheric horizontal gradients.*

Please revise the Abstract including quantitative results from Section 3 to support your conclusions (e.g. systematic differences or errors are quantified in Table 2 and 3). Nothing is said about which gradient mapping function should be used.

*We added a sentence regarding the gradient mapping function selection in the abstract. On the other hand, we prefer to keep the abstract in the current shape to keep it compact and easily understandable. The quantitative results are summarized in the Conclusion section.*

Section 3: move the introductory text of this section (P7L26-P8L8) including Fig. 2 to a new subsection 2.4 "Comparison of gradient estimates". The general information belongs logically to the data and methods section as it informs about the data/station selection method, time sampling, and shows an illustration of time series which is actually not further analysed in Section 3 but is quite useful to get familiar with the magnitude of gradient parameters. This change implies very minor editing.

*Section 2.4 was created.*

This new sub-section should also be completed with the screening information that you mention in your answer to my previous comments. Especially, regarding the unrealistic cases with the RT3 solution, you write that some of them may not have been detected, and this is what I suspect from the bad results reported in Table 2 and 3 for RT3. I think the detection should be improved to remove all the unrealistic cases and statistics recomputed. This should not be difficult to implement given the specific features of these cases illustrated in Fig. 4 (e.g. compute an epoch-wise correlation coefficient of RT3 vs. ERA5 and detect the values below a fixed threshold).

*Unrealistic cases with the RT3 solution were detected, the statistics were re-computed and updated in Table 2 and Table 3. Description of applied screening was added into the Section 2.4.*

P7L12: "Results for individual… in Table 2" do you mean that the bias and SD in Table 2 are computed directly from the ZTD and gradient differences of all pairs of values (55 days x 243 stations x 288 estimates per day)? Another approach was followed by Dousa et al., 2016, 2017, who computed bias and SD of differences for each station and then statistics over the ensemble of stations. Please clarify which approach was used.

*Yes, we used the first approach which you describe – statistics were computed directly from the ZTD and gradient differences of all pairs of values (55 days x 243 stations x 288 estimates per day).*

P7L14: "standard deviation (SDEV) indicates a negligible impact" but later (P8L4-8) you conclude that all the differences are significant. Please be consistent.

*The sentence was rewritten.*

P7L16-17: "It should be noted that GLONASS observations were down-weighted by a factor of 1.5 in

dual-constellation variants of solution." This sentence should be moved in Section 2.2 and completed with an explanation (why did you down-weight GLONASS, what happens when one doesn't do it?).
*We added the reasoning: "It should be noted that GLONASS observations were down-weighted by a factor of 1.5 in dual-constellation variants of solution to reflect both a lower quality of precise products and observations."*

P7L21 to P8L4: Isn't it quite obvious that your GNSS comparisons will be more consistent than those in Dousa et al. 2016 (I guess you refer to Table 6 in this publication) who compared GNSS to NWM estimates where GNSS estimates were computed from different software? I suggest removing this paragraph. At least it is not useful for the analysis of your results in Table 2 which don't involve NWM results.
*We removed this part of the text.*

P8: The Wilcoxon signed-rank test is designed to test the null hypothesis that data come from a distribution whose median is zero. You write that "in all cases, the differences were found to be statistically significant". Are you sure about this? It is quite surprising that the null or very small ZTD biases (0.0 to 0.2 mm) and gradient biases (0.00 and +/- 0.01 mm) reported in Table 2 are significant. Please check.
*We were also surprised by the results, however we checked them carefully and they are correct. To confirm them, we extra tested ZTD values from the pair GRCH3-GRBS3 in the IBM SPSS Statistics software and we got the same results. We provide graphical outputs from the SPSS testing here (solution1 = GRCH3, solution2 = GRBS3):*

| | Null Hypothesis | Test | Sig. | Decision |
|---|---|---|---|---|
| **Hypothesis Test Summary** | | | | |
| 1 | The distribution of solution1 is normal with mean 2,411.801 and standard deviation 39.538. | One-Sample Kolmogorov-Smirnov Test | .000 | Reject the null hypothesis. |
| 2 | The distribution of solution2 is normal with mean 2,411.881 and standard deviation 39.586. | One-Sample Kolmogorov-Smirnov Test | .000 | Reject the null hypothesis. |
| Asymptotic significances are displayed. The significance level is .05. | | | | |

**Hypothesis Test Summary**

| | Null Hypothesis | Test | Sig. | Decision |
|---|---|---|---|---|
| 1 | The median of differences between solution1 and solution2 equals 0. | Related-Samples Sign Test | ,000 | Reject the null hypothesis. |
| 2 | The median of differences between solution1 and solution2 equals 0. | Related-Samples Wilcoxon Signed Rank Test | ,000 | Reject the null hypothesis. |
| 3 | The distributions of solution1 and solution2 are the same. | Related-Samples Friedman's Two-Way Analysis of Variance by Ranks | ,000 | Reject the null hypothesis. |
| 4 | The distributions of solution1 and solution2 are the same. | Related-Samples Kendall's Coefficient of Concordance | ,000 | Reject the null hypothesis. |

Asymptotic significances are displayed. The significance level is ,05.

P8L6: provide a reference to the Wilcoxon signed-rank test
*Reference added: https://docs.scipy.org/doc/scipy/reference/generated/scipy.stats.wilcoxon.html.*

P7-P8: please review and revise your comments on results from Table 2. They would gain in legibility if you discuss ZTD first and then gradients, and proceed in the order bias, SD, and CC, and row by row. Especially, the impact of changing the mfg on the bias is noticeable and consistent with what is expected from Fig. 1. So at least it would be good to start with this one.
*Whole text discussing results from Table 2 was edited in order to increase its legibility. Although we do not discuss firstly ZTD and then gradients as you suggested, we keep the order of the parameters – mean difference, SD, CC.*

Table 2: update signs in gradient mean differences (due to reversal of differences RT* - GxCH3 from previous version of the manuscript)
*Corrected, thank you for notification.*

P10: significance of biases reported in Table 3: again check the results because it is suspicious that the small values are significant.
*We checked the results, they are correct.*

P9-11: please review and revise your comments on results from Table 3.
*We slightly edited the text describing results given in Table 3.*

P11L6-10: I suggest that you remove the comparison between GNSS/BS and NWMs.
*We decided to keep them in the paper for completeness.*

Section 3.3: this sub-section would be usefully moved at the head of section 3 as it introduces the overall characteristics of the data set and provides a first intercomparison of the impact of the various processing options and the consistency with the NWMs. This knowledge would then help in

the interpretation and discussion of the results from Table 2 and 3, namely regarding the significance of the results (mean and SD of difference of gradients are very small < 1 mm). The title could be changed to "Comparison of mean gradients and formal errors).
*We moved this sub-section to the beginning of Section 3.*

P15L2-3: "This discrepancy might be attributed to a slightly worse modelling of low-elevation observations when using the GPT+GMF" can you provide a reference to this?
*Douša et al. (2017) indicated also worse results when using GPT+GMF compared to VMF1 which can be attributed to modelling errors in the former particularly if applied in PPP (Kouba 2009).*

*Douša, J., Václavovic, P. and Eliaš, M.: Tropospheric products of the second European GNSS reprocessing (1996-2014), Atmospheric Measurement Techniques, 10, 3589–3607, doi:10.5194/amt-10-3589-2017, 2017.*

*Kouba, J.: Testing of global pressure/temperature (GPT) model and global mapping function (GMF) in GPS analyses, Journal of Geodesy, 83, 199–208, doi:10.1007/s00190-008-0229-6, 2009.*

P15L4-5: "We also notice a slightly better performance in case of the BS mfg when compared to the CH mfg." Please apply the statistical test to check whether the difference is significant or not. This might have a strong implication on the conclusions since position repeatability can be regarded as an objective criterion for the assessment of the accuracy of the GNSS solution and help to select the optimal processing variant.
*We firstly tested the normality of coordinates repeatability in all three components for GRCH3 and GRBS3 solutions and found out that they do not follow a normal distribution. Therefore, we applied the Wilcoxon signed-rank test and according to its results the difference is statistically significant in North and Up component at the 5% significance level. We added this information to the manuscript.*

P15L8-9: "lower quality of the IGS03 RT product during some periods, see Figure 4." Again, the results should not be corrupted by outliers as this prevents from assessing the real accuracy of the RT3 solution. Please compute again these statistics after removing the erroneous cases.
*Tropospheric gradients were during the GNSS data processing estimated epoch-wisely while the coordinates were estimated on a daily basis. Since epochs with unrealistic tropospheric gradients in the RT3GxCH3 solution were identified in 28 days, we would have to exclude these 28 days from the coordinates repeatability computation. Due to this reason we keep the original results in the paper.*

*On the other hand, we re-computed all other results in tables 4 and 5 (formal errors of tropospheric gradients, mean gradient angles and magnitudes) and updated the manuscript accordingly.*

Section 4: the title is not reflecting the content. I suggest to change to: "Additional assessment of processing options".
*Changed to: "Systematic effects induced by gradient mapping functions and elevation-dependent weighting"*

P16L9-10: "Magnitudes of individually estimated gradients from nearby stations show better consistency…" you suggest that a more homogeneous gradient field is of better quality? Why?
*Considering the gradients estimated in PPP are not affected by the errors stemming from the precise products, these should reflect actual weather conditions. Fortunately, the PPP is stand-alone method (data processed independently at each station), thus gradients estimated from two nearby stations should reflect similar conditions and thus similar values can be considered highly realistic. And even in opposite way, the gradients should not indicate significantly different magnitudes or directions for such stations or within local variabilities or patterns in a dense network.*

Results from Figure 5: it is not possible to decide if one of the 8 displayed gradient maps is more accurate/realistic without comparing them to a reference map and/or using an objective metrics (RMSE, etc). You can only comment on the differences and the impact of OEW and mfg settings.

*We agree, but we haven't tried this. However, as we fully revised the section, we believe there is no any such statement.*

P18L2-5: "Such differences depend on both the magnitude and direction of estimated gradients when these are decomposed into two components. In our case, positive differences in north and east component appear when the estimated gradients point to south and west, respectively, and negative differences occur when the gradients point to opposite directions." => this sentence could be clarified as "We have seen previously that the magnitude of CH gradients is larger compared to BS gradients. The sign of the gradient differences depends thus on the direction (north/south for GN and east/west for GE) of the CH gradients, i.e. positive differences in north and east component appear when the estimated gradients point to south and west, respectively, and negative differences occur when the gradients point to opposite directions". However, in this reasoning it is assumed that for any given pair of gradients, the magnitude of CH gradients is larger than that of BS gradients. This is not demonstrated (Fig. 1 shows the overall distribution but not the point by point relationship). Hence, a scatterplot of BS gradient vs. CH gradient should be rather shown.

*The sentence was clarified as suggested. The higher magnitudes of CH gradients were clearly visible in Figure 5 (now in Figure 6) and it was discussed in Section 2.2 as it is a product of the gradient term in Eq. 1.*
*We do believe it does not need a scatter plot (thought we added them), which does not give clearer picture.*

P20L1-4: "The SINEL2 OEW scheme in the left panel shows more homogenous distribution of carrierphase post-fit residuals above the elevation angle of 30° when compared to the EQUAL scheme (right panel) …" => this is not what is seen in the Figure: the EQUAL residuals are more homogeneous while the SINEL2 residuals vary roughly as $1/\sin^2(e)$ as one can expect from the applied OEW scheme. I don't think this figure adds something to the analysis of the gradient modelling schemes.

*The paragraph was fully reworded. Anyway, we assessed the homogeneity of post-fit residuals mainly for high elevations (above 30deg at least) and close to the zenith, where any contributing errors should be the smallest.*

Conclusions

P20L25: reference to (Guerova et al., 2016)
Guerova, G., Jones, J., Dousa, J., Dick, G., de Haan, S., Pottiaux, E., Bock, O., Pacione, R., Elgered, G., Vedel, H., and Bender, M.: Review of the state-of-the-art and future prospects of the ground-based GNSS meteorology, Atmos. Meas. Tech., 9, 5385-5406, 2016
*Reference added.*

P21L28-29: "It affects the gradient magnitudes, not their directions, however, the gradient direction results in different projections into gradient components." Awkward sentence. Please revise.
*Reworded: "While the mfg choice affects the magnitude of estimated gradient, it does not affect the direction of the gradient. However, any difference in the magnitude causes systematic errors in gradient components which depend on the gradient direction too."*

Syntax
Replace all PP acronyms with post-processing (only a few times in the document)
*Done.*

Replace all SDEV acronyms with SD

*Done.*

[revised manuscript text omitted]

---

## Editor Decision (ED3)

Editor comments on angeo-2018-93-manuscript-version8

Comments on yours answers (in green) to my previous review

*We do not understand "Given the above remark this result was actually expected".*

One can expect that the two gradient computation methods using NWM data would agree better than one of the NWM gradients agrees with GNSS gradients, because both use NWM data and NWM fields are smoother than reality which is leading to small gradient estimates.

*We cannot state that tropospheric gradients estimated with CH are more (or less) accurate than tropospheric gradients estimated with BS.*

This situation is annoying if one thinks about using GNSS gradients for data assimilation or climate monitoring. For comparison, consider the ZTD estimates. A lot of efforts have been made to achieve accurate mapping functions such that the ZTD estimates have now negligible biases and you don't need to know which mapping function was used when you want to assimilate GNSS ZTD data. It should be emphasized that more work is still necessary to improve the gradient mapping functions.

*The impact can be also study during events with significant gradients in a dense network only while it easily remains hidden in most other cases.*

Good point. You can mention this when you study the case of 31 May 2013 in Section 4.

*From the global map (Appendix), maximum values can be clearly identified, and these are rather stable over time.*

The maximum values cannot be se clearly identified because of the continuous colour shades. Could mention in which region(s) the maximum value is observed? It would also be useful to report the value of the bias in the benchmark region.

The figure shows a mean map, so it cannot be speculated if the results are stable or not in time. Actually the time series shown in Figure 11 show quite large variability so the stability in time is disputable.

*Unrealistic cases with the RT3 solution were detected, the statistics were re-computed and updated in Table 2 and Table 3.*

I noticed that results for RT1 also changed in these Tables. What is the reason for this?

*Yes, we used the first approach which you describe – statistics were computed directly from the ZTD and gradient differences of all pairs of values (55 days x 243 stations x 288 estimates per day).*

Please mention it in the manuscript as it implies that the results in the Tables are representative of all stations mixed together (i.e. region-average statistics) rather than statistical for a "typical" station (i.e. station-mean statistics) such as in Dousa et al., 2017.

*We do believe it does not need a scatter plot (thought we added them), which does not give clearer picture.*

What you added actually are histograms and not scatter plots (e.g. GN/CH vs. GN/BS plots).

Minor corrections on the new manuscript (angeo-2018-93-manuscript-version8)

P2L1: Numerical Weather Prediction models (NWM) => Numerical Weather Models (NWM) or Numerical Weather Prediction (NWP) models ; change consistently throughout the manuscript is the latter is kept

P3L20: From the formula (1) is evident that GNSS gradient represents a gradient of both hydrostatic and wet part of the delay, therefore a total delay gradient. => The GNSS gradient modelled by Eq. (1) represents a total gradient (the hydrostatic and wet components are not explicit in this formulation).

P4L13: We can thus further focus on BS and CH mfg only… => In the following we focus on BS and CH mfg only…

P9L22: Naturally, smaller formal errors correspond to the lower elevation angle cut-off which can be observed for both ZTDs and tropospheric gradients in Table 3. => This can be observed in Table 3 when the elevation cutoff is increased.

P10L19-20: The gradients estimated with improved geometry and using more observations are expected to provide more accurate and reliable estimates. => The gradients estimated with improved geometry and using more observations are expected to be more accurate and reliable.

P10L24: "The ZTDs were thus practically unaffected by different gradient models." Remove this sentence. The correlation coefficients of 1.000 for the ZTD estimates is very likely biased because they are computed from all stations mixed together (a well know artefact when the mean values are different from one station to another).

P11: number of section is 3.3 not 3.2

P12L10: the reference to Appendix A is not relevant, unless you indicate the value of the bias in the Benchmark region.

P15: title of section 4: "Systematic effects" is misleading as mainly one day (initially, and now 8 days) of the Benchmark period and region are studied. Suggest to change to "Impact of different gradient mapping functions and elevation-dependent weighting".

P16L1-2: I think both plots show results close to the expected behaviour: smaller residuals near the zenith and larger at low elevations. SINEL2 is preferred not because of the residual properties but because more accurate parameters are estimated (ZTD, coordinates, etc.).

P16L5: do you have a reference for your previous finding? If not, summarise the results or remove this sentence.

P16L8: Thought => though

---

## Author Response (AR4)

*May 9, 2019*

*Dear editor,*
*We provide point-to-point reactions (in red) to your below given comments as well as an updated version of the manuscript.*

*Yours Sincerely,*
*Authors.*

Editor comments on angeo-2018-93-manuscript-version8

Comments on yours answers (in green) to my previous review

*We do not understand "Given the above remark this result was actually expected".*

One can expect that the two gradient computation methods using NWM data would agree better than one of the NWM gradients agrees with GNSS gradients, because both use NWM data and NWM fields are smoother than reality which is leading to small gradient estimates.

*We agree. We misunderstood your comment. What we meant is that the NWM gradients derived by least square fitting better fit to the GNSS gradients than the NWM gradients derived with the closed form expression.*

*We cannot state that tropospheric gradients estimated with CH are more (or less) accurate than tropospheric gradients estimated with BS.*

This situation is annoying if one thinks about using GNSS gradients for data assimilation or climate monitoring. For comparison, consider the ZTD estimates. A lot of efforts have been made to achieve accurate mapping functions such that the ZTD estimates have now negligible biases and you don't need to know which mapping function was used when you want to assimilate GNSS ZTD data. It should be emphasized that more work is still necessary to improve the gradient mapping functions.

*The impact of mapping functions on ZHD (ZWD) and gradients is not equal or fully comparable. The magnitude of gradient is more sensitive to MF, compared to the ZHD (ZWD), because it is estimated from all the satellites (observations) and directly scaled with the actual gradient MF. In the case of ZHD (ZWD), the MF affects the estimated zenith delay depending on elevations of individual satellites (observations). Improving gradient MF will only be possible if high-resolution and high-accurate NWM data sets are available. Optimal selection will be available only if we have another high-accuracy and independent observations for the gradients.*

*We added an extra sentence to the Conclusion section to more emphasize the necessity to deal more with an optimal gradient mapping function finding.*

*The impact can be also study during events with significant gradients in a dense network only while it easily remains hidden in most other cases.*

Good point. You can mention this when you study the case of 31 May 2013 in Section 4.

*We added one sentence to Section 4 (P16L16).*

*From the global map (Appendix), maximum values can be clearly identified, and these are rather stable over time.*

The maximum values cannot be se clearly identified because of the continuous colour shades. Could mention in which region(s) the maximum value is observed? It would also be useful to report the value of the bias in the benchmark region.

The figure shows a mean map, so it cannot be speculated if the results are stable or not in time. Actually the time series shown in Figure 11 show quite large variability so the stability in time is disputable.

*If we exclude oceans, the maximum values can be found in north-east America and north-east Asia. In the Benchmark region, the correct number is 0.15 mm. We added this information to Appendix A in the manuscript.*

*With regards to the stability we agree that we do not analyse stability in the manuscript and therefore we remove the statement "...and rather stable over time" from our last reply to your comment.*

*Unrealistic cases with the RT3 solution were detected, the statistics were re-computed and updated in Table 2 and Table 3.*

I noticed that results for RT1 also changed in these Tables. What is the reason for this?

*Time periods with unrealistic cases influencing strongly the quality of tropospheric gradients from RT3 were fully excluded from statistical evaluation. This step partly influenced also RT1 results.*

*Yes, we used the first approach which you describe – statistics were computed directly from the ZTD and gradient differences of all pairs of values (55 days x 243 stations x 288 estimates per day).*

Please mention it in the manuscript as it implies that the results in the Tables are representative of all stations mixed together (i.e. region-average statistics) rather than statistical for a "typical" station (i.e. station-mean statistics) such as in Dousa et al., 2017.

*Now mentioned in section 2.4.*

*We do believe it does not need a scatter plot (thought we added them), which does not give clearer picture.*

What you added actually are histograms and not scatter plots (e.g. GN/CH vs. GN/BS plots).

*You are right, we used histograms. We apologize for mixing these two terms in our answer and in the manuscript itself. From our point of view the used histograms served better for the needed purpose.*

Minor corrections on the new manuscript (angeo-2018-93-manuscript-version8)

P2L1: Numerical Weather Prediction models (NWM) => Numerical Weather Models (NWM) or Numerical Weather Prediction (NWP) models; change consistently throughout the manuscript is the latter is kept

*Numerical Weather models (NWM) is now used consistently.*

P3L20: From the formula (1) is evident that GNSS gradient represents a gradient of both hydrostatic and wet part of the delay, therefore a total delay gradient. => The GNSS gradient modelled by Eq. (1) represents a total gradient (the hydrostatic and wet components are not explicit in this formulation).

*Replaced with suggested sentence.*

P4L13: We can thus further focus on BS and CH mfg only… => In the following we focus on BS and CH mfg only…

*Replaced with suggested version.*

P9L22: Naturally, smaller formal errors correspond to the lower elevation angle cut-off which can be observed for both ZTDs and tropospheric gradients in Table 3. => This can be observed in Table 3 when the elevation cutoff is increased.

*Replaced with suggested sentence.*

P10L19-20: The gradients estimated with improved geometry and using more observations are expected to provide more accurate and reliable estimates. => The gradients estimated with improved geometry and using more observations are expected to be more accurate and reliable.

*Replaced with suggested sentence.*

P10L24: "The ZTDs were thus practically unaffected by different gradient models." Remove this sentence. The correlation coefficients of 1.000 for the ZTD estimates is very likely biased because they are computed from all stations mixed together (a well know artefact when the mean values are different from one station to another).

*Sentence removed.*

P11: number of section is 3.3 not 3.2

*Corrected.*

P12L10: the reference to Appendix A is not relevant, unless you indicate the value of the bias in the Benchmark region.

*Reference to Appendix A removed.*

P15: title of section 4: "Systematic effects" is misleading as mainly one day (initially, and now 8 days) of the Benchmark period and region are studied. Suggest to change to "Impact of different gradient mapping functions and elevation-dependent weighting".

*Replaced with suggested version.*

P16L1-2: I think both plots show results close to the expected behaviour: smaller residuals near the zenith and larger at low elevations. SINEL2 is preferred not because of the residual properties but because more accurate parameters are estimated (ZTD, coordinates, etc.).

*You are right, but still there are differences in the distribution of residuals as we describe them in the manuscript. We don't see any reason for degradation close to the zenith. Therefore, we keep the text in its previous version.*

P16L5: do you have a reference for your previous finding? If not, summarise the results or remove this sentence.

*We have no reference, the sentence was removed.*

P16L8: Thought => though

*Corrected.*

[revised manuscript text omitted]

---

## Editor Decision (ED4)

Editor comments on angeo-2018-93-manuscript-version8

My comments (in black) to yours answers (in green) to my previous review

*The impact of mapping functions on ZHD (ZWD) and gradients is not equal or fully comparable. The magnitude of gradient is more sensitive to MF, compared to the ZHD (ZWD), because it is estimated from all the satellites (observations) and directly scaled with the actual gradient MF. In the case of ZHD (ZWD), the MF affects the estimated zenith delay depending on elevations of individual satellites (observations). Improving gradient MF will only be possible if high-resolution and high-accurate NWM data sets are available. Optimal selection will be available only if we have another high-accuracy and independent observations for the gradients.*

I don't understand your argumentation. Both types of parameters (ZWD and gradients) are estimated from all the satellites and are directly scaled by their MF, see your Eq. (1). I don't see how the sensitivity of parameters can be compared directly (which one is more/less sensitive) without a quantitative assessment. Do you have any results, documentation, or reference of this? Please use in the manuscript.

*Time periods with unrealistic cases influencing strongly the quality of tropospheric gradients from RT3 were fully excluded from statistical evaluation. This step partly influenced also RT1 results.*

Can you be more specific and explain the detection procedure used? Is it the same for both RT streams? How many cases were excluded from the RT1 and RT3 datasets? This information may be useful to other users of these products. => add a short explanation and numbers at the end of Section 2.4 when you mention "data screening".

*Now mentioned in section 2.4.*

Add to the text inserted P7L17 the numbers of values used or compared (55 days x 243 stations x 288 estimates per day ~ 3.4x10^6 values). Since this applies to all three Tables, the information can be removed from section 3.2.

*P16L1-2: I think both plots show results close to the expected behaviour: smaller residuals near the zenith and larger at low elevations. SINEL2 is preferred not because of the residual properties but because more accurate parameters are estimated (ZTD, coordinates, etc.).*

*You are right, but still there are differences in the distribution of residuals as we describe them in the manuscript. We don't see any reason for degradation close to the zenith. Therefore, we keep the text in its previous version.*

There is no degradation at the zenith, just a distribution of residuals consistent with the observation weighting model (more uniform). The sentence could be revised as: "It is closer to the expected behaviour…" => "Both show the expected behaviour…"

Comments on manuscript version 9:

P20L14: "Since no meteorological data providing any information about prevailing atmospheric conditions during the evaluated time period entered the GNSS data processing," This is not generally true, e.g. when VMF1 and ECMWF a priori ZHD/ZWD data are used. In your case GMF and GPT and used, so it is important in the conclusion section to be more specific about what is a general result and what is specific. => add "(because we used empirical mapping functions and a priori tropospheric delays)"

Add the missing sections at the end of the manuscript (see https://www.annales-geophysicae.net/for_authors/manuscript_preparation.html):

• Code availability

• Data availability

• Sample availability

• Team list

• Author contribution

• Competing interests

See especially the Data policy of the journal:

https://www.annales-geophysicae.net/about/data_policy.html

---

## Author Response (AR5)

*May 31, 2019*

*Dear editor,*
*We provide point-to-point reactions (in red) to your below given comments together with an updated version of the manuscript.*

*Yours Sincerely,*
*Authors.*

**Editor comments on angeo-2018-93-manuscript-version8**

My comments (in black) to yours answers (in green) to my previous review

*The impact of mapping functions on ZHD (ZWD) and gradients is not equal or fully comparable. The magnitude of gradient is more sensitive to MF, compared to the ZHD (ZWD), because it is estimated from all the satellites (observations) and directly scaled with the actual gradient MF. In the case of ZHD (ZWD), the MF affects the estimated zenith delay depending on elevations of individual satellites (observations). Improving gradient MF will only be possible if high-resolution and high-accurate NWM data sets are available. Optimal selection will be available only if we have another high-accuracy and independent observations for the gradients.*

I don't understand your argumentation. Both types of parameters (ZWD and gradients) are estimated from all the satellites and are directly scaled by their MF, see your Eq. (1). I don't see how the sensitivity of parameters can be compared directly (which one is more/less sensitive) without a quantitative assessment. Do you have any results, documentation, or reference of this? Please use in the manuscript.

*We aimed to explain the fact obvious from eq. 2 where the approximation of gradient mapping function is expressed combining $mf_w$ and cot(e) functions. Assuming that the error in $mf_g$ is equivalent to those in $mf_w$ (or is coming from $mf_w$), the impact on scaling gradients depends mainly on cot(e) function which increases for low-elevation angles. The gradients are practically estimated using low-elevation observations (representing the 2nd order tropospheric effect) the impact on scaling significantly grows compared to ZTD. To clarify this point, we added (optionally) a few sentences in the manuscript:*

*"Impacts of mapping functions on estimated ZHD (ZWD) and gradient parameters are different, though both represent kind of an elevation-dependent parameter scaling. The latter is more sensitive to the mapping function compared to the former, additionally considering their relative magnitudes. The gradient mapping function is strongly driven by the cot(e) approximation (Eq. 2) which is growing fast for low elevation angles. Because gradients represent the 2nd order effect of the tropospheric delay, fast growing with the distance from the station, they are practically estimated using low-elevation observations and, consequently, the impact of mfg becomes significant."*

*Time periods with unrealistic cases influencing strongly the quality of tropospheric gradients from RT3 were fully excluded from statistical evaluation. This step partly influenced also RT1 results.*

Can you be more specific and explain the detection procedure used? Is it the same for both RT streams? How many cases were excluded from the RT1 and RT3 datasets? This information may be useful to other users of these products. => add a short explanation and numbers at the end of Section 2.4 when you mention "data screening".

*We are now more specific in describing the data screening in Section 2.4 and we provide actual numbers of differences used for statistics computation for each compared pair of GNSS (NWM) solutions in Table 4 and 5. They allow the reader to find out how many values were excluded from each realized comparison.*

*Now mentioned in section 2.4.*
Add to the text inserted P7L17 the numbers of values used or compared (55 days x 243 stations x 288 estimates per day ~ 3.4x10^6 values). Since this applies to all three Tables, the information can be removed from section 3.2.
*The text was updated as suggested.*

*P16L1-2: I think both plots show results close to the expected behaviour: smaller residuals near the zenith and larger at low elevations. SINEL2 is preferred not because of the residual properties but because more accurate parameters are estimated (ZTD, coordinates, etc.).*
*You are right, but still there are differences in the distribution of residuals as we describe them in the manuscript. We don't see any reason for degradation close to the zenith. Therefore, we keep the text in its previous version.*
There is no degradation at the zenith, just a distribution of residuals consistent with the observation weighting model (more uniform). The sentence could be revised as: "It is closer to the expected behaviour…" => "Both show the expected behaviour…"
*We modified the paragraph as follows: "Above 30° elevation angle, the distribution of residuals is smoother for the SINEL2 compared to the EQUAL and more stable according to our experience with many other stations. This is particularly visible when comparing the distribution of residuals at the lowest and the highest elevation angles between variants though both generally follows the expected behaviour when considering errors in GNSS observations and models. These errors include contributions from the atmosphere, multipath, uncertainty of receiver antenna phase centre variations, lower signal-to-noise ratio, cycle slips; all usually increasing with the decrease of the observation elevation angle and with the smallest errors in the zenith direction."*

**Comments on manuscript version 9:**

P20L14: "Since no meteorological data providing any information about prevailing atmospheric conditions during the evaluated time period entered the GNSS data processing," This is not generally true, e.g. when VMF1 and ECMWF a priori ZHD/ZWD data are used. In your case GMF and GPT and used, so it is important in the conclusion section to be more specific about what is a general result and what is specific. => add "(because we used empirical mapping functions and a priori tropospheric delays)"
*We added the proposed text into the discussed sentence.*

Add the missing sections at the end of the manuscript (see https://www.annales-geophysicae.net/for_authors/manuscript_preparation.html):
• Code availability
• Data availability
• Sample availability
• Team list
• Author contribution
• Competing interests
See especially the Data policy of the journal:
https://www.annales-geophysicae.net/about/data_policy.html

*We added sections Data availability, Author contribution and Competing interests to the manuscript.*

[revised manuscript text omitted]

Figure 5 displays example distributions of carrier-phase post-fit residuals with respect to the elevation for the SINEL2 observation weighting (left panel), and without any weighting, i.e. EQUAL (right panel). While the residuals from the former are affected by the *mfg* only below 15° elevation, the residuals in the latter are affected at any elevation angles even close to the zenith direction. Above 30° elevation angle, the distribution of residuals is smoother for the SINEL2 compared to the EQUAL and more stable according to our experience with many other stations. This is particularly visible when comparing the distribution of residuals at the lowest and the highest elevation angles between variants though both generally follows the expected behaviour when considering errors in GNSS observations and models. These errors include contributions from the atmosphere, multipath, uncertainty of receiver antenna phase centre variations, lower signal-to-noise ratio, cycle slips; all usually increasing with the decrease of the observation elevation angle and with the smallest errors in 
[revised manuscript text omitted]

**Data availability**

GNSS data from the EUREF Permanent Network (EPN) stations are freely available through the anonymous FTP, e.g. from the EPN historical data centre at ftp://epncb.oma.be/pub/obs/ maintained by the Royal Observatory of Belgium. Other GNSS data were primarily collected for the purpose of the COST Action ES1206 (GNSS4SWEC project; see Douša et al., 2016) and cannot be distributed. The ECMWF is acknowledged for making publicly available ERA5 reanalysis fields that were generated using Copernicus Climate Change Service Information 2018 (https://www.ecmwf.int/en/forecasts/datasets/archive-datasets/reanalysis-datasets/era5).  The Global Forecast System data were provided by the National Centers for Environmental Prediction (http://nomads.ncdc.noaa.gov/data/gfsanl). All the validation results in the form of figures and tables for all types of presented comparisons and stations can be provided by request to michal.kacmarik@vsb.cz.

**Appendix A**

[revised manuscript text omitted]

**Author Contributions**

M.K., F.Z, J.D., G.D. and J.W. designed the whole study. M.K. calculated statistics and evaluated all results, prepared gradient maps and wrote major parts of the paper draft. J.D. and P.V. processed all the GNSS solutions in the G-Nut/Tefnut software, J.D. prepared Section 4 and revised the paper. F.Z. run NWM WRF and estimated the tropospheric parameters from NWM ERA5 and NWM WRF and prepared Section 2.3, Appendix A and Appendix B. K.B. prepared NWM ERA5 fields for the study and prepared NWM ERA5 outputs for Appendix A. G.D. and J.W. provided insights and contributed with careful reading and improving the text.

**Competing interests**

The authors declare that they have no conflict of interest.

[revised manuscript text omitted]